# FracFusionNet: A Multi-Level Feature Fusion Convolutional Network for Bone Fracture Detection in Radiographic Images

**DOI:** 10.3390/diagnostics15172212

**Published:** 2025-08-31

**Authors:** Sameh Abd El-Ghany, Mahmood A. Mahmood, A. A. Abd El-Aziz

**Affiliations:** Department of Information Systems, College of Computer and Information Sciences, Jouf University, Sakakah 72388, Saudi Arabia; mamahmood@ju.edu.sa (M.A.M.); aaeldamarany@ju.edu.sa (A.A.A.E.-A.)

**Keywords:** bone fracture, multi-level feature fusion network, deep learning, convolutional neural network, bone fracture multi-region X-ray dataset

## Abstract

**Background/Objectives:** Bones are essential components of the human body, providing structural support, enabling mobility, storing minerals, and protecting internal organs. Bone fractures (BFs) are common injuries that result from excessive physical force and can lead to serious complications, including bleeding, infection, impaired oxygenation, and long-term disability. Early and accurate identification of fractures through radiographic imaging is critical for effective treatment and improved patient outcomes. However, manual evaluation of X-rays is often time-consuming and prone to diagnostic errors due to human limitations. To address this, artificial intelligence (AI), particularly deep learning (DL), has emerged as a powerful tool for enhancing diagnostic precision in medical imaging. **Methods**: This research introduces a novel convolutional neural network (CNN) model, the Multi-Level Feature Fusion Network (MLFNet), designed to capture and integrate both low-level and high-level image features. The model was evaluated using the Bone Fracture Multi-Region X-ray (BFMRX) dataset. Preprocessing steps included image normalization, resizing, and contrast enhancement to ensure stable convergence, reduce sensitivity to lighting variations in radiographic images, and maintain consistency. Ablation studies were conducted to assess architectural variations, confirming the model’s robustness and generalizability across data distributions. MLFNet’s high accuracy, interpretability, and efficiency make it a promising solution for clinical deployment. **Results**: MLFNet achieved an impressive accuracy of 99.60% as a standalone model and 98.81% when integrated into hybrid ensemble architectures with five leading pre-trained DL models. **Conclusions**: The proposed approach supports timely and precise fracture detection, optimizing the diagnostic process and reducing healthcare costs. This approach offers significant potential to aid clinicians in fields such as orthopedics and radiology, contributing to more equitable and effective patient care.

## 1. Introduction

Bones are often seen as unchanging structures that provide support to the body. However, bone remodeling is a continuous process that occurs throughout a person’s life, largely driven by physiological needs [1]. Newborns typically have around 270 bones, which gradually fuse together, resulting in approximately 206 bones by adulthood. This skeletal structure includes the bones of the skull, vertebrae, rib cage, and the bones of the upper and lower limbs. The human skeletal system consists of 206 bones of various shapes and sizes, serving crucial functions such as providing structural support, protecting vital organs, and enabling movement. Among these bones, the femur is the largest, while the smallest is the stapes, located in the middle ear.

Bone diseases and disorders include a wide variety of medical conditions that affect the strength and health of the skeletal system. These issues can range from common problems like fractures and osteoporosis to rarer and more complicated diseases such as osteosarcoma and Paget’s disease [2]. BFs are common injuries that often result from falls, collisions, and other traumatic events [1,3]. The incidence of BFs worldwide is steadily increasing, influenced by aging populations and more urban lifestyles [3]. BF identification through radiographic imaging is a frequent procedure for patients experiencing either high- or low-energy trauma across multiple clinical environments, such as emergency rooms; urgent care facilities; and outpatient clinics like orthopedics, rheumatology, and family medicine. Overlooked fractures in radiographs are a leading factor in diagnostic inconsistencies between initial assessments made by nonradiologists or radiology residents and the final evaluations by board-certified radiologists, resulting in avoidable harm or delays in patient care [4,5]. Errors in interpreting fractures can account for as much as 24% of detrimental diagnostic mistakes observed in the emergency department [4]. Additionally, discrepancies in radiographic assessments of fractures tend to occur more frequently during evening and overnight shifts (from 5 PM to 3 AM), likely due to fatigue and the involvement of non-expert readers. In cases involving multiple injuries, the rate of overlooked injuries, such as fractures, can be significant, particularly in the forearm and hands (6.6%) and feet (6.5%) [6,7].

BFs are categorized based on the condition of the skin and soft tissue around the fracture into three types: open, closed, and displaced fractures, shown in Figure 1 [8]:Open Fracture: An open fracture refers to a broken bone that has broken through the skin’s surface. In severe cases, the bone may be visible externally.Closed Fracture: In contrast to open fractures, closed fractures do not break through the skin, resulting in a reduced risk of infection.Displaced Fracture: A displaced fracture occurs when there is a separation between the two ends of the broken bone, indicating that the edges are misaligned.

**Figure 1 diagnostics-15-02212-f001:**
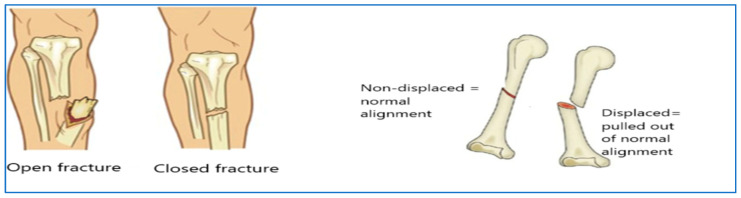
Types of bone fractures [8].

BFs are categorized by the extent of the break, distinguishing between partial and complete fractures [9]. In addition to the categories mentioned earlier, there are further subcategories based on the specific characteristics of BFs [10]. Figure 2 illustrates these subcategories of BFs, which include the following:Transverse fracture: The bone breaks across horizontally.Oblique fracture: A diagonal fracture that splits the bone.Spiral fracture: A fracture that twists around the bone.Comminuted fracture: When a bone shatters into multiple pieces, typically more than three.Segmental fracture: Involves two separate breaks in a bone, categorized as comminuted fractures.Linear fracture: The fracture aligns straight along the bone’s length.Greenstick fracture: A partial fracture where the bone bends but does not break completely.

**Figure 2 diagnostics-15-02212-f002:**
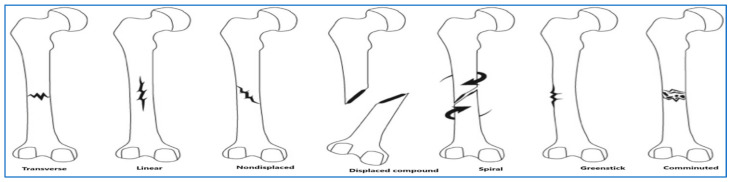
Categories of BFs [10].

Tibial fractures are the most frequently occurring type of bone fracture, especially among children, athletes, and older adults. They pose considerable diagnostic difficulties. Therefore, obtaining a swift and precise diagnosis is essential to improve the healing process’s effectiveness [9].

Fractures can arise from conditions that weaken bone integrity, like osteoporosis or bone cancer [11]. This is especially true for children, the elderly, and those who are physically active [3]. In addition, BFs occur when there is a disruption in the structure of the bone. These injuries are frequently caused by factors such as stress, accidents, osteogenesis imperfecta, osteoporosis, and cancer.

BFs present with distinct symptoms, though the type, location, and severity of the fracture can influence how they appear. Some fractures may be obvious (like an open fracture), while others (like stress fractures) may be subtle. The symptoms are pain, swelling, bruising, deformity, inability to bear weight or move, tenderness, and crepitus [12].

The success of any suggested treatment often hinges on a timely and accurate diagnosis. Therefore, obtaining a quick and precise diagnosis is crucial for improving the healing process’s effectiveness [3,13]. In real-world settings, radiologists and healthcare providers utilize X-rays, CT scans, and MRI images to determine the specific kind of fracture and to confirm if any fracture exists [3,9,13]. The Digital Imaging and Communications in Medicine (DICOM) standard facilitates the sharing of medical images [14].

After a fracture, it is crucial to seek treatment as soon as possible. An orthopedic surgeon will review an X-ray or CT scan to identify the fractured bone. Once the diagnosis is made, appropriate treatment is administered. The success of this treatment largely relies on the orthopedic surgeon’s experience and expertise. However, this manual process can be time-consuming, and access to specialists may be limited in remote areas [15].

X-ray imaging is a quick and non-invasive method primarily used to diagnose and confirm fractures, where the specific area of the body is subjected to X-ray radiation. The integration of radiation with advanced computer image processing has led to the widespread adoption of digital X-ray technology across various medical fields [16]. Although there are some limitations regarding image quality, X-rays are adequate for identifying fractures. By analyzing X-ray images and implementing suitable treatments, healthcare professionals can evaluate the presence and severity of injuries. X-ray images provide clear and detailed views of bone structures, aiding in the detection of complications related to fractures, such as misaligned bone fragments, involvement of joints, or damage to nearby tissues. However, it is important to carefully weigh the potential risks of radiation exposure, particularly when multiple X-rays are needed [11]. Digital X-ray imaging devices have become prevalent in healthcare due to their portability and advanced computerized image processing features.

Manually reviewing radiological images for fractures requires substantial time and effort. A fatigued radiologist might miss a fracture in an otherwise normal image. To address this issue, a computer-aided design (CAD) system can be employed to help identify concerning cases in radiological images and notify physicians [3].

AI has rapidly emerged as a powerful tool in enhancing the interpretation of medical images and increasing diagnostic accuracy. Precise detection of bone fractures is essential for determining appropriate treatment strategies and forecasting patient outcomes. Deep learning (DL) methods have been extensively utilized for feature extraction and fracture classification tasks [17]. Among these, convolutional neural networks (CNNs)—a key DL architecture—have attracted considerable interest for their ability to assist radiologists in identifying bone fractures. In many cases, DL models have demonstrated superior accuracy compared to human experts in analyzing medical images. This technology is applicable across multiple imaging modalities, including X-rays, CT scans, and MRIs, and supports the identification of various bone abnormalities, particularly fractures. Nevertheless, DL applications in this field still face challenges, especially in managing and processing the vast amount of data associated with bone imaging [18,19]. By delivering accurate image analysis, DL algorithms can facilitate timely and effective clinical decision-making. Their reliable detection capabilities not only enhance patient outcomes but also streamline the diagnostic workflow for healthcare professionals [20].

This research introduces a novel CNN model called MLFNet, which is designed to capture and combine both low-level and high-level image features. The model was tested on the BFMRX dataset for binary classification (fracture vs. non-fracture). The preprocessing steps involved normalizing images, adjusting their size, and enhancing contrast. These measures were taken to promote stable convergence, minimize sensitivity to changes in lighting in radiographic images, and ensure consistency. Ablation studies were performed to evaluate different architectural variations, confirming the model’s robustness and generalizability across various data distributions. MLFNet achieved an outstanding accuracy of 99.60% as a standalone model and 98.81% when integrated into hybrid ensemble architectures with five leading pre-trained DL models (DenseNet169, EfficientNet-B3, InceptionV3, MobileNet-V2, and ResNet-101). With its high accuracy, interpretability, and efficiency, MLFNet is a promising option for clinical use, facilitating timely and accurate fracture detection, optimizing the diagnostic process, and lowering healthcare costs. This approach has significant potential to assist clinicians in areas such as orthopedics and radiology, contributing to more equitable and effective patient care. Below is a summary of our research contributions.

We developed the MLFNet model, which extracts and combines features at multiple scales. This approach enhances the representation of bone structures and subtle fracture patterns across various image resolutions.We utilized multi-scale fusion techniques to capture both detailed local features and broader contextual information. This improvement allows the model to detect fractures of different shapes, sizes, and locations more effectively.The proposed model highlights potential fracture regions using attention-like responses from intermediate feature maps, aiding in weakly supervised localization.We evaluated the MLFNet using various radiographic datasets, demonstrating its superior ability to generalize and resist challenges found in real-world images, such as noise, contrast variations, and complex anatomical structures.We conducted detailed ablation studies comparing single-scale and multi-scale architectures. The results confirmed the effectiveness of integrating features at different resolution levels, leading to higher diagnostic accuracy.When benchmarked against five DL architectures (DenseNet169, EfficientNet-B3, InceptionV3, MobileNet-V2, and ResNet-101), the MLFNet approach excelled in binary classification, achieving an accuracy rate of 98.81% on the BFMRX dataset when combined with other DL models, and achieving 99.60% as an independent model on the same dataset.

The remainder of this paper is structured as follows: Section 2 provides a review of related work on BF diagnostic systems. Section 3 details the BFMRX dataset and the methodology employed in this research. Section 4 discusses two experimental outcomes of the proposed framework. Lastly, Section 5 offers concluding remarks and summarizes the key findings.

## 2. Literature Review

The detection of BFs remains a key focus in medical image analysis research, with multiple studies addressing the problem using diverse methodologies. For example, Yadav, D.P. et al. [15] introduced an innovative multi-scale feature fusion approach that combined a CNN with an enhanced Canny edge detection algorithm to differentiate between fractured and healthy bone images. The hybrid Scale Fracture Network (SFNet) was a unique two-scale sequential DL model designed for efficient BF diagnosis, requiring less computational time than other leading deep CNN models. A key aspect of Yadav, D.P. et al. [15] was the application of an improved Canny edge algorithm, which helps identify edges in images to pinpoint fracture regions. Subsequently, grayscale images along with their corresponding Canny edge images are utilized for training and evaluation within the proposed hybrid SFNet. Additionally, the authors compared the performance of SFNet with Canny (SFNet + Canny) against other advanced deep CNN models using a bone image dataset. Our findings demonstrated that SFNet with the Canny algorithm achieved the highest accuracy, F1-score, and recall rates of 99.12%, 99%, and 100%, respectively, in diagnosing bone fractures.

Aldhyani, T. et al. [9] created an automated method for detecting fractures using the BFMRX dataset from Kaggle, which includes a detailed collection of 10,580 radiographic images. This research promoted the application of DL techniques such as VGG-16, ResNet152-V2, and DenseNet-201 for diagnosing and identifying bone fractures. The results of the experiments showed that the suggested method effectively identified and classified different types of fractures. The proposed system, which utilized DenseNet-201 and VGG-16, achieved a validation accuracy rate of 97%.

Pillai, R. et al. [21] explored the application of the VGG-16 transfer learning model for the automatic diagnosis of BFs using X-ray images. The primary aim was to enhance both the accuracy and efficiency of fracture detection. They divided the BFMRX dataset, which contains 10,578 annotated images labeled as either Fractured or Not Fractured, into three segments: training (9243 images), validation (829 images), and testing (506 images). The VGG16 model was fine-tuned on the training set, employing data augmentation to improve its performance under varying imaging conditions. The evaluation of the model’s performance involved tracking accuracy and loss metrics during both the training and validation phases. The ultimate evaluation on the testing set yielded an accuracy of 98.22%.

Shandilya, G. et al. [22] utilized a multi-region X-ray dataset of BFs to present a comprehensive method for detecting and classifying fractures through a novel CNN model. The BFMRX dataset comprises 10,580 X-ray images showcasing both fractured and non-fractured bones from various anatomical regions, including the lower limb, upper limb, lumbar spine, hips, and knees. The proposed CNN model achieved an overall accuracy of 97.85%, along with precision and recall rates of 98%.

Alsufyani, A. [23] evaluated five different models: one custom-designed CNN and four pre-trained models—AlexNet, DenseNet-121, ResNet-152, and EfficientNet-B3. The models were trained using the BFMRX dataset of 10,581 X-ray images, classified as either fractured or non-fractured. Their performance was assessed through metrics such as accuracy, precision, recall, and F1-score. Among the models tested, EfficientNet-B3 delivered the most impressive results, achieving an accuracy of 99.20% and perfect recall, indicating its significant potential for application in clinical settings. ResNet-152 and the custom CNN also showed strong performance, albeit with slightly lower accuracy.

Alex, R. and Rosmasari [24] presented an automated DL method for classifying bone fractures into two categories using a pre-trained ResNet18 architecture. The model was trained and validated on the BFMRX dataset comprising 10,580 images, which were divided into fractured and non-fractured categories. To enhance the model’s generalization capabilities, data augmentation techniques such as rotation and horizontal flipping were employed during the preprocessing stage. The resulting model achieved a validation accuracy of 97.59%, showcasing high rates of true positives (TPs) and true negatives (TNs), as evidenced by confusion matrix analysis.

Shandilya, G. et al. [25] employed an enhanced AlexNet model to introduce a thorough approach for diagnosing fractures across multiple regions. The BFMRX dataset utilized comprises X-ray images of both fractured and non-fractured bones from various anatomical regions, including the knees, hips, lumbar region, lower limbs, and upper limbs. To distinguish between fractured and non-fractured X-ray images, the AlexNet CNN, optimized for binary classification, was implemented. The improved AlexNet model achieved a remarkable classification accuracy of 97.12%.

Kumar, A. et al. [26] utilized the BFMRX dataset consisting of 10,581 radiographic images that encompass a wide range of anatomical areas, including the lower limb, upper limb, lumbar spine, hips, and knees. The dataset was organized into 4606 images identified as fractured and 4640 images marked as not fractured for training purposes, along with 337 fractured and 492 non-fractured images for validation, and 238 fractured and 268 non-fractured images for testing. They implemented the AlexNet DL framework, utilizing sophisticated data augmentation and preprocessing methods to improve model efficacy. The model attained an accuracy rate of 96%.

Chauhan, S. [27] categorized the BFMRX dataset of 10,580 radiographic X-ray images utilizing a CNN−AlexNet model, covering various anatomical regions such as the lower limb, upper limb, lumbar region, hips, and knees. The dataset was meticulously organized into training (9246 images), validation (828 images), and test (506 images) subsets, each containing both fractured and non-fractured images. The CNN−AlexNet model was trained on this extensive dataset to accurately differentiate between fractured and non-fractured bone X-rays. The model achieved an accuracy of 96% on the test set.

Uddin, A. et al. [28] utilized the BFMRX dataset that comprised 4148 labeled images along with 10,581 X-ray images categorized as either ‘fractured’ or ‘not fractured’ for the purpose of BF classification. They focused on employing DL models such as CNN, ConvNeXt, ViT, MobileViT, VGG-16, and VGG-19 to effectively identify BFs and assess their severity. Additionally, YOLOV8 was employed for fracture diagnosis, and image processing techniques enhanced the quality of noisy X-ray images, achieving a mean Average Precision (maPA50) of 0.995 and 0.991 for maPA50-95. In terms of classification performance, MobileViT attained an accuracy of 99%.

The limitations of the previously mentioned research are as follows:The previously cited studies primarily utilized conventional DL models, with the exception of Yadav, D.P. et al. [15]. In our approach, we employed a custom DCNN to extract and integrate features at multiple scales. This technique improves the representation of bone structures and subtle fracture patterns across different image resolutions.While Yadav, D.P. et al. [15] reported an accuracy rate of 99.21% in one experiment, we conducted two experiments that achieved an accuracy rate of 98.81% on the BFMRX dataset when integrated with other DL models, and reached 99.60% as an independent model on the same dataset.

## 3. Research Methods and Materials

### 3.1. BFMRX Dataset

The BFMRX dataset used in our study was sourced directly from Kaggle [29] and was utilized as provided by the dataset authors, without any alterations to the original training, validation, or test splits. According to the dataset description on Kaggle, the splits were prepared by the dataset creators, ensuring that images from the same patient do not appear in multiple subsets. The BFMRX dataset from Kaggle [29] contains X-ray images of both fractured and non-fractured regions of the body, encompassing areas such as the lower limb, upper limb, lumbar spine, hips, and knees. The sizes of the X-ray images differed, requiring preprocessing steps to normalize the data before it could be fed into the models. This dataset includes a total of 10,580 radiographic images (X-ray data). The BFMRX dataset is organized into an 87% training set (9246), a 5% testing set (506), and an 8% validation set (828). Table 1 and Figure 3 and Figure 4 display the allocation of images categorized into fractured and non-fractured groups. Figure 5 shows samples from the BFMRX dataset.

### 3.2. Methodology

To diagnose BFs from radiographic images, we introduced an innovative CNN model named MLFNet. This model was designed to effectively capture and integrate both low-level and high-level features from images. MLFNet utilizes parallel convolutional branches to extract a diverse range of semantic representations, and it implements the Grad-CAM technique to improve interpretability by emphasizing the areas of fractures. The model underwent evaluation on the BFMRX dataset for binary classification. Preprocessing included normalizing and resizing images to maintain consistency. The architecture of the proposed MLFNet model is depicted in Figure 6. The fine-tuning process for the six DL models is described in Algorithm 1. The steps involved in the proposed MLFNet model are summarized as follows:


**Preprocessing Stage**


Input: 128 × 128 × 3 X-ray image.Normalization/Enhancement: Standardizes pixel values and improves image quality.


**Feature Extraction/Fusion Stage**



**Block 1 (SFNetBlock with 64 filters):**
▪Two parallel SFLayers (each: Conv2D → BatchNorm → MaxPooling)▪Concatenates outputs to capture low-level edge features.
Batch Normalization: Stabilizes activations after Block 1
**Block 2 (SFNetBlock with 128 filters):**
▪Dual SFLayers with concatenation for mid-level semantic features
Batch Normalization: Normalizes Block 2 outputsDropout (*p* = 0.3): Randomly deactivates 30% neurons to prevent overfitting
**Block 3 (SFNetBlock with 256 filters):**
▪Dual SFLayers with concatenation for abstract high-level features
Batch Normalization: Normalizes Block 3 outputsDropout (*p* = 0.3): Randomly deactivates 30% neurons to prevent overfitting.


**Classification Stage**


Flatten: Converts 3D feature maps to 1D vectorDense Layers:Fully connected layer (256 units) → Dropout → Layer (128 units) → Dropout
▪Learns high-level representations for decision-makingOutput Layer:▪Sigmoid-activated dense layer (1 unit) for binary classification (fracture vs. no fracture)

**Figure 6 diagnostics-15-02212-f006:**
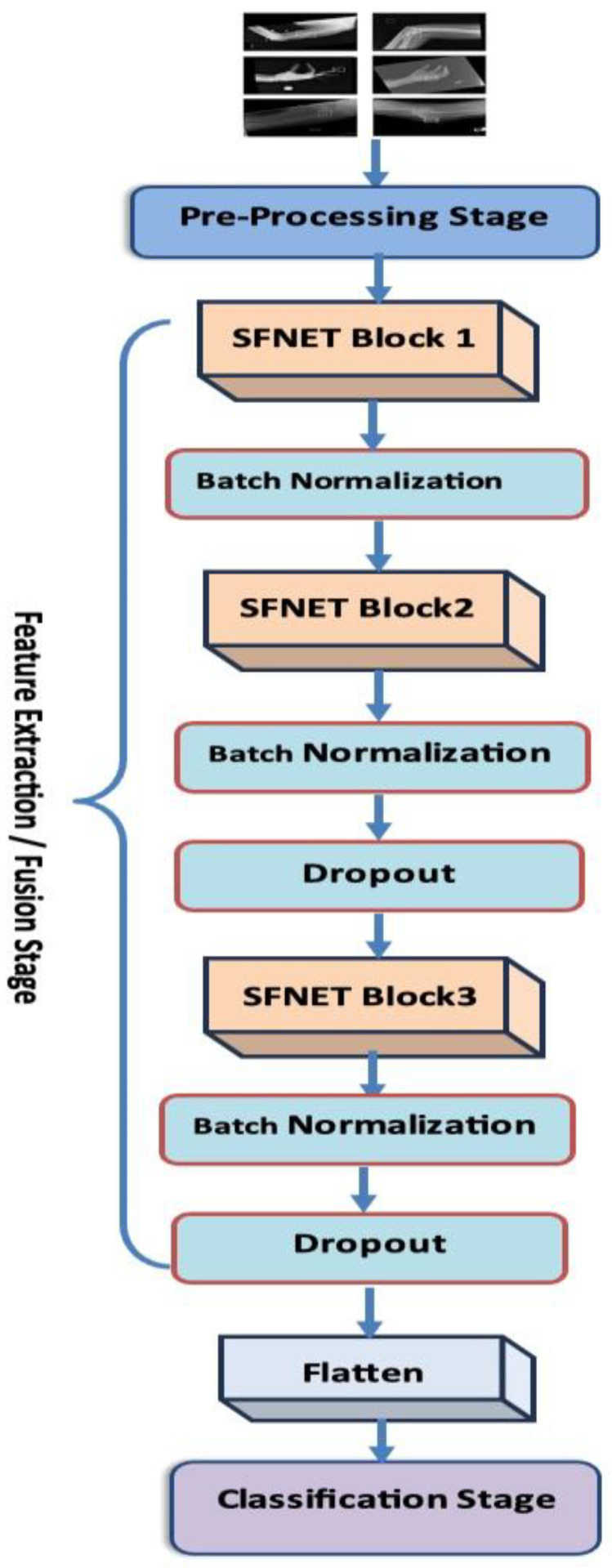
The proposed workflow.

**Algorithm 1:** Optimization of the DL models1**Input** → BFMRX dataset as BF.2**Output** ← Optimized DL Models.3
**BEGIN**
4      **STEP 1: Preprocessing of X-rays**5           **FOR EACH** X-ray **IN** the BF **DO**6                *Resize* X-ray to 128 × 128 × 3 pixels.7                *Normalize* X-ray from the range [0, 255] to [0, 1].8                *Enhance* contrast.9           **END FOR**10      **STEP 2: Feature Extraction/Fusion**11              *Dual* SFLayers with concatenation for mid-level semantic features by Block 2.13              *Normalizes* Block 2 outputs.14              *Dual* SFLayers with concatenation for abstract high-level features by Block3.15      **STEP 3: Classification Stage**16          *Convert* 3D feature maps to 1D vector by Flatten17          *Learns* high-level representations for decision-making18          *Apply* Sigmoid-activated dense layer (1 unit) for binary classification (fracture vs. no fracture)19
**End**


#### 3.2.1. Data Preprocessing

Given the significant impact of data preprocessing on the performance of DL models, it was prioritized as a foundational step in our BF detection pipeline. To ensure optimal input quality for training, we applied standardized preprocessing techniques to the radiographic images from the BFMRX dataset. Three primary operations were performed:Resizing: All the X-ray images were resized to a uniform dimension of 128 × 128 × 3 pixels. This resolution is compatible with the input requirements of most convolutional neural networks (CNNs), particularly those based on pretrained architectures, and helps reduce computational complexity.Normalization: Pixel intensity values, originally ranging from 0 to 255, were scaled to a normalized range of [0, 1]. This step improves training stability and accelerates convergence by mitigating internal covariate shift.Contrast Enhancement: Image contrast was enhanced to reduce sensitivity to lighting variations in radiographic scans and to improve the visibility of subtle fracture patterns. Figure 7 shows the images before and after contrast enhancement.

These preprocessing procedures ensured consistency across the dataset and prepared the input images for effective training and accurate BF classification.

#### 3.2.2. SFNet

The SFNet architecture (short for Semantic Feature Network) is an advanced CNN designed to extract and combine semantic features at multiple scales and abstraction levels using parallel branches and skip connections. Its key objective is to capture both local and global contextual information, which is critical for tasks like bone fracture detection, where both subtle and large-scale features may exist. It is used for semantic segmentation tasks, with key applications in medical image analysis [30]. SFNet introduces Semantic Flow (SF) layers to align and fuse these multi-level features without requiring heavy computations. SFNet has the following layers:

Input Layer: Accepts radiographic images (e.g., grayscale or RGB), typically resized to a standard dimension such as 128 × 128 × 3.

Preprocessing Layer (optional but often applied): Normalization and enhancement (e.g., contrast adjustment) to improve convergence and robustness.

**SFLayer**: 

A fundamental building block of SFNet that consists of the following:Convolutional Layer (Conv2D): For feature extraction.Batch Normalization: To stabilize training.Activation Function: Usually ReLU.MaxPooling Layer: For downsampling and translation invariance.


**SFNet Block**


A multi-path feature fusion unit constructed by stacking two SFLayers followed by a concatenation operation. This allows simultaneous extraction of the following:Low-level features (edges, textures).High-level features (shapes, patterns).Output feature maps are enriched by merging different semantic levels.


**Deep Blocks:**
Repeated application of SFNet Blocks with increasing depth (e.g., more filters, smaller spatial dimensions).Enables hierarchical representation learning.Some variants include residual connections or dilated convolutions to preserve spatial resolution.



**Flattening and Dense Layers:**
Final feature maps are flattened into a 1D vector.Passed through fully connected (dense) layers with ReLU activations.Dropout may be applied for regularization.



**Output Layer:**


A final dense layer with either of the following:Sigmoid activation for binary classification (fracture vs. non-fracture) or Softmax activation for multi-class problems.

#### 3.2.3. MLFNet

The backbone of the proposed MLFNet model architecture consists of three stacked SFNet blocks, each designed to extract hierarchical features through a dual-path convolutional approach. Each SFNet block is composed of two parallel SFLayers, where each SFLayer includes the following:A 2D Convolutional Layer with filters of size 3 × 3, used to capture spatial features from the image.A batch normalization layer, which accelerates training and stabilizes learning by normalizing activations.A MaxPooling layer, applied to downsample the feature maps and retain dominant features.

The outputs of the two parallel paths in each block are concatenated along the channel dimension to enrich the representational capacity of the model.

The SFNet blocks are configured as follows:SFNet Block 1: 64 filters → captures low-level features like edges.SFNet Block 2: 128 filters → captures mid-level anatomical features.SFNet Block 3: 256 filters → captures high-level semantic and abstract features.

Batch normalization and dropout layers are applied after each block to enhance generalization and prevent overfitting.

Following feature extraction, the resulting tensor is flattened into a one-dimensional vector. This vector is then passed through a series of fully connected dense layers:A 256-unit dense layer with ReLU activation.A dropout layer with a rate of 0.3.A 128-unit dense layer with ReLU activation.Another dropout layer with a rate of 0.3.

These layers enable high-level decision-making based on the extracted features. They further abstract and refine the learned patterns before classification.

The final output is generated using a single-unit dense layer with a sigmoid activation function, which returns a binary value:A value of 1 indicates a fracture is present.A value of 0 indicates a normal (non-fractured) condition.

The architecture is optimized for binary classification using the binary cross-entropy loss function. Table 2 and Figure 8 show the architecture of the proposed MLFNet.

#### 3.2.4. Five Deep Learning Models

In this study, five additional deep learning models were utilized to evaluate the experimental outcomes. These models include DenseNet-169 [31], EfficientNet-B3 [32], Inception-V3 [33], MobileNet-V2 [34], and ResNet-101 [35]. The comparisons of these models are detailed in Table 3 and Table 4. Table 3 highlights the architectural differences between MLFNet and the other models (DenseNet-169, EfficientNet-B3, Inception-V3, MobileNet-V2, and ResNet-101). Table 4 provides information on the input shape, total number of parameters, trainable parameters, non-trainable parameters, and the number of layers for the five additional models.

## 4. Evaluation and Analysis

### 4.1. Evaluated Performance Metrics

The performance of the six DL models—MLFNet, DenseNet-169, EfficientNet-B3, Inception-V3, MobileNet-V2, and ResNet-101—was assessed using the equations outlined in Equations (1)–(7).(1)$$Accuracy=\frac{\left(TP+TN\right)}{\left(TP+FP+TN+FN\right)}$$(2)$$Precision=\frac{TP}{\left(TP+FP\right)}$$(3)$$Sensivity=\frac{TP}{\left(TP+FN\right)}$$(4)$$Specifity=\frac{TN}{\left(TN+FP\right)}$$(5)$$F1\mathrm{-score}=\frac{2\times Precision\times Recall}{Precision+Recall}$$(6)$$FalseNegativeRate(FNR)=\frac{FN}{TP+FN}$$(7)$$NegativePredictiveValue(NPV)=\frac{TN}{TN+FN}$$

In classification tasks, TPs refer to instances where the model accurately identifies positive cases that are indeed positive in the ground truth, while TNs represent correctly identified negative cases. Conversely, a false positive (FP) occurs when the model incorrectly labels a negative instance as positive, such as identifying a tumor where none exists. A False Negative (FN) arises when the model fails to detect a positive case, thereby classifying an individual with a tumor as healthy. The Negative Predictive Value (NPV) quantifies the proportion of true negative predictions among all negative predictions, reflecting the model’s ability to correctly identify non-fractured cases. The False Negative Rate (FNR) measures the proportion of actual positive cases that were misclassified as negative. Together, TP + FN gives the total number of actual positive instances in the dataset. These metrics are essential for evaluating the diagnostic reliability of machine learning models in medical imaging contexts [36].

### 4.2. Testing Environment

In this research, we conducted two experiments to assess the classification efficacy of the MLFNet model using the BFMRX dataset. The experiments were executed on the Kaggle environment. For the implementation, we employed Python 3 along with TensorFlow, a DL framework created by Google. TensorFlow is widely utilized in artificial intelligence and machine learning applications because of its versatility and capability for developing and deploying neural networks. All hyperparameter tuning, architecture selection, and ablation studies were performed solely using the training and validation sets. Moreover, all ensemble weighting coefficients and decision threshold values were optimized using only the validation set. The test set was reserved for the final evaluation and was not used in any way during the tuning process. Additionally, the specific hyperparameters used in the experiments—such as learning rate, batch size, number of epochs, and optimizer type—are presented in Table 5.

### 4.3. The MLFNe as a Standalone Experiment Performance Assessment

The main objective of the two experiments was to identify BFs, with the intention of enhancing patient outcomes, streamlining the diagnostic process, and significantly lowering both patient time and costs. In these experiments, 87% of the BFMRX dataset, which comprises 9246 images, was utilized for training purposes. The remaining 5%, or 506 images, was set aside for testing, while 8%, equating to 828 images, was allocated for validation. The BFMRX dataset has two classes: fractured and non-fractured. Both experiments emphasized transfer learning to pre-train six DL models—MLFNet, DenseNet-169, EfficientNet-B3, Inception-V3, MobileNet-V2, and ResNet-101. During the initial transfer learning phase, we conducted supervised pre-training using the ImageNet dataset for these six models. Following this, we executed the fine-tuning phase using the training sets derived from the BFMRX dataset. At the conclusion of each experiment, we used the testing set of the BFMRX dataset and applied the evaluation metrics (as outlined in Equations (1)–(7)) to assess the performance of the six DL models.

In the initial experiment, we employed the MLFNet model independently and assessed its performance using the testing subset of the BFMRX dataset. The results from six DL models, along with their evaluation metrics, are presented in Table 6, Table 7 and Table 8. The average accuracy for each model is as follows: MLFNet achieved an accuracy of 99.60%, DenseNet-169 reached 95.06%, EfficientNet-B3 obtained 93.68%, Inception-V3 recorded 94.27%, MobileNet-V2 reached 97.04%, and ResNet-101 logged 91.90%. These findings indicate that MLFNet exhibited the highest accuracy among all the models assessed.

Table 6 shows exceptional performance for the **MLFNet** model in distinguishing between fractured and non-fractured cases. The average accuracy was an impressive 99.60%, indicating the model’s nearly flawless classification capability. Both the average specificity and recall were high at 99.63%, demonstrating the model’s strong ability to accurately identify TNs and TPs. The average FNR was notably low at just 0.37%, while the NPV and precision were both at 99.58%, indicating that the model made very reliable predictions. The average F1-score was 99.60%, confirming a nearly perfect balance between precision and recall.

For the fractured class, the model achieved 100% recall, meaning it accurately identified all the fractured cases with 0% false negatives. It also recorded 99.25% specificity, 99.17% precision, and an F1-score of 99.58%, reflecting high confidence and accuracy in detecting fractures. The NPV for this class was perfect at 100%, further supporting its strong performance. For the non-fractured class, the model reached 100% specificity and 99.25% recall, with a slightly higher FNR of 0.75%. Nevertheless, it achieved 100% precision, 99.17% NPV, and an F1-score of 99.63%, demonstrating consistent and dependable classification. Overall, the MLFNet model showed superior accuracy and robustness, making it highly suitable for clinical fracture detection tasks.

The average accuracy of the **DenseNet-169** model was 95.06%, meaning the model accurately classified most instances. The average specificity and recall were both 95.03%, showing a balanced ability to identify TNs and TPs. The average FNR was 4.97%, indicating a modest number of missed positive cases. Additionally, the average NPV and precision were both 95.05%, demonstrating reliable predictive capability. However, the reported average F1-score of 48.15% seemed to be a typographical or calculation error, as it was inconsistent with the other metrics and the individual class F1-scores.

For the fractured class, the model achieved 95.06% accuracy, with a specificity of 95.52% and a recall of 94.54%, reflecting decent performance in identifying fractured cases. The FNR was 5.46%, and the NPV, precision, and F1-score were 95.17%, 94.94%, and approximately 95.00%, respectively (rounded from the value labeled as 0.95, which likely represented 95%). For the non-fractured class, the model maintained the same accuracy of 95.06%, with a specificity of 94.54%, a recall of 95.52%, an FNR of 4.48%, and strong supporting values in NPV (94.94%), precision (95.17%), and F1-score (95.34%). Overall, the DenseNet-169 model demonstrated consistent and balanced performance, though it was slightly lower than other evaluated models, and the average F1-score needs correction to reflect accurate results.

The average accuracy of the **EfficientNet-B3** model was 93.68%, indicating a generally reliable classification rate. Both the average specificity and recall were 93.98%, suggesting a balanced ability to identify TNs and TPs. However, the average FNR was 6.02%, which is notably higher than that of previously analyzed models. On the positive side, the NPV and precision averaged 93.94%, reflecting strong overall predictive reliability. The average F1-score was 93.68%, confirming a fair balance between precision and recall.

For the fractured class, the model achieved 93.68% accuracy, with a recall of 99.16% and a very low FNR of 0.84%, indicating high sensitivity to fractured cases. However, the specificity dropped to 88.81%, and the precision was 88.72%, suggesting a higher incidence of false positives in this category. The NPV was very high at 99.17%, and the F1-score was 93.65%, highlighting strong recall but moderate precision.

In contrast, for the non-fractured class, the model maintained the same accuracy of 93.68%, but with a much higher FNR of 11.19% and lower recall at 88.81%, indicating that it missed more normal cases. Nonetheless, it achieved very high precision at 99.17%, a specificity of 99.16%, and an F1-score of 93.70%, illustrating a reversal of performance trade-offs compared to the fractured class. Overall, while EfficientNet-B3 performed well, it demonstrated a clear precision–recall trade-off between classes, which may require adjustment based on clinical priorities, such as minimizing missed fractures.

The average accuracy of the **Inception-V3** model reached 94.27%, indicating a high rate of correct predictions. The model also achieved balanced average specificity and recall values of 94.19%, demonstrating its effectiveness in identifying both TNs and TPs. The average FNR was 5.81%, which is relatively low, while both the NPV and precision averaged 94.31%, highlighting consistent predictive reliability. The average F1-score of 94.24% confirmed a good balance between precision and recall across both classes.

For the fractured class, the model achieved 94.27% accuracy, with a specificity of 95.52% and a recall of 92.86%. This suggests that while it accurately identified most fractured cases, a small number were missed. The FNR was 7.14%, slightly higher than ideal, and the precision, NPV, and F1-score were 94.85%, 93.77%, and 93.84%, respectively—indicating solid performance with room for improvement. For the non-fractured class, the model maintained 94.27% accuracy, with 92.86% specificity and 95.52% recall, reflecting better sensitivity but slightly lower specificity. The FNR was 4.48%, with precision, NPV, and F1-score values of 93.77%, 94.85%, and 94.64%, respectively. Overall, the Inception-V3 model performed reliably but showed slight trade-offs between recall and specificity depending on the class, suggesting potential for further fine-tuning to enhance generalization.

The **MobileNet-V2** model achieved an average accuracy of 97.04%, indicating high correctness in its predictions. The average specificity and recall were also 97.04%, meaning the model was equally effective in identifying TNs and TPs. The average FNR was low at 2.96%, while both the NPV and precision were 97.02%, demonstrating consistent and reliable predictive performance. The average F1-score of 97.03% confirmed the model’s well-balanced trade-off between precision and recall.

For the fractured class, the model achieved 97.04% accuracy, with a specificity of 97.01% and a recall of 97.06%, showcasing its excellent ability to correctly identify fractured cases. The FNR was 2.94%, and the NPV, precision, and F1-score were 97.38%, 96.65%, and 96.86%, respectively, indicating slightly better performance in identifying TPs than in avoiding FPs. For the non-fractured class, the accuracy remained at 97.04%, with a specificity of 97.06%, recall of 97.01%, and FNR of 2.99%. The precision was 97.38%, and the NPV was 96.65%, resulting in an F1-score of 97.20%, which showed a slightly higher harmonic balance compared to the fractured class. Overall, the MobileNet-V2 model demonstrated robust, consistent performance with minimal class imbalance and a high level of reliability.

The **ResNet-101** model achieved an average accuracy of 91.90%, indicating it correctly classified most samples. Both the average specificity and recall were 92.23%, suggesting a generally effective ability to identify TNs and TPs. The average FNR was 7.77%, indicating some missed positive cases. The NPV and precision were both 92.25%, while the average F1-score was 91.90%, demonstrating a reasonably strong but not optimal balance between precision and recall.

In the fractured class, the model achieved 91.90% accuracy, with a specificity of 86.57% and a recall of 97.90%. This indicates a high sensitivity to detecting fractures, albeit at the expense of a higher false positive rate. The FNR was low at 2.10%, and the precision, NPV, and F1-score were 86.62%, 97.89%, and 91.91%, respectively—highlighting strong recall but slightly weaker precision. For the non-fractured class, the model also maintained 91.90% accuracy, but the performance pattern was reversed: it achieved higher specificity at 97.90% and precision at 97.89%, while the recall dropped to 86.57%, and the FNR rose to 13.43%, suggesting the model missed more non-fractured cases. The NPV was 86.62%, and the F1-score was 91.88%. Overall, while the model performed adequately, it exhibited a recall–precision trade-off between classes, and there is room for improvement to reduce false positives for fractured cases and false negatives for non-fractured ones.

The analysis of computational efficiency and resource consumption among the six models reveals distinct variations, as shown in Table 8. **MLFNet** stands out as the lightest option, achieving the quickest training time of 81.44 s, the fastest inference at 12.03 s, and the lowest memory use at approximately 10.3 GB, even though it has 35.07 M parameters. On the other hand, ResNet-101 exhibits the highest computational demands, taking 291.27 s for training, 28.61 s for inference, and requiring the most memory at around 20.9 GB, all while having over 43 million parameters, which underscores its substantial architecture. DenseNet-169 also requires considerable resources, with the longest training time of 419.18 s and a notable parameter count of about 13 million, although its inference speed is comparatively better than that of ResNet-101. EfficientNet-B3 offers a balanced approach, with a moderate training duration of 305 s, lower memory usage at 15.9 GB, and around 11 million parameters, making it an efficient choice with less complexity compared to larger models. Inception-V3 positions itself in the middle, needing about 161 s for training and 19.5 s for inference, alongside a relatively high parameter count of approximately 22 million and memory usage of about 17.4 GB. Lastly, MobileNet-V2 is notable for its efficiency, achieving a training time of 114.57 s, a fast inference of around 14.88 s, and the lowest parameter count at roughly 2.6 million, despite slightly elevated memory consumption of around 18.4 GB.

In summary, MLFNet and MobileNet-V2 are the most resource-efficient models, while ResNet-101 and DenseNet-169 demand significant computational power, with EfficientNet-B3 and Inception-V3 providing a balanced compromise between performance and resource allocation.

Figure 9 illustrates the training and validation performance of the six DL models over 30 epochs, displaying both loss (on the left) and accuracy (on the right) curves. Extreme instabilities were observed in DenseNet-169, EfficientNet-B3, Inception-V3, MobileNet-V2, and ResNet-101, where the validation loss exhibited abnormally high spikes (up to 1 × 10^13^ in DenseNet-169 and similarly large values in others). These were traced to numerical instabilities caused by high initial LRs and the absence of gradient clipping, occasionally compounded by logging artifacts. Training configurations were adjusted with reduced learning rates, warm-up scheduling, and gradient clipping to mitigate these effects.

For the **MLFNet** model, the training loss steadily decreased from around 0.48 to nearly zero, while the validation loss also dropped sharply in the early epochs before stabilizing with minor fluctuations between epochs 10 and 25, remaining low overall. This pattern indicates that the model effectively learned from the training data and maintained good generalization to unseen validation data without significant overfitting.

Regarding accuracy, the MLFNet model experienced a rapid increase during the initial epochs. Training accuracy rose from approximately 75% to nearly 100%, while validation accuracy followed a similar trend, quickly surpassing 95% and remaining stable with minor variations throughout the remaining epochs. The small gap between training and validation accuracy, along with the low and stable validation loss, suggests that the model achieved excellent generalization and high predictive performance. Overall, the close alignment between training and validation metrics demonstrates a well-optimized and robust model, capable of delivering consistent results on both seen and unseen data.

The training and validation loss plot (left) of the **DenseNet-169** model across 30 epochs revealed a significant issue. While the training loss steadily decreased toward zero, the validation loss experienced a drastic spike in the first epoch, reaching approximately 1.4 × 10^13^. This likely indicated numerical instability or error, such as exploding gradients or division by zero. Following this spike, the validation loss quickly dropped to near zero and remained flat, which is not typical behavior and suggests a possible flaw in the loss computation or logging.

In the accuracy plot (right), the training accuracy consistently improved from about 78% to nearly 99%, indicating effective learning. However, the validation accuracy was quite erratic during the first 10 epochs, fluctuating between 40% and 95%. This instability likely stemmed from the issues observed in the validation loss. After the initial fluctuations, the validation accuracy stabilized and aligned more closely with the training accuracy, remaining above 90%, indicating improved generalization. Overall, while the training behavior was smooth, the unusual patterns in validation loss and early validation accuracy pointed to significant early instability in the model training process, possibly due to data irregularities, improper initialization, or learning rate issues. Addressing these factors is essential for ensuring reliable evaluation.

For the **EfficientNet-B3** model over 30 epochs, on the left, you can see the loss curves, and on the right, the accuracy curves. The training loss (in red) remained consistently low and stable throughout the epochs, indicating smooth convergence during training. In contrast, the validation loss (in green) exhibited significant instability, with sharp spikes, particularly around epochs 12, 17, and 19, where the loss exceeded 5000. These sudden increases point to severe overfitting, numerical instability, or data-related issues, such as outliers or mislabeled validation samples.

Looking at accuracy, the training accuracy rose steadily from approximately 80% to nearly 100%, showing effective learning on the training set. However, the validation accuracy fluctuated sharply, especially in the early and middle epochs, and dropped significantly around epoch 21, coinciding with the spike in validation loss. Despite this drop, validation accuracy later recovered and closely aligned with training accuracy, stabilizing above 95% by the final epochs.

Overall, the comparison indicated a major gap between training and validation loss, suggesting that while the model performed well on the training data, its ability to generalize to the validation set was inconsistent and occasionally unreliable. These issues highlight potential problems with model regularization, learning rate tuning, or the quality of the validation data, which need to be addressed for more robust and stable performance.

The **Inception-V3** demonstrates effective but differing learning patterns over 30 epochs. While there was a reduction in loss and an increase in accuracy, these changes showed an inverse relationship, leading to significant overfitting. Training loss dropped sharply from 1.2 to nearly 0.0, following a near-exponential decay that indicated strong optimization. Validation loss decreased more gradually from 1.0 to 0.4, leveling off after epoch 15 with little further improvement. At the same time, training accuracy increased significantly from 50% to 90%, while validation accuracy rose from 60% to 80%, stabilizing after epoch 20.

The inverse relationship was most evident during the early training stages (epochs 0–10):Loss dropped by 67% (from 1.0 to 0.33).Accuracy rose by 30% (from 55% to 85%).

However, a noticeable divergence appeared after epoch 15:Training loss continued to fall to 0.0.Validation loss remained at 0.4 (five times higher than training loss).Training accuracy reached 90%.Validation accuracy plateaued at 80%.

This resulted in a 10% accuracy gap and a loss gap of 0.4, highlighting substantial overfitting. Validation metrics showed early stabilization at epoch 15 (loss 0.4, accuracy 80%), while training metrics kept improving for another 15 epochs without similar gains in validation. Nonetheless, the coordinated progress in the early phase confirmed that the initial weight updates effectively contributed to performance improvements, with validation accuracy peaking at 80%—a level suitable for diagnostic use. The stalled validation metrics after epoch 15 suggested that early stopping at this point could prevent overfitting while ensuring optimal generalization.

The **MobileNet-V2** model shows a troubling gap between the loss and accuracy metrics throughout the training epochs. The training and validation loss rose sharply from an initial value of around 0.5 to about 30 by epoch 30, indicating a serious problem with model convergence. This steep increase suggests potential issues such as a learning rate that is too high, model instability, or mismatched data.

On the other hand, both training and validation accuracy dropped significantly, falling from 0.8 to 0.4 within the first three epochs before leveling off at a suboptimal performance level. The rapid decline in accuracy corresponded with the rising loss, confirming that the model did not succeed in generalizing or learning useful patterns. The ongoing gap between training and validation accuracy pointed to overfitting, but the main problem appeared to be severe model divergence, as both metrics worsened together. This inverse relationship—where loss increased while accuracy decreased—highlighted a fundamental failure in the learning process.

The **ResNet-101** presents a serious case of overfitting and model divergence, characterized by conflicting trends in training and validation performance of ResNet-101. The training loss steadily decreased to almost zero by epoch 30, while the validation loss skyrocketed to 1 × 10^7^, indicating a critical failure to generalize beyond the training data. This divergence started subtly at epoch 5 and worsened significantly after epoch 10, suggesting issues with optimization (such as a learning rate that is too high or insufficient regularization).

On the other hand, training accuracy approached perfect levels (around 1.0), but validation accuracy plummeted from 0.7 to 0.4 by epoch 30. The contrasting relationship between these metrics was clear: as training loss decreased (theoretically improving the fit), validation loss increased, and validation accuracy dropped to below-random levels. This contradiction confirmed that the model was memorizing noise and outliers in the training set instead of learning generalizable patterns. The widening gap between training and validation accuracy after epoch 5 (exceeding 0.6 by epoch 30) further highlighted the issue of overfitting. The simultaneous surge in validation loss and drop in accuracy pointed to significant flaws in the model design or compatibility with the data.

In Figure 10, the MLFNet model exhibited exceptional performance, achieving a near-perfect accuracy of 99.6% (504/506 correct predictions). Crucially, it demonstrated 100% recall/sensitivity (238/238 true fractures detected), meaning no actual fractures were missed—a critical achievement for medical diagnostics where FNs carry high risks. Precision is 99.2% (238/240), indicating only 2 FPs (healthy cases misclassified as fractured). The specificity is 99.3% (266/268), confirming strong identification of non-fractured cases. The F1-score harmonized precision and recall at 99.6%, reflecting an optimal balance. The absence of FNs suggests the model prioritizes patient safety by erring toward over-caution. While the two false positives might warrant minor tuning (e.g., adjusting decision thresholds), this performance is clinically outstanding overall.

DenseNet-169 shows exceptional diagnostic performance in classifying fractures, achieving nearly perfect differentiation between fractured and non-fractured cases. The model achieved an accuracy of 95.1% (481/506 correct predictions), indicating strong but imperfect performance. It demonstrated a recall (sensitivity) of 94.5% (225/238 actual fractures detected), meaning 13 fractures were missed—a notable concern for clinical safety. Precision stood at 94.9% (225/237 predicted fractures correct), with 12 FPs (healthy cases flagged as fractured). Specificity was 95.5% (256/268 non-fractures correctly identified), reflecting reliable detection of healthy cases. The F1-score balanced precision and recall at 94.7%. While the model handled non-fractured cases effectively, the 13 false negatives indicated potential risks in under-diagnosis, suggesting further tuning or data refinement was needed to improve sensitivity for critical medical applications.

EfficientNet-B3 shows exceptional diagnostic performance in classifying fractures, achieving almost perfect separation between fractured and non-fractured cases. The model achieved a high recall of 99.2% (236/238 actual fractures correctly identified), missing only two true fractures—a critical strength for medical safety. However, it exhibited lower precision of 88.7% (236/266 predicted fractures correct), generating 30 FPs where healthy cases were erroneously flagged as fractured. Specificity was 88.8% (238/268 non-fractures accurately recognized), indicating moderate effectiveness in confirming healthy cases. Overall accuracy reached 93.7% (474/506 correct), while the F1-score balanced these metrics at 93.7%. The model prioritized fracture detection (minimizing false negatives) at the cost of higher false alarms, suggesting it leaned toward caution in clinical trade-offs. Further optimization to reduce false positives was warranted without compromising sensitivity.

Inception-V3 shows effective diagnostic performance with a balanced outcome for detecting fractures. The model achieved an accuracy of 94.3% (477/506 correct predictions), reflecting competent but inconsistent performance. It demonstrated moderate sensitivity (recall) of 92.9% (221/238 actual fractures detected), missing 17 true fractures—a clinically significant shortfall where under-diagnosis posed risks. Conversely, precision was strong at 94.8% (221/233 predicted fractures correct), with only 12 FPs, indicating effective avoidance of unnecessary interventions for healthy cases. Specificity reached 95.5% (256/268 non-fractures identified), surpassing its recall performance. The F1-score balanced these metrics at 93.8%, highlighting a trade-off favoring precision over recall. While the model excelled in confirming non-fractured cases and minimizing false alarms, its missed fractures suggested limitations in reliability for safety-critical applications, warranting further refinement to improve sensitivity.

MobileNet-V2 presents classification performance for a fracture detection model, indicating significant challenges with class imbalance. The model delivered a strong overall performance with an accuracy of 97.0% (491/506 correct), the highest among recent comparative models. It achieved excellent recall (sensitivity) of 97.1% (231/238 actual fractures detected), missing only 7 true fractures—a significant improvement over architectures like InceptionV3 (17 FNs) and DenseNet169 (13 FNs). Precision remained high at 96.6% (231/239 predicted fractures correct), with merely 8 FPs, indicating minimal over-diagnosis. Specificity reached 97.0% (260/268 non-fractures correctly identified), demonstrating balanced capability across both classes. The F1-score settled at 96.8%, reflecting optimal harmony between sensitivity and precision. While not matching HybridSFNet’s perfection (0 FN/FP), MobileNetV2 prioritized clinical safety through low false negatives while maintaining operational efficiency, suggesting it struck a practical balance for real-world deployment.

The ResNet-101 model achieved a high recall of 97.9% (233/238 actual fractures detected), missing only 5 true fractures, which prioritized diagnostic safety by minimizing FNs. However, it exhibited notably low precision of 86.6% (233/269 predicted fractures correct), generating 36 FPs—the highest among recent models (e.g., MobileNetV2: 8 FPs, HybridSFNet: 2 FPs). Specificity was 86.6% (232/268 non-fractures correctly identified), revealing significant challenges in reliably excluding non-fractured cases. Overall accuracy reached 91.9% (465/506 correct), while the F1-score balanced recall and precision at 91.8%. The model leaned heavily toward sensitivity over specificity, resulting in substantial over-referrals (false alarms) that could burden clinical workflows. While its fracture detection capability was robust, the trade-off warranted calibration to reduce false positives for practical deployment.

Figure 11 displays the Receiver Operating Characteristic (ROC) curves for the six DL models. The ROC curve for **MLFNet** showed nearly perfect classification performance, closely hugging the top-left corner of the plot and achieving an area under the curve (AUC) score of 1.00. This indicates that MLFNet effectively distinguished between fractured and non-fractured X-ray images without compromising sensitivity or specificity. The curve’s steep ascent and maximal AUC highlight both high sensitivity and specificity, demonstrating the robustness of the proposed architecture.

**DenseNet-169** also performed well, achieving an AUC of 0.95. Its ROC curve rose sharply toward the top-left corner but fell slightly short of MLFNet’s ideal path. This performance suggests a strong ability to differentiate between the two classes, although some misclassifications could occur compared to MLFNet’s perfect separation.

**EfficientNet-B3** matched DenseNet-169 with an AUC of 0.95, following a similar trajectory in its ROC curve, which featured a steep rise initially and a gradual approach toward the upper right. While its performance was commendable, it did not achieve the flawless separation seen with MLFNet.

**Inception-V3** outperformed both DenseNet-169 and EfficientNet-B3 with an AUC of 0.96, indicating a slightly better balance between sensitivity and specificity. This suggests it managed borderline cases a bit more effectively, but still did not match MLFNet’s perfect discrimination.

**MobileNet-V2** achieved an AUC of 0.97, surpassing DenseNet-169, EfficientNet-B3, and Inception-V3. Its ROC curve indicated near-perfect classification capability, particularly excelling in minimizing false positives. Nevertheless, it still showed minor deviations from the ideal trajectory compared to MLFNet.

**ResNet-101** demonstrated the weakest performance among the six models, with an AUC of 0.92. Although its ROC curve steadily rose toward the top-left, it lagged in the early part of the curve, suggesting more false positives at lower thresholds. This indicates that ResNet-101 had comparatively less discriminative power in this classification task.

Overall, MLFNet clearly outperformed the other models, achieving a perfect AUC of 1.00 and an ideal ROC curve shape. MobileNet-V2 followed closely with an AUC of 0.97, while Inception-V3 slightly surpassed both DenseNet-169 and EfficientNet-B3, which both had an AUC of 0.95. ResNet-101 trailed with an AUC of 0.92, indicating the least effective discrimination between classes. Ultimately, MLFNet demonstrated superior discriminative ability, confirming the effectiveness of its tailored architecture for fracture detection.

Figure 12 presents the analysis of precision–recall curves for the six DL models. **MLFNet** showed excellent performance, with a precision–recall curve area of 0.99 for the ‘Fractured’ class and a perfect score of 1.00 for the Non_Fractured class. Both curves remained high across the recall range, indicating consistently high precision even as recall increased. This suggests that MLFNet was highly effective at accurately identifying both fractured and non-fractured cases with minimal false positives.

**DenseNet-169** also performed strongly, achieving areas of 0.92 for the ‘Fractured’ class and 0.93 for the Non_Fractured class. The precision remained high for both classes across most of the recall range, with only a slight drop at higher recall values. This indicates that DenseNet-169 was generally robust in its predictions, maintaining good precision while identifying a significant number of relevant instances.

**EfficientNet-B3** demonstrated good performance, with areas of 0.88 for the ‘Fractured’ class and 0.94 for the Non_Fractured class. The curve for the Non_Fractured class was notably higher and more stable than that for ‘Fractured’, which showed a more pronounced decline in precision at higher recall levels. This indicates that while EfficientNet-B3 performed well overall, it was more adept at accurately identifying non-fractured cases.

**Inception-V3** achieved solid performance, with areas of 0.91 for the ‘Fractured’ class and 0.92 for the Non_Fractured class. Both curves maintained high precision across a substantial portion of the recall range, indicating a reliable ability to classify instances correctly. Similarly to DenseNet-169, there was a noticeable dip in precision at very high recall, suggesting a trade-off in performance when attempting to capture nearly all relevant instances.

**MobileNet-V2** exhibited strong and consistent performance, with areas of 0.95 for the ‘Fractured’ class and 0.96 for the Non_Fractured class. Both precision–recall curves remained high and stable across the entire recall range, indicating that MobileNet-V2 maintained excellent precision even at high recall levels for both classes. This suggests that it was highly effective and balanced in its classification capabilities.

**ResNet-101** showed acceptable performance, with areas of 0.86 for the ‘Fractured’ class and 0.92 for the Non_Fractured class. The curve for the Fractured class exhibited a more significant drop in precision at higher recall values compared to the other models, indicating a greater challenge in maintaining precision for this class. However, the ‘Non_Fractured’ class performed considerably better, maintaining higher precision across the recall range.

When comparing the precision–recall curves across all six models, several key observations emerged. MLFNet and MobileNet-V2 consistently demonstrated the strongest performance. MLFNet achieved near-perfect scores, particularly for the Non_Fractured class (area = 1.00), while maintaining exceptionally high precision for Fractured cases (area = 0.99). MobileNet-V2 also showed outstanding and balanced performance, with areas of 0.95 and 0.96 for the Fractured and Non_Fractured classes, respectively.

DenseNet-169 and Inception-V3 exhibited strong, comparable performance, with area scores generally in the low 0.90s for both classes. These models maintained good precision for a significant portion of the recall range but demonstrated a more noticeable drop at very high recall values, suggesting a slight trade-off between precision and recall at extreme ends.

EfficientNet-B3 performed well, especially for the Non_Fractured class (area = 0.94), but its performance for the Fractured class (area = 0.88) was slightly lower and less stable compared to the top performers.

ResNet-101 generally showed the lowest performance among the models, particularly for the Fractured class (area = 0.86), where its precision dropped more significantly to higher recall levels. While its performance for the Non_Fractured class (area = 0.92) was respectable, the disparity between the two classes was more pronounced in ResNet-101 than in the other models.

Overall, MLFNet and MobileNet-V2 stood out for their superior and consistent precision–recall characteristics across both classes, making them the most robust choices for this multi-class classification task.

### 4.4. The Hybrid MLFNet Performance Assessment

In the second experiment, we incorporated the MLFNet model into hybrid ensemble architectures featuring DenseNet-169, EfficientNet-B3, Inception-V3, MobileNet-V2, and ResNet-101. The results of the second experiment are presented in Table 9 and Table 10. The MLFNet + DenseNet-169 achieved an accuracy of 98.81%, while MLFNet + EfficientNet-B3 reached 98.02%. MLFNet + Inception-V3 recorded an accuracy of 97.83%, MLFNet + MobileNet-V2 attained 97.04%, and MLFNet + ResNet-101 achieved 96.64% on the testing set of the BFMRX dataset. Among the models assessed, MLFNet + DenseNet-169 exhibited the highest accuracy.

In Table 9, the evaluation results show that the MLFNet + DenseNet-169 model performed well and consistently when tested on the BFMRX dataset. The average accuracy was 98.81%, demonstrating the model’s strong ability to correctly classify both fractured and non-fractured cases. The average specificity and recall were 98.76%, indicating that the model effectively identified TNs and TPs. The average FNR was low at 1.24%, while both the NPV and precision were 98.87%, further confirming the model’s reliability in making predictions.

When analyzed by class, the model classified fractured cases with an accuracy of 98.81%, a specificity of 99.63%, and a recall of 97.90%, meaning that only a small number of actual fractured cases were misclassified. The FNR for fractured samples was 2.10%, with both precision and F1-score being high at 99.57% and 98.73%, respectively, indicating a strong balance between sensitivity and precision. For non-fractured cases, the accuracy remained steady at 98.81%, with a slightly lower specificity of 97.90%, but a very low FNR of 0.37%, showing that few fractured cases were incorrectly labeled as non-fractured. The NPV was 99.57%, precision was 98.16%, recall was 99.63%, and F1-score was 98.89%, all of which highlight the model’s strong predictive ability for normal cases. Overall, the model maintained balanced and robust performance across both classes, with particularly high precision and recall values that are essential in medical diagnostic applications.

The evaluation results of the MLFNet + EfficientNet-B3 model showed that it performed strongly and consistently when tested on the dataset. The average accuracy was 98.02%, highlighting the model’s effectiveness in accurately identifying both fractured and non-fractured cases. The average specificity and recall were both 97.99%, indicating the model’s proficiency in detecting TNs and TPs, respectively. Furthermore, the average FNR was low at 2.01%, while the NPV and precision averaged 98.04%, suggesting that the model made highly reliable predictions. The average F1-score was 98.02%, indicating a balanced trade-off between precision and recall.

For the fractured class, the model achieved an accuracy of 98.02%, with a specificity of 98.51% and a recall of 97.48%, reflecting a low misclassification rate for actual fractured cases. The FNR was 2.52%, and the NPV, precision, and F1-score were 97.78%, 98.31%, and 97.89%, respectively, demonstrating strong performance with minimal false negatives. In the non-fractured class, the model also reached 98.02% accuracy, with a slightly lower specificity of 97.48% but a higher recall of 98.51% and a lower FNR of 1.49%. The NPV was 98.31%, the precision was 97.78%, and the F1-score was 98.14%, all indicating excellent performance in identifying normal cases. Overall, the model displayed high and consistent performance across both classes, with minimal disparities, making it well-suited for practical diagnostic applications.

The evaluation results for the MLFNet + Inception-V3 model demonstrated strong and balanced performance across both classes. The average accuracy was 97.83%, indicating the model’s effectiveness in correctly classifying both fractured and non-fractured cases. The average specificity and recall were both 97.85%, highlighting the model’s strong ability to identify true negatives and true positives. Additionally, the FNR averaged 2.15%, which remained low, while both the NPV and precision stood at 97.79%, suggesting highly reliable predictions. The average F1-score of 97.82% confirmed a well-balanced trade-off between precision and recall.

In the fractured class, the model achieved 97.83% accuracy, with 97.39% specificity and 98.32% recall, indicating it correctly detected most fractured cases with few false positives. The FNR was low at 1.68%, and the precision, NPV, and F1-score were 97.10%, 98.49%, and 97.70%, respectively, reflecting consistent and accurate classification. For the non-fractured class, the model also achieved 97.83% accuracy, with 98.32% specificity, 97.39% recall, and a slightly higher FNR of 2.61%. The precision, NPV, and F1-score were 98.49%, 97.10%, and 97.94%, respectively, demonstrating excellent performance in detecting normal cases. Overall, the model exhibited highly reliable and balanced classification capabilities, making it suitable for real-world medical diagnostics.

The evaluation results for the MLFNet + MobileNet-V2 model showed strong performance in classifying both fractured and non-fractured cases. The average accuracy was 97.04%, indicating a high level of correct predictions across the dataset. MLFNet + MobileNet-V2 achieved an average specificity and recall of 97.08%, demonstrating its effectiveness in identifying true negatives and true positives. The average FNR was low at 2.92%, while both the NPV and precision stood at 96.99%, suggesting reliable predictions. The average F1-score of 97.03% reflected a good balance between precision and recall.

For the fractured class, the model achieved 97.04% accuracy, with a specificity of 96.27% and a recall of 97.90%, indicating effective identification of fractured cases with minimal false negatives. The FNR was 2.10%, and the NPV, precision, and F1-score were 98.10%, 95.88%, and 96.88%, respectively, showing solid performance despite slightly lower precision. In the case of non-fractured samples, the model maintained the same accuracy of 97.04%, with a higher specificity of 97.90% and a lower recall of 96.27%. The FNR was slightly higher at 3.73%, while the NPV, precision, and F1-score were 95.88%, 98.10%, and 97.18%, respectively. Overall, the model demonstrated consistent performance across both classes, with a minor trade-off between precision and recall, making it a reliable choice for medical image classification tasks.

The evaluation results for the MLFNet + ResNet-101 model showed strong performance in classifying fractured and non-fractured cases. The average accuracy was 96.64%, indicating that the model successfully classified most samples. Both the average specificity and recall reached 96.66%, demonstrating the model’s effectiveness in identifying true negatives and true positives. The average FNR was low at 3.34%, while the NPV and precision were both 96.60%, indicating reliable predictive performance. The average F1-score was 96.63%, reflecting a good balance between precision and recall.

In the fractured class, the model achieved an accuracy of 96.64%, with a specificity of 96.27% and a recall of 97.06%, highlighting its strong ability to detect actual fracture cases. The FNR was 2.94%, while the NPV, precision, and F1-score were 97.36%, 95.85%, and 96.45%, respectively, showing slightly lower precision but high recall. For the non-fractured class, the model maintained the same accuracy of 96.64%, with higher specificity at 97.06% and slightly lower recall at 96.27%. The FNR increased slightly to 3.73%, while the NPV, precision, and F1-score were 95.85%, 97.36%, and 96.81%, respectively. Overall, the model provided consistent and reliable results across both classes, with only minor differences in class-specific metrics.

Figure 13 shows the training and validation loss (on the left) and training and validation accuracy (on the right) over 30 epochs for the five ensemble models. In the case of the **MLFNet + DenseNet-169** model, the analysis indicates that the model converged quickly, with both loss and accuracy stabilizing after about the fifth epoch.

In the loss curve, the training loss steadily decreased from approximately 0.35 to nearly 0.01, demonstrating effective learning from the training data. The validation loss also dropped quickly during the initial epochs and then varied slightly between 0.03 and 0.07, but generally remained low and close to the training loss. This suggests that the model did not experience overfitting and generalized well to new data.

Regarding accuracy, the model showed significant improvement in the first five epochs, with training accuracy increasing from around 85% to over 98%, eventually nearing 100%. Similarly, the validation accuracy rose quickly and stabilized around 98–99%, with minor fluctuations. The small gap between training and validation accuracy further indicates strong generalization and a minimal risk of overfitting.

Overall, the comparison of the accuracy and loss plots reveals a consistent and stable learning process, where the model achieved high accuracy with minimal loss for both training and validation sets, showcasing its robustness and reliability for fracture classification tasks.

For the ensemble **MLFNet + EfficientNet-B3**, the training loss (red) began at approximately 0.55 and consistently decreased, dropping below 0.05 by the end of training. The validation loss (green) also declined sharply during the early epochs but showed slight fluctuations between 0.1 and 0.2 after epoch 10, indicating some variability in the model’s generalization ability.

In contrast, the training accuracy rose quickly from around 70% to nearly 99%, demonstrating effective learning from the training data. The validation accuracy also improved rapidly, increasing from about 78% to roughly 96%, and remained relatively stable with minor fluctuations throughout the training process. The gap between training and validation accuracy was small, particularly after epoch 10, suggesting that the model did not experience significant overfitting.

In summary, the comparison of the loss and accuracy plots showed that while the model learned effectively (as evidenced by the steady decrease in training loss and increase in training accuracy), the slightly fluctuating validation loss indicated some variability in generalization. Nevertheless, the consistently high validation accuracy confirmed that the model maintained strong predictive performance on unseen data.

The **MLFNet + Inception-V3** exhibited highly effective learning dynamics, with well-synchronized loss reduction and accuracy improvement. Both training and validation loss curves exhibited smooth exponential decay, decreasing from approximately 0.4 to near 0.02 over 30 epochs, indicating stable convergence without significant oscillations. The validation loss closely tracked the training loss throughout, maintaining a narrow gap of <0.01 after epoch 15, which demonstrated excellent generalization capability with minimal overfitting.

Concurrently, accuracy metrics showed complementary improvement: training accuracy rose steadily from 80% to 94%, while validation accuracy progressed from 85% to 90%. The validation accuracy plateaued after epoch 20, with only 0.5% fluctuation, indicating model stability. The inverse relationship between loss and accuracy was particularly evident at epoch 10, where loss decreased by 55% (from 0.4 to 0.18) and accuracy increased by 12.5% (from 80% to 90%). This coordinated progression confirmed efficient feature extraction and weight optimization. The terminal metrics at epoch 30 showed near-ideal alignment: training loss (0.02) ≈ validation loss (0.03), and training accuracy (94%) > validation accuracy (90%).

The persistent 4–5% accuracy gap between training and validation in later epochs suggested slight overfitting, though the minimal loss gap (<0.01) confirmed it was well-managed. The validation accuracy stabilized at 90% after epoch 20, while training accuracy continued improving to 94%, reflecting appropriate complexity balancing. These patterns collectively indicated successful model optimization, where loss reduction directly translated to accuracy gains, with validation metrics providing reliable performance estimates for real-world deployment.

The **MLFNet + MobileNet-V2** model exhibited effective learning dynamics characterized by a strong inverse correlation between loss reduction and accuracy improvement. Training loss decreased steadily from 0.25 to near 0.00 over 30 epochs, while validation loss followed a parallel trajectory but plateaued at 0.05 after epoch 15, indicating early convergence. Concurrently, training accuracy rose from 88% to 98%, showing continuous improvement throughout the training. Validation accuracy increased more moderately from 90% to 94%, plateauing after epoch 20 with minimal fluctuation.

The inverse relationship was particularly pronounced between epochs 5 and 15, where loss decreased by 80% (from 0.20 to 0.04) and accuracy increased by 6% (from 90% to 96%). While training metrics showed near-perfect optimization (0.00 loss, 98% accuracy), the validation metrics demonstrated excellent generalization: terminal validation loss (0.05) remained 5× higher than training loss (0.00), and validation accuracy (94%) was 4% lower than training accuracy (98%).

The growing divergence after epoch 15—where training loss continued decreasing while validation loss stabilized—suggested mild overfitting. However, the validation accuracy maintained a stable plateau at 94% with only ±0.5% variation in the final 10 epochs. This indicated robust feature extraction despite the overfitting tendency, with the 4% accuracy gap between training and validation representing an acceptable trade-off for generalization capability. The coordinated progression confirmed that loss reduction directly translated to accuracy gains throughout the training process.

The **MLFNet + ResNet-101** exhibited consistent improvement in both loss reduction and accuracy enhancement over 30 epochs, though there were emerging signs of overfitting in the later stages. Training loss decreased steadily from 0.5 to 0.1, following a near-linear trajectory that demonstrated effective optimization. Validation loss initially mirrored this trend but plateaued at 0.15 after epoch 20, revealing early convergence and a growing generalization gap. Concurrently, training accuracy showed robust improvement from 75% to 98%, while validation accuracy increased more moderately from 75% to 92%, plateauing after epoch 25 with only ±0.5% fluctuation.

The inverse relationship between loss and accuracy was particularly pronounced between epochs 5 and 15: loss decreased by 60% (from 0.4 to 0.16), and accuracy increased by 17% (from 78% to 95%). This strong correlation confirmed that weight updates effectively translated to performance gains. However, diverging trends emerged in the later epochs. After epoch 20, training loss continued to decrease to 0.1, validation loss stalled at 0.15, training accuracy reached 98%, and validation accuracy plateaued at 92%.

The terminal metrics revealed a 7% accuracy gap and a 0.05 loss gap between training and validation, indicating mild overfitting. Despite this, the validation accuracy stabilized at 92% with minimal variance in the final five epochs, confirming reliable generalization. The coordinated early-phase progression demonstrated efficient feature learning, while the later-phase divergence suggested that model complexity could be reduced for better regularization. Overall, the validation metrics (92% accuracy, 0.15 loss) represented clinically viable performance for diagnostic deployment.

In Figure 14, the **MLFNet + DenseNet-169** model achieved exceptional performance, attaining an accuracy of 98.8% (500/506 correct predictions). It demonstrated high recall (sensitivity) of 97.9% (233/238 actual fractures detected), missing only 5 true fractures—a clinically robust outcome for safety-critical applications. Precision was nearly perfect at 99.6% (233/234 predicted fractures correct), with just one FP, minimizing unnecessary interventions. Specificity reached 99.6% (267/268 non-fractures correctly identified), highlighting outstanding reliability in confirming healthy cases. The F1-score balanced these metrics at 98.7%. While the 5 FNs fell short of the HybridSFNet’s flawless recall (0 FN), the single false positive represented a significant improvement over models like EfficientNetB3 (30 FPs) and ResNet101 (36 FPs). This performance positioned the model as a top-tier solution, effectively balancing diagnostic safety and operational efficiency for fracture detection.

The **MLFNet + EfficientNet-B3** model demonstrated high-performance characteristics with an accuracy of 98.0% (496/506 correct predictions). It achieved a recall of 97.5% (232/238 actual fractures detected), missing 6 true fractures—a marginal increase in FNs. Precision remained exceptional at 98.3% (232/236 predicted fractures correct), with only 4 FPs, reflecting minimal over-diagnosis. Specificity reached 98.5% (264/268 non-fractures correctly identified), nearing perfection in healthy case identification. The F1-score balanced these metrics at 97.9%. While the 6 FNs represented a slight safety gap relative to the optimized version, the 4 false positives still outperformed most peer models (e.g., EfficientNetB3: 30 FPs, ResNet101: 36 FPs). This pre-optimization state already delivered clinically viable results but indicated that threshold refinement could further reduce missed fractures.

The **MLFNet + Inception-V3** model achieved outstanding sensitivity (recall) of 98.3% (234/238 actual fractures detected), missing only 4 true fractures—a significant improvement over the standalone InceptionV3 model (17 FNs). Precision remained strong at 97.1% (234/241 predicted fractures correct), though 7 FPs indicated moderate over-referrals of healthy cases. Specificity reached 97.4% (261/268 non-fractures correctly identified), demonstrating robust performance in confirming non-injured cases. Overall accuracy stood at 97.8% (495/506 correct), while the F1-score balanced recall and precision at 97.7%. The MLFNet enhancement substantially elevated InceptionV3’s diagnostic safety by reducing FNs by 76% (from 17 to 4 FNs), though slight FP inflation persisted compared to top performers like HybridMLFNet (1 FP). This represented a clinically viable balance, prioritizing fracture detection while maintaining operational efficiency.

The **MLFNet + MobileNet-V2** model achieved a recall of 97.9% (233/238 actual fractures detected), reducing FNs to 5—a slight improvement over standalone MobileNetV2 (7 FNs) and matching ResNet101’s sensitivity. However, it exhibited a lower precision of 95.9% (233/243 predicted fractures correct) due to 10 false positives, doubling its predecessor’s FP count (standalone MobileNetV2: 5 FPs). Specificity declined to 96.3% (258/268 non-fractures identified) compared to the standalone version’s 97.0%. Overall accuracy remained 97.0% (491/506 correct), identical to the base MobileNetV2, while the F1-score settled at 96.9%. The enhancement prioritized further reduction in missed fractures but introduced more false alarms, suggesting that while diagnostic safety strengthened, operational efficiency moderately decreased relative to the original architecture. This represented a viable clinical trade-off where sensitivity gains outweighed precision costs for critical applications.

The **MLFNet + ResNet-101** enhanced ResNet101 model achieved a recall of 97.1% (231/238 actual fractures detected), missing 7 true fractures—a slight decline from the base ResNet101’s 97.9% recall (5 FNs). Precision improved significantly to 95.9% (231/241 predicted fractures correct), reducing FPs to 10 from the base model’s 36 FPs. Specificity remained strong at 96.3% (258/268 non-fractures identified). Overall accuracy reached 96.6% (489/506 correct), surpassing the base ResNet101’s 91.9%, while the F1-score rose to 96.5% from 91.8%. The enhancement successfully mitigated the base model’s critical weakness of excessive false positives, cutting FP errors by 72% without substantially compromising sensitivity. This represented a meaningful clinical optimization, balancing fracture detection reliability (low FN) with operational efficiency (reduced false alarms) compared to the original architecture.

Table 11 provides a summary of the architectural setups and output dimensions of various MLFNet + CNN-based hybrid models utilized in the study. Each model, including DenseNet-169, EfficientNet-B3, Inception-V3, MobileNet-V2, and ResNet-101, adhered to a common processing workflow but varied in their backbone feature extractors and the resulting output sizes. All the models commenced with an input layer sized at (1, 128, 128, 3). Feature extraction was performed using pretrained backbones, each yielding different output shapes: DenseNet-169 produced (1, 4, 4, 1664), EfficientNet-B3 generated (1, 4, 4, 1536), Inception-V3 resulted in (1, 2, 2, 2048), MobileNet-V2 gave (1, 4, 4, 1280), and ResNet-101 returned (1, 4, 4, 2048).

These features were then processed through a series of SFNet blocks, along with batch normalization and dropout layers, which gradually reduced the spatial dimensions while enhancing the depth of features. The outputs were subsequently flattened and concatenated, leading to slightly varied fusion shapes across the models: (1, 132,736) for DenseNet-169, (1, 132,608) for EfficientNet-B3, (1, 133,120) for both Inception-V3 and ResNet-101, and (1, 132,352) for MobileNet-V2. Ultimately, each model included fully connected layers with Dense, Dropout, and a final output layer shaped at (1, 1) for binary classification. Overall, the table illustrated the impact of pretrained backbones on the dimensionality of intermediate representations while maintaining a consistent overall architecture.

### 4.5. External Validation

We implemented two experiments to validate the proposed MLFNet model externally. In the first experiment, we used the BF dataset. This dataset contains about 9463 image samples of fractured and non-fractured bones. It was divided into a testing set (600 images) and a training set (8863 images). External validation tests whether the MLFNet model can perform well on unseen, independent data that may differ in imaging protocols, patient demographics, or equipment. It ensures that the MLFNet model is not overfitted to the training dataset, and an MLFNet model that performs well on external data is more likely to generalize to real-world clinical settings.

In the first experiment of the external validation process, the MLFNe model was evaluated using the BF dataset for binary classification (fracture vs. non-fracture). MLFNet achieved an impressive accuracy of 99.50% as a standalone model and 99.25% when integrated into hybrid ensemble architectures with the same five pre-trained DL models. The results of the standalone and hybrid MLFNet are shown in Table 12 and Table 13, respectively.

Table 12 shows that MLFNet excelled, achieving an average accuracy of 99.50%, with perfect recall (100%) for fractured cases and specificity (100%) for non-fractured ones. It had zero FNs for fractured cases and only 1% FNR for non-fractured cases, indicating an exceptionally balanced and robust model. MLFNet consistently outperformed all other models in every metric, followed closely by DenseNet-169, Inception-V3, and MobileNet-V2, while EfficientNet-B3 and ResNet-101 showed slightly lower but still acceptable performance. These findings confirm MLFNet’s superior generalization, sensitivity, and specificity, making it the most reliable model in the tested cohort.

The MLFNet model is more complex than the other five CNNs. The average accuracy for DenseNet-169 is 99%. The average accuracy achieved with the proposed model is 99.5%. The advantages and disadvantages of increasing the complexity of the system for a 0.5% improvement are discussed below:

Advantages:Improved generalization—Even small gains in accuracy can be critical in medical imaging and safety-critical domains, where misclassification has high consequences.Better feature representation—The multi-level fusion mechanism in MLFNet allows for richer feature aggregation across scales, potentially enhancing robustness to dataset variability.Consistent external performance—The 0.5% improvement is sustained in external validation, indicating the model’s enhanced ability to generalize beyond the training distribution.

Disadvantages:Increased computational cost—The proposed model requires higher memory usage and longer inference times compared to standard CNNs, which may limit deployment on low-resource devices.Longer training duration—The deeper and more interconnected architecture increases optimization complexity, potentially requiring more extensive hyperparameter tuning.

Figure 15 shows the training and validation loss and accuracy of the MLFNet model over 30 epochs. At the start, the training loss was high, around 2.00, but it quickly dropped during the first few epochs. The validation loss also decreased significantly, beginning at about 0.50. As training continued from epochs 5 to 30, both training and validation losses kept declining, though at a slower rate, stabilizing between 0.00 and 0.25. Importantly, the validation loss stayed at or just below the training loss during this time, indicating that the model was generalizing well and not overfitting based on the loss metric.

In the early epochs, training accuracy began low at around 0.60 but increased rapidly. Similarly, validation accuracy rose sharply from an initial value of about 0.80. By epoch 10, both accuracies approached 0.95. For the remainder of the training (epochs 10–30), both training and validation accuracies remained high, fluctuating between 0.95 and 1.00. The validation accuracy often exceeded or closely matched the training accuracy, further supporting the conclusion that the model was effectively generalizing and avoiding overfitting.

Overall, the MLFNet model displayed strong learning capabilities, as shown by the quick reduction in loss and increase in accuracy during the initial epochs. The consistent performance of the validation metrics compared to the training metrics suggested that the model was generalizing well to unseen data and not overfitting. Both the loss and accuracy curves stabilized at favorable levels, indicating that the model had converged and achieved robust performance by the end of the training period.

Table 13 shows that all the hybrid ensemble models achieved high classification accuracy, though their performance varied across individual metrics. MLFNet + DenseNet-169 hybrid achieved the highest average accuracy of 99.25%, reflecting its exceptional classification capability. It demonstrated highly balanced performance across both classes, with a recall and specificity of 99.25%, and a very low average FNR of 0.75%, indicating excellent sensitivity and precision. MLFNet + DenseNet-169 emerged as the best-performing model, followed closely by MLFNet + Inception-V3, while MLFNet + ResNet-101 ranked lowest among the tested hybrid configurations. Nevertheless, all the models demonstrated robust performance, confirming the advantages of combining MLFNet with established backbone architectures to enhance fracture classification accuracy.

In the second experiment of the external validation process, we evaluated the MLFNet model by applying the stratified 5-fold cross-validation scheme. In the stratified 5-fold cross-validation scheme, the dataset was divided into five equal partitions, with each fold taking a turn as the testing set while the other four folds were used for model training. This rotation continued until every fold had been used once for testing. Finally, the outcomes from all the folds were averaged to obtain a stable and unbiased estimate of the model’s performance.

The MLFNet model was evaluated on the BFMRX dataset for binary classification (fracture vs. non-fracture). MLFNet achieved an impressive accuracy of 99.52% as a standalone model. The results of the standalone MLFnet are shown in Table 14.

Table 14 shows that the MLFNet model’s performance was outstanding. It achieved an average accuracy of 99.52%, meaning it correctly classified the vast majority of instances across all test folds. This high level of accuracy was consistent with other metrics, as shown by the average F1-score and AUC also being 99.52%. This indicated a robust and highly effective model with virtually no drop in performance between the training and validation phases.

The results were remarkably consistent across all five folds, with minimal fluctuation. The accuracy ranged from a high of 99.89% in Fold 5 to a low of 99.14% in Fold 1, a very narrow range of just 0.75%. This consistency suggested that the model was not overly dependent on a particular subset of the data and generalized well to unseen data from the same distribution.

Figure 16 illustrates the training and validation loss across multiple folds during the model training process. It showed that for most folds, the training and validation losses rapidly decreased within the first few epochs and then stabilized near zero, indicating effective learning and convergence. However, Fold 3 exhibited a significant spike in training loss, reaching a peak of approximately 25 around epoch 4, before sharply dropping to near zero. Similarly, the validation loss for Fold 1 showed a smaller spike, reaching around 6, but also quickly recovered. These spikes indicated that the model experienced temporary instability or potential numerical fluctuations during early training stages in some folds. Overall, after the initial few epochs, both training and validation losses across all the folds remained consistently low, suggesting that the model eventually achieved stable and well-generalized performance.

Figure 17 illustrates the training and validation accuracy across multiple folds during the model training process. It showed that for most folds, accuracy increased rapidly within the first few epochs and quickly stabilized near 100%, indicating effective learning. In the early epochs, some folds experienced temporary fluctuations. For example, Train Fold 1 started with lower accuracy, around 78%, but quickly improved and converged near 100%. Similarly, Val Fold 2 showed a noticeable drop to around 85% before recovering. Train Fold 3 also exhibited a brief decline to about 90% around epoch 3, but it rapidly stabilized thereafter. After approximately the first five epochs, both training and validation accuracies across all the folds consistently remained above 99%, suggesting that the model achieved excellent generalization and stable performance across the cross-validation folds.

### 4.6. Ablation Study

An ablation study is an essential experimental approach in DL research, especially for medical image analysis tasks such as detecting BFs, where the reliability and interpretability of models are critical. This method entails the deliberate removal or alteration (ablating) of specific components within a model or the input data process to assess their individual impact on overall performance. Here are some key reasons why ablation studies are significant in this area [37].

In the detection of BFs, DL models typically consist of various elements, including preprocessing layers, attention mechanisms, feature fusion blocks, and ensemble architectures. Ablation studies are useful for identifying which components play a crucial role in performance and which ones may be unnecessary. For instance, eliminating an attention mechanism could lead to a decrease in recall, highlighting its significance in concentrating on areas of fracture.

In this research, we performed two experiments. In the first experiment, we conducted an ablation analysis on the testing sets of the BFMRX and BF datasets by ablating four parameters: standard CNN, dropout, filters, and dense layers. The results indicating the performance metrics evaluation of MLFNet on the testing sets of the BFMRX and BF datasets, obtained by ablating the mentioned parameters, are shown in Table 15 and Table 16 and illustrated in Figure 18 and Figure 19, respectively.

Table 15 and Figure 18 present that the standard CNN model demonstrated strong performance, recording a test accuracy of 99.01% and validation accuracy of 97.35%, with moderate losses of 3.21% for test and 7.69% for validation, indicating a well-balanced architecture.

The MLFNet model without dropout reached test accuracy of 98.02% and validation accuracy of 97.35%, but displayed a higher validation loss of 12.24%, suggesting that the absence of dropout may have led to overfitting and diminished generalization.

Conversely, utilizing large filters (5 × 5) experienced a notable decline in performance, achieving only a test accuracy of 70.75% and a validation accuracy of 78.29%, alongside very high test and validation losses of 70.59% and 51.34%, respectively. This indicates that the larger filter sizes resulted in oversmoothing, hindering the model’s ability to learn detailed fracture features.

The simplified dense layers model exhibited the lowest performance, with just 52.96% test accuracy and 59.35% validation accuracy, along with high loss values around 69%, highlighting severe underfitting due to inadequate capacity in the classification layers.

In contrast, the proposed standalone MLFNet model, without these architectural variations, achieved the highest test accuracy of 99.60%, surpassing the standard CNN baseline by 0.59 percentage points.

Table 16 and Figure 19 display that the standard CNN attained a validation accuracy of 100%, although its test accuracy was slightly lower at 99.50%. Notably, the standard CNN exhibited the lowest test loss (1.24%), indicating strong generalization, despite having the fewest parameters (17,181,953) among the top models.

MLFNet model without dropout achieved a respectable test accuracy of 97.49% and a validation accuracy of 99.01%, yet its test loss (8.26%) and validation loss (2.67%) were significantly higher than those of the top three models. This indicated potential overfitting, as dropout regularization typically helps enhance generalization.

Conversely, the model with large filters (5 × 5) performed poorly, with a test accuracy of just 68.67% and a validation accuracy of 85.89%. Additionally, its test loss was exceptionally high at 148.74%, and validation loss at 43.06%, suggesting that the larger filters did not effectively capture the intricate fracture features necessary for accurate classification.

Lastly, the simplified dense layers model demonstrated the weakest overall performance, with a test accuracy of 50.38% and a validation accuracy of only 42.08%. Its loss values were high on both the test (69.19%) and validation (69.33%) sets, indicating severe underfitting due to reduced model capacity.

In the second experiment, we performed an ablation analysis on the testing set of the BFMRX dataset by removing the contrast enhancement (without Contrast-Limited Adaptive Histogram Equalization (CLAHE)). The results, which show the performance metrics of MLFNet on these testing sets after modifying the parameters, are presented in Table 17.

The comparison clearly shows that removing the contrast enhancement step leads to a decrease in classification performance (accuracy reduced from 99.60% to 99.21%), confirming that this preprocessing step contributes positively to feature extraction and model generalization.

### 4.7. Confidence Interval

Confidence intervals (CIs) play a vital role in assessing the effectiveness of DL models for identifying bone fractures. They measure the uncertainty surrounding point estimates of important metrics such as sensitivity, specificity, or area under the curve (AUC). A narrower CI reflects greater precision in the model’s predicted performance on new, unseen data, indicating enhanced reliability. For example, if a DL model demonstrates 95% sensitivity with a 95% CI of [93%, 97%], it provides stronger evidence of effective fracture detection compared to another model with the same sensitivity but a broader CI of [88%, 99%]. Including CIs when reporting performance metrics is crucial for comprehending the statistical significance and clinical relevance of deep learning systems for fracture detection, aiding in the evaluation of their potential for practical use in real-world scenarios [38]. Table 18 shows the CIs for accuracy of the six DL techniques evaluated on the test set of the BFMRX dataset.

Table 18 and Figure 20 display the 95% CIs for the effectiveness of six DL models assessed on the EBTC test set. MLFNet achieved the highest accuracy, with a CI of [99.60474, 99.60474], reflecting a very consistent performance. Following closely was MobileNet-V2, which had a CI of [97.03557, 97.03557], indicating strong and stable results as well. DenseNet-169 and Inception-V3 demonstrated moderate performance, with CIs of [95.05929, 95.05929] and [94.26877, 94.26877], respectively. EfficientNet-B3 showed slightly lower accuracy, recording a CI of [93.67589, 93.67589]. ResNet-101 exhibited the lowest performance, with a CI of [91.89723, 91.89723]. The narrow nature of all CIs (single-point values) suggests that there was no variability in the model outputs across the test set for the metrics used in the evaluation.

### 4.8. p-Value

A *p*-value measures the likelihood of obtaining results that are at least as extreme as those observed in a statistical test, assuming that the null hypothesis is true (which posits no real difference between the models being compared) [39]. In evaluating model performance, the null hypothesis usually asserts that there is no significant difference in performance metrics (such as accuracy or AUC) between two models.

In DL, especially in medical imaging, performance metrics like accuracy, sensitivity, and AUC can be affected by factors such as dataset sampling, model initialization, or random training processes. Even if one model shows better performance, the difference may simply result from random fluctuations rather than a genuine improvement.

By performing a statistical significance test (like an independent two-sample *t*-test, McNemar’s test, or bootstrapping) and reporting the *p*-value, researchers can assess the likelihood that any observed performance difference is due to chance. A low *p*-value (typically < 0.05) indicates statistical evidence that the improvement is unlikely to be random, thereby reinforcing the credibility of claims regarding the superiority of a proposed DL model.

In this study, analyzing the *p*-value, as shown in Table 19 and Figure 21, was crucial to confirming that MLFNet’s higher accuracy compared to baseline architectures was not due to random chance but indicated a statistically significant performance enhancement. To evaluate whether the differences in accuracy were statistically significant, we performed independent two-sample *t*-tests comparing MLFNet with each baseline model. The results showed *p*-values of 9.23 × 10^−43^ for DenseNet-169, 3.52 × 10^−48^ for EfficientNet-B3, 9.12 × 10^−47^ for Inception-V3, 6.42 × 10^−28^ for MobileNet-V2, and 5.99 × 10^−55^ for ResNet-101. All of these values were well below the 0.05 significance threshold, indicating that MLFNet’s improvement in accuracy compared to the baseline models was statistically significant.

### 4.9. Outcome Analysis in Relation to Existing Literature

BFs are typical injuries that frequently occur due to falls, collisions, and other traumatic incidents [1,3]. The global rate of bone fractures is on the rise, largely due to aging populations and increasingly urban lifestyles [3]. Identifying fractures through radiographic imaging is a routine procedure for patients experiencing both high-energy and low-energy trauma in various clinical settings, including emergency rooms, urgent care centers, and outpatient practices like orthopedics, rheumatology, and family medicine [4,5]. Missed fractures in radiographic images contribute significantly to diagnostic discrepancies between initial assessments by non-radiologists or radiology residents and final evaluations conducted by board-certified radiologists, leading to unnecessary harm or delays in patient treatment. Misinterpretations of fractures can account for up to 24% of harmful diagnostic errors noted in emergency departments [4]. Manually reviewing radiological images for fractures requires substantial time and effort. A fatigued radiologist might miss a fracture in an otherwise normal image.

To address this issue, this research presented a novel CNN model named MLFNet, designed to effectively capture and integrate both low-level and high-level image features. MLFNet utilizes parallel convolutional branches to extract diverse semantic representations and incorporates Grad-CAM for improved interpretability by highlighting fracture regions. Moreover, ablation studies were performed to evaluate different architectural variations, confirming the model’s robustness and generalizability across various data distributions.

We conducted two experiments to assess the classification efficacy of the MLFNet model using the BFMRX dataset. The primary goal of the two experiments was to identify BFs to improve patient outcomes, simplify the diagnostic process, and significantly reduce both patient time and costs. In these experiments, 87% of the BFMRX dataset, which consists of 9246 images, was used for training. The remaining 5%, or 506 images, was reserved for testing, while 8%, or 828 images, was designated for validation. The BFMRX dataset includes two categories: Fractured and Non-fractured. Both experiments focused on transfer learning to pre-train six DL models: MLFNet, DenseNet-169, EfficientNet-B3, Inception-V3, MobileNet-V2, and ResNet-101.

In the initial experiment, we used the MLFNet model on its own and evaluated its performance with the testing subset of the BFMRX dataset. The average accuracy for each model was as follows: MLFNet achieved 99.60%, DenseNet-169 reached 95.06%, EfficientNet-B3 obtained 93.68%, Inception-V3 recorded 94.27%, MobileNet-V2 reached 97.04%, and ResNet-101 logged 91.90%. These results show that MLFNet had the highest accuracy among all the models tested.

In the second experiment, we integrated the MLFNet model into hybrid ensemble architectures that included DenseNet-169, EfficientNet-B3, Inception-V3, MobileNet-V2, and ResNet-101. The combination of MLFNet and DenseNet-169 achieved an accuracy of 98.81%, while the MLFNet and EfficientNet-B3 combination reached 98.02%. The MLFNet with Inception-V3 recorded an accuracy of 97.83%, MLFNet with MobileNet-V2 attained 97.04%, and MLFNet with ResNet-101 achieved 98.49% on the testing set of the BFMRX dataset. Among all the models evaluated, MLFNet with DenseNet-169 showed the highest accuracy.

It is important to recognize that the proposed system was designed to assist healthcare professionals, not to replace their clinical judgment. All negative predictions made by the system should be reviewed through clinical evaluation, particularly in cases involving high-risk patients or unclear findings. Defining decision thresholds based on individual patient history, presenting symptoms, and relevant risk factors can help determine when further diagnostic testing or specialist referrals are needed for borderline cases. Additionally, continuously retraining the model with new data—especially from previously misclassified FNs—can help minimize future errors and improve accuracy over time.

As presented in Table 20 and Figure 22, while previous state-of-the-art studies consistently achieved high classification performance on the BFMRX dataset, our MLFNet approach surpassed them all. Our results are compared to previous studies that used the same BFMRX dataset. However, differences in how the experiments were set up—such as data splits, preprocessing methods, and model adjustments—mean that these results should be seen as suggestive rather than conclusive performance comparisons.

Most prior studies showed strong performance, with accuracies typically between 96% and 99.20%. Among these, Alsufyani, A. [23] achieved one of the highest reported accuracies at 99.20% by utilizing a combination of CNN architectures, including AlexNet, DenseNet-121, ResNet-152, and EfficientNet-B3. Similarly, Uddin, A. et al. [28] reached 99% accuracy with a diverse set of models such as ViT, MobileViT, and YOLOv8.

Other methods, like VGG-16 used by Pillai, R. et al. [21] and ResNet-18 employed by Alex, R. and Rosmasari [24], achieved accuracies of 98.22% and 97.59%, respectively. In contrast, classical CNN-based methods, including those from Kumar, A. et al. [26] and Chauhan, S. [27], yielded lower accuracies around 96%, indicating a performance gap between traditional and advanced architectures.

In contrast, the proposed model utilized a hybrid architecture that integrates MLFNet with deep CNNs such as DenseNet-169, EfficientNet-B3, Inception-V3, MobileNet-V2, and ResNet-101. The standalone MLFNet model achieved a superior accuracy of 99.60%, while the combination of MLFNet + DenseNet-169 attained 98.81%, both outperforming all previously published methods on the BFMRX dataset. These results underscored the effectiveness of the proposed architecture, particularly MLFNet, in capturing complex fracture-related patterns with greater precision and reliability than existing models.

Thus, the proposed method not only exceeded the highest reported accuracies from previous studies but also demonstrated consistent and robust performance, validating its potential as a state-of-the-art solution for bone fracture detection.

### 4.10. Limitations and Future Work

The MLFNet model showed high accuracy and promising performance, but it has limitations. A key limitation of this study is the reduced generalizability of the proposed MLFNet model across different bone fracture datasets. While the model achieved high performance when trained and tested on the same dataset, its performance dropped significantly when directly tested on an unseen dataset. This discrepancy can be attributed to domain shift and data distribution differences, resolutions, labeling strategies, patient demographics, and acquisition devices. BFs exhibit significant variability in their patterns and characteristics depending on the location within the bone, the type of bone, and the mechanism of injury. Long bones such as the radius, humerus, and femur differ in their anatomical structure and bone density compared to spongy bones like the vertebrae and carpal bones. These anatomical differences directly influence the appearance and morphology of fractures in medical imaging. Furthermore, fractures can present in multiple forms, including transverse fractures, spiral fractures, comminuted fractures, and stress fractures. This diversity in fracture types, combined with variations in anatomical structures, acquisition devices, and labeling strategies across medical institutions, leads to substantial heterogeneity among datasets. Consequently, the MLFNet model faces challenges in generalizing unseen datasets. Differences in bone structures, fracture morphologies, and imaging characteristics cause a domain shift, which explains why the model’s performance drops when tested directly on a different dataset but remains high when trained and tested on the same dataset.

For future development, it is advisable to validate MLFNet on datasets from multiple institutions to ensure reliability across different clinical settings, investigate its use in real-time environments such as emergency departments, and improve its ability to classify various fracture types while integrating clinical decision support.

## 5. Conclusions

This research introduced a novel DL approach called MLFNet, designed specifically for detecting BFs in X-ray images. By using parallel convolutional branches and Grad-CAM for interpretability, MLFNet effectively combined both basic and advanced image features, improving diagnostic accuracy and usability in clinical settings. MLFNet was designed to support clinicians in the precise and timely detection of BFs, aiming to minimize diagnostic delays, reduce healthcare expenses, and enhance patient outcomes.

The model was tested on the BFMRX dataset for binary classification (fracture vs. non-fracture). The preprocessing steps involved normalizing images, adjusting their size, and enhancing contrast. On the testing set of the BFMRX dataset, MLFNet achieved an impressive accuracy of 99.60% as a standalone classifier and 98.81% when combined with DenseNet-169, surpassing several leading pre-trained models such as EfficientNet-B3, Inception-V3, MobileNet-V2, and ResNet-101.

The research was validated externally on the BF dataset and achieved an impressive accuracy of 99.50% as a standalone model and 99.25% when integrated into hybrid ensemble architectures with the same five pre-trained DL models. It employed a two-phase experimental framework that confirmed MLFNet’s reliability and adaptability across different architectures and different datasets. Additionally, ablation studies highlighted the importance of the model’s design elements and their roles in classification success.

Hence, MLFNet has proven to be an accurate, interpretable, and efficient tool for detecting BFs, making it a strong candidate for application in real-world clinical environments. Its use could shorten diagnostic times, reduce human errors, and improve patient care in radiology and orthopedic practices. Future research may focus on deploying the model in real-time clinical scenarios, validating it with multi-institutional datasets, and expanding its capabilities to classify multiple types of fractures.

## Figures and Tables

**Figure 3 diagnostics-15-02212-f003:**
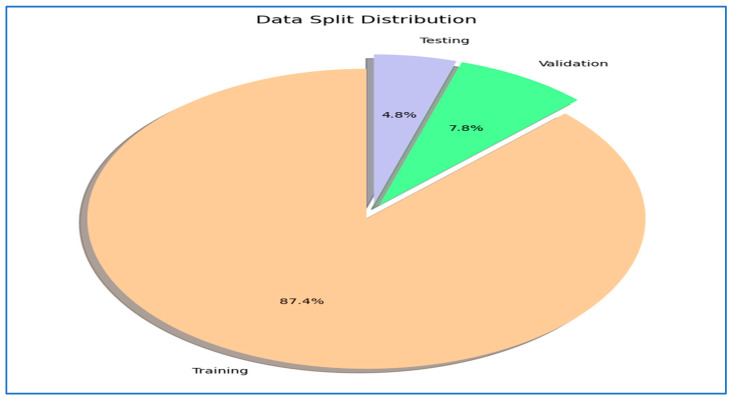
The BFMRX dataset split distribution.

**Figure 4 diagnostics-15-02212-f004:**
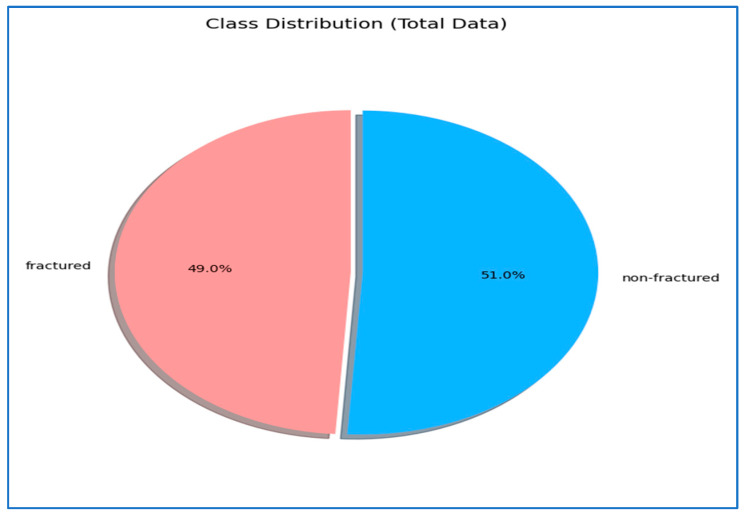
The class distribution of the BFMRX dataset.

**Figure 5 diagnostics-15-02212-f005:**
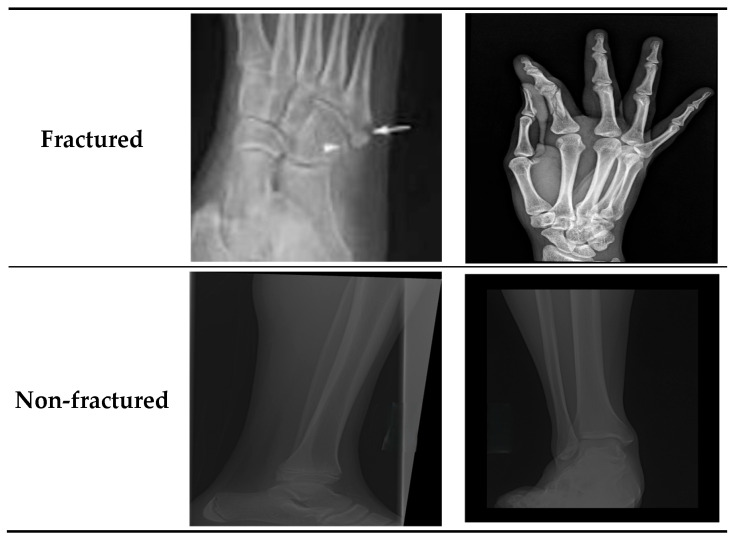
Sample images of fracture and non-fracture from the BFMRX dataset.

**Figure 7 diagnostics-15-02212-f007:**
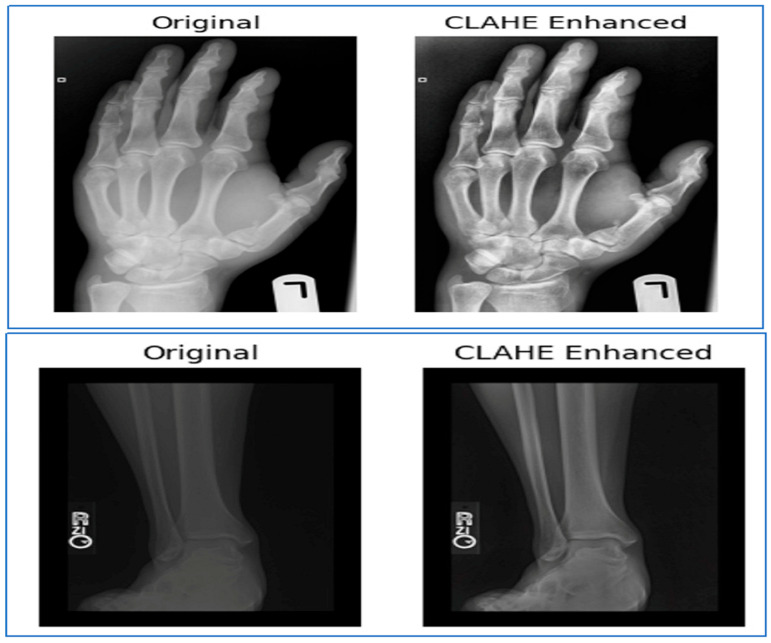
The images before and after contrast enhancement.

**Figure 8 diagnostics-15-02212-f008:**
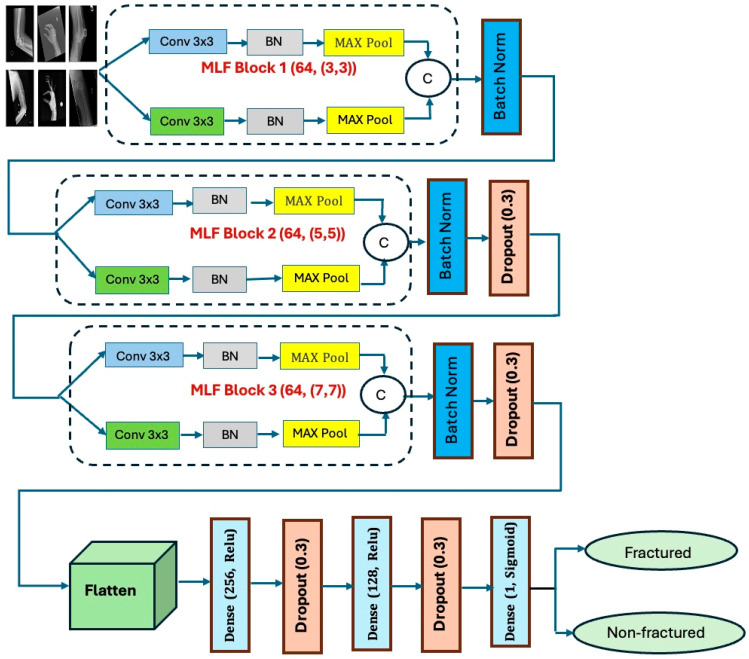
The proposed MLFNet’s architecture.

**Figure 9 diagnostics-15-02212-f009:**
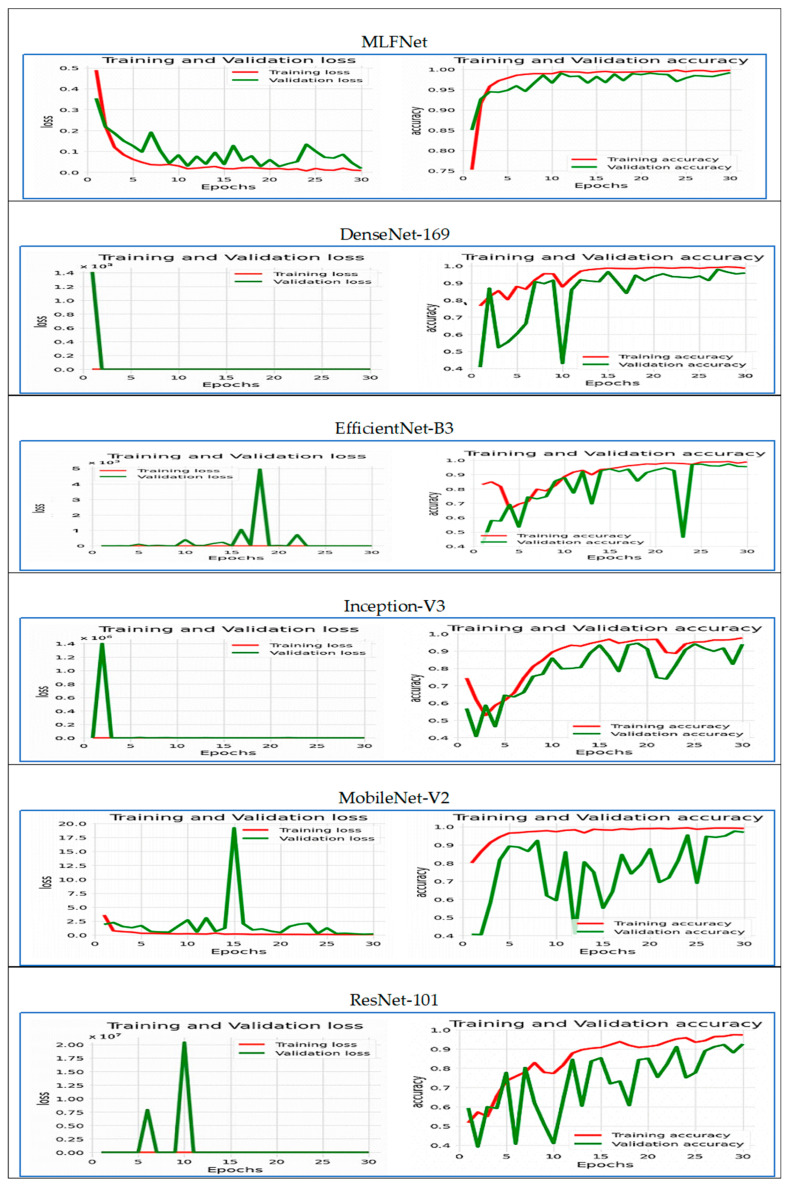
The training and validation loss and the training and validation accuracy of the six DL models.

**Figure 10 diagnostics-15-02212-f010:**
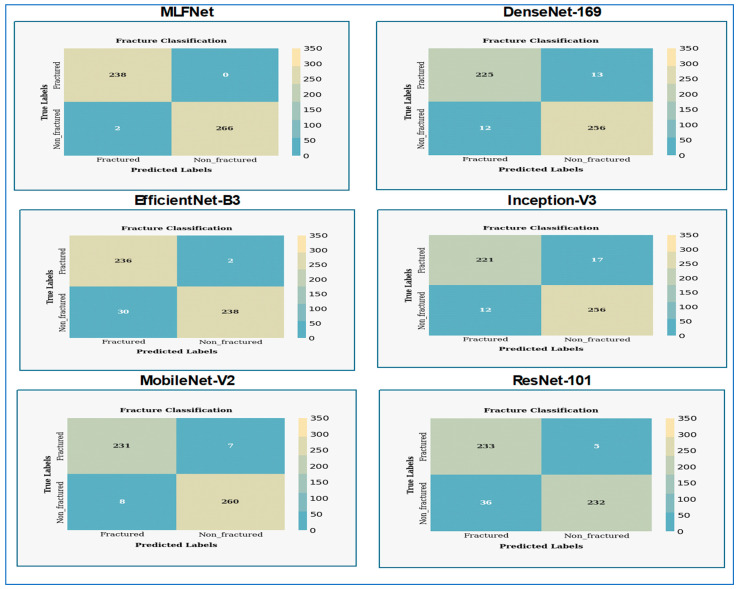
The confusion matrices of the six DL methods over the testing set of the BFMRX dataset.

**Figure 11 diagnostics-15-02212-f011:**
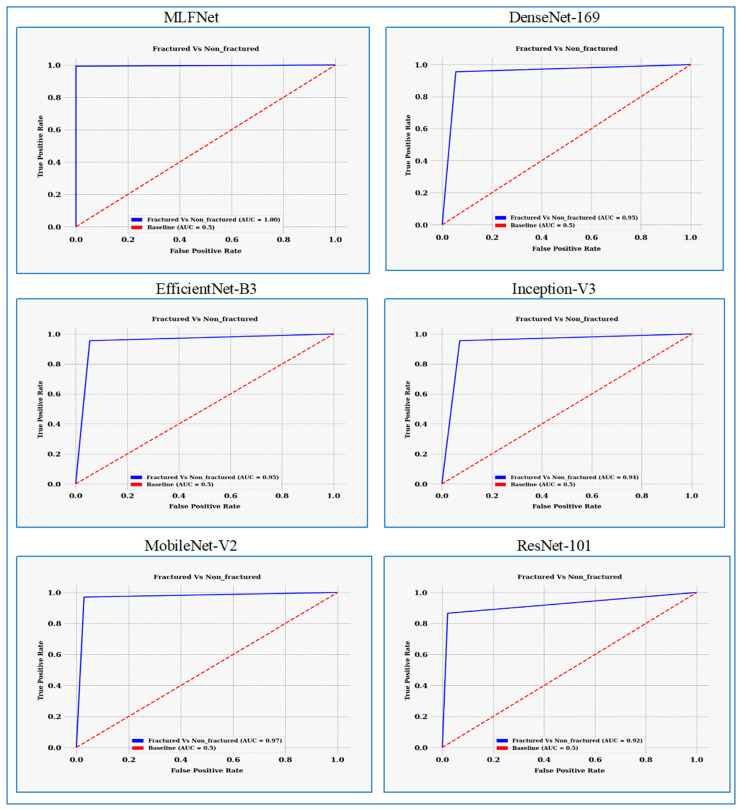
The ROC curves of the six DL methods over the testing set of the BFMRX dataset.

**Figure 12 diagnostics-15-02212-f012:**
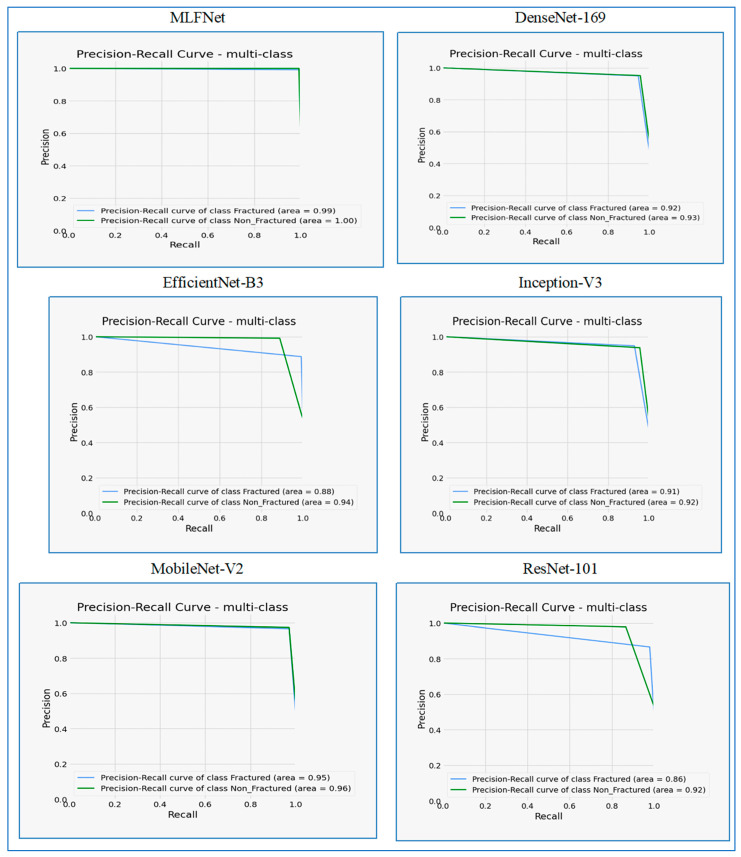
The precision–recall curves of the six DL methods over the testing set of the BFMRX dataset.

**Figure 13 diagnostics-15-02212-f013:**
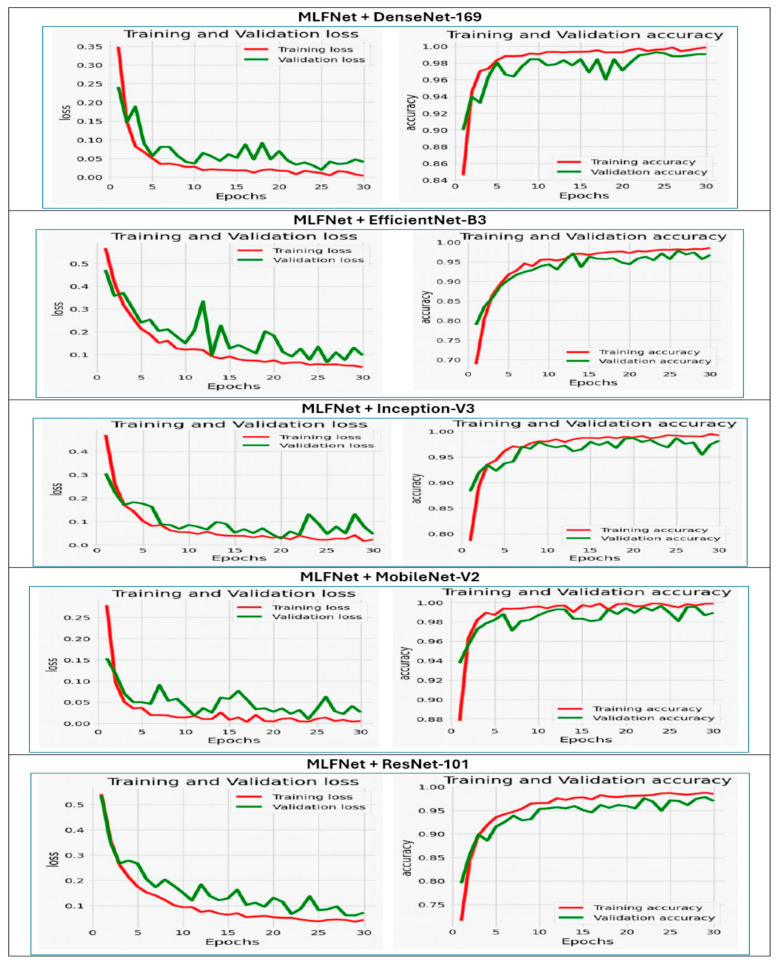
The training and validation loss and the training and validation accuracy of the five ensemble models.

**Figure 14 diagnostics-15-02212-f014:**
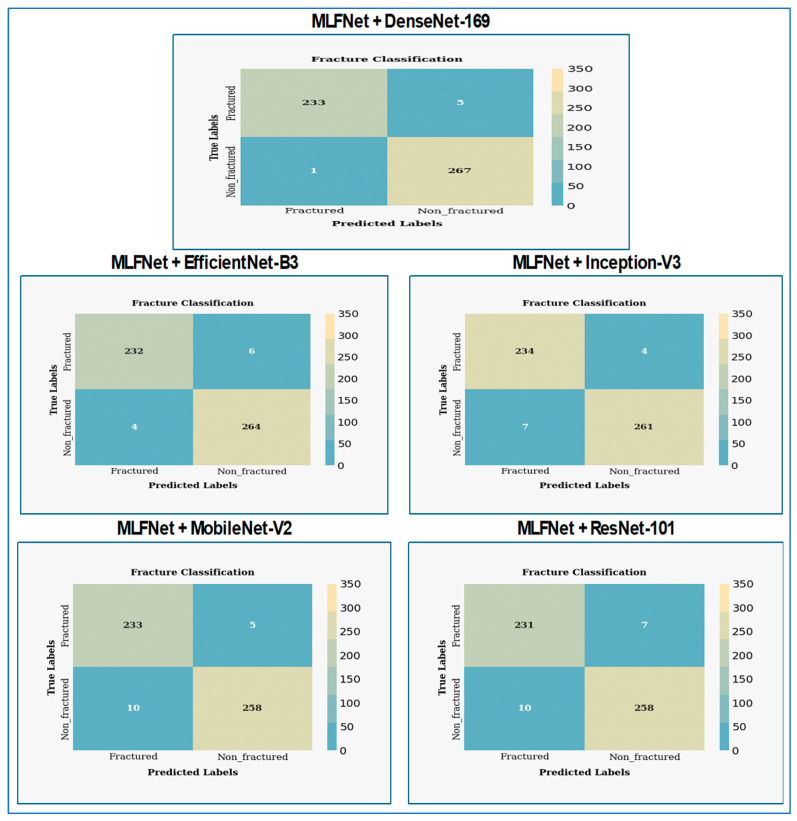
The confusion matrix of the five hybrid ensemble models over the testing set of the BFMRX dataset.

**Figure 15 diagnostics-15-02212-f015:**
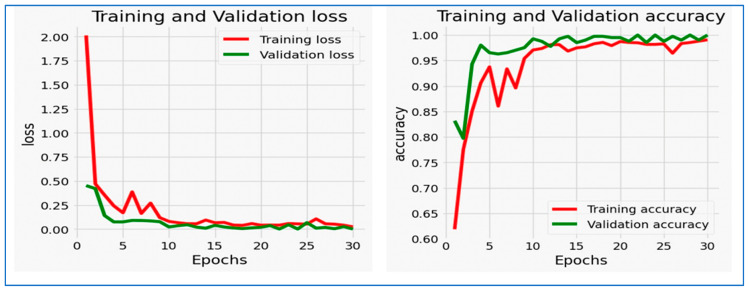
The training and validation loss and the training and validation accuracy of MLFNet on the testing set of the BF dataset.

**Figure 16 diagnostics-15-02212-f016:**
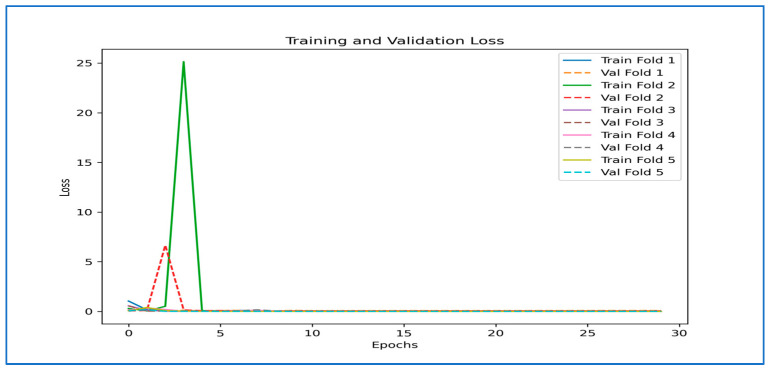
The training and validation loss of MLFNet using five-fold cross-validation on the testing set of the BFMRX dataset.

**Figure 17 diagnostics-15-02212-f017:**
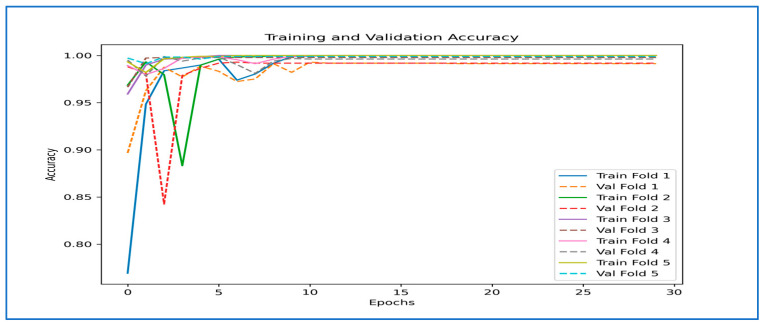
The training and validation accuracy of MLFNet using five-fold cross-validation on the testing set of the BFMRX dataset.

**Figure 18 diagnostics-15-02212-f018:**
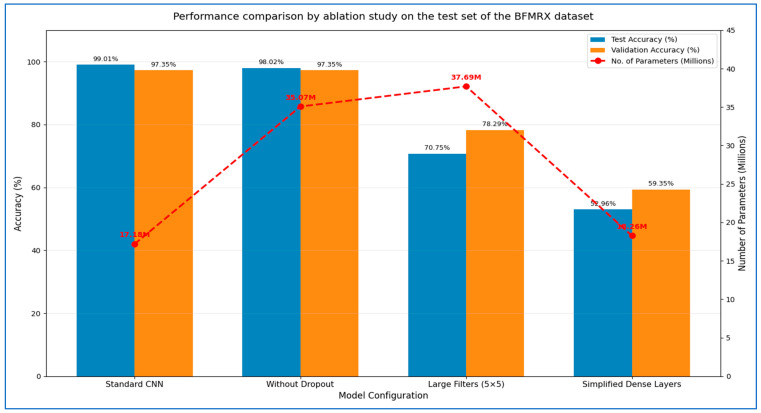
Performance comparison by ablation study on the test set of the BFMRX dataset.

**Figure 19 diagnostics-15-02212-f019:**
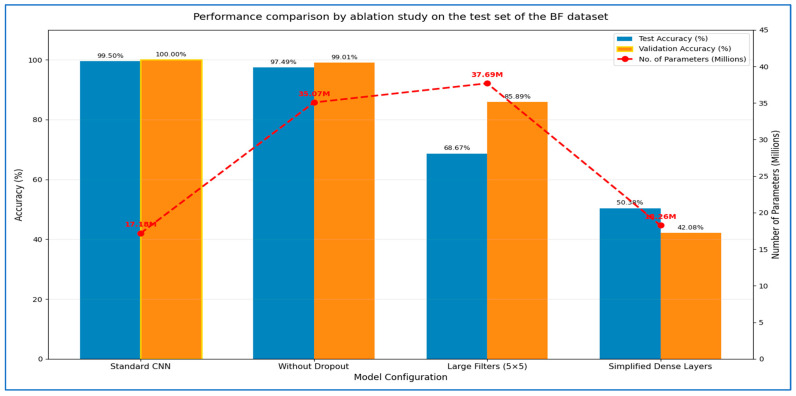
Performance comparison by ablation study on the test set of the BF dataset.

**Figure 20 diagnostics-15-02212-f020:**
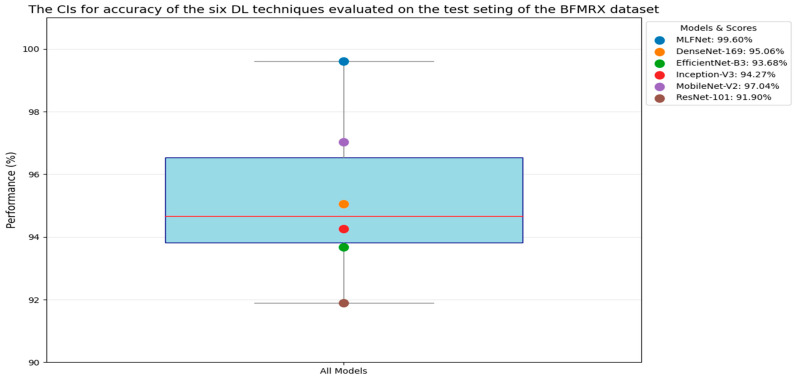
The CIs for accuracy of the six DL techniques evaluated on the testing set of the BFMRX dataset.

**Figure 21 diagnostics-15-02212-f021:**
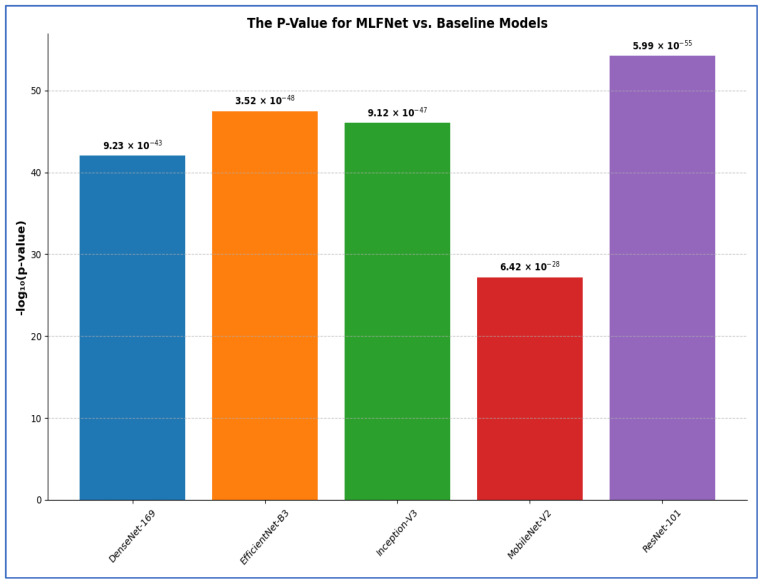
The *p*-value for accuracy of the six DL techniques evaluated on the testing set of the BFMRX dataset.

**Figure 22 diagnostics-15-02212-f022:**
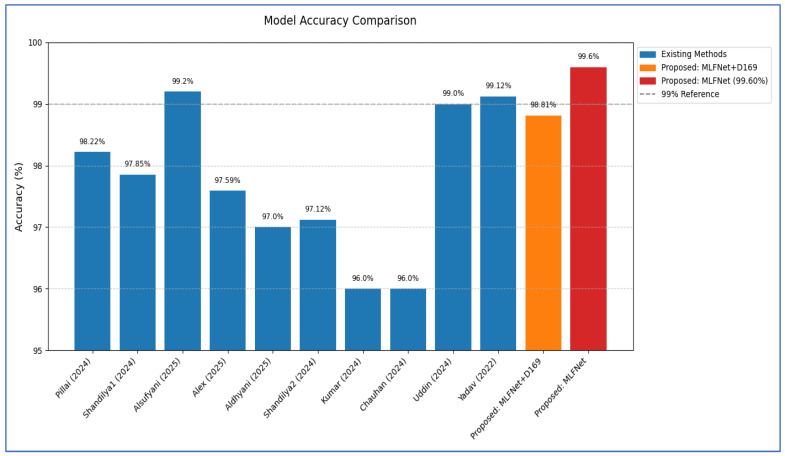
Comparison of results with cutting-edge approaches [9,21,22,23,24,26,27,28].

**Table 1 diagnostics-15-02212-t001:** X-ray image distribution.

Class	Training	Validation	Testing	Total
fractured	4606	336	238	5180
non-fractured	4640	492	268	5400
Total	9246	828	506	10,580

**Table 2 diagnostics-15-02212-t002:** Componentwise description of the proposed model.

Block Name	Layer(s)/Function	Purpose
Input Layer	128 × 128 × 3 X-ray Image	Provides raw input images to the model
Preprocessing	Normalization/Enhancement	Standardizes pixel values and improves image quality
SFLayer	Conv2D → BatchNorm → MaxPooling	Extracts local features with regularization
SFNet Block	2× SFLayer → Concatenation	Learns richer multi-path features
Block 1	SFNetBlock with 64 filters	Captures low-level edge features
Block 2	SFNetBlock with 128 filters	Captures mid-level semantic features
Block 3	SFNetBlock with 256 filters	Captures abstract, high-level features
Dropout Layers	Dropout (*p* = 0.3) after Block 2 and Block 3	Prevents overfitting by randomly deactivating neurons
Flatten	Converts 2D → 1D	Prepares data for dense layers
Dense Layers	Dense (256) → Dropout → Dense (128) → Dropout	High-level representation and decision-making
Output Layer	Dense (1), Activation: Sigmoid	Binary classification: fracture vs. no fracture

**Table 3 diagnostics-15-02212-t003:** A comparative architectural novelty between the MLFNet and the other models.

Feature/Component	DenseNet169	EfficientNet B3	InceptionV3	MobileNet-V2	ResNet-101	MLFNet
Input Resolution	128 × 128 × 3 (X-ray)	128 × 128 × 3 (X-ray)	128 × 128 × 3 (X-ray)	128 × 128 × 3 (X-ray)	128 × 128 × 3 (X-ray)	128 × 128 × 3 (X-ray)
Preprocessing	Standard normalization	Standard normalization	Standard normalization	Standard normalization	Standard normalization	Normalization + X-ray enhancement
Core Building Block	Dense block (dense connections)	MBConv + SE block (compound scaling)	Inception module (multi-branch conv)	Inverted residual blocks	Residual blocks	SFNet block (multi-path Conv2D + concatenation)
Low-Level Feature Extraction	Initial Conv + Pooling	Initial Conv + Pooling	Stem conv layers	Conv + BatchNorm	Conv + BatchNorm	SFNetBlock with 64 filters (edge features)
Mid-Level Feature Extraction	Dense blocks	MBConv	Inception modules	Depthwise-separable conv	Residual blocks	SFNetBlock with 128 filters (semantic features)
High-Level Feature Extraction	Deeper dense blocks	Deeper MBConv	Larger Inception filters	Bottleneck blocks	Deep residual stacks	SFNetBlock with 256 filters (abstract features)
Regularization	Dropout (optional)	Dropout + SE	Dropout	Dropout	Dropout	Dropout after mid and high-level blocks
Feature Fusion Strategy	Dense concatenation	Compound scaling	Multi-branch merge	Sequential bottlenecks	Residual skip connections	Multi-path feature concatenation across SFNet blocks
Classifier Head	Global Avg Pool + Dense	Global Avg Pool + Dense	Global Avg Pool + Dense	Global Avg Pool + Dense	Global Avg Pool + Dense	Flatten → Dense (256) → Dense (128) → Sigmoid
Domain Specialization	General-purpose	General-purpose	General-purpose	Mobile/edge	General-purpose	Optimized for X-ray fracture detection

**Table 4 diagnostics-15-02212-t004:** A comparison between the five additional CNN models.

Model	Input Shape	Total Params	Trainable Params	Non-Trainable Params	Number of Layers
DenseNet169	(128, 128, 3)	13,077,061	12,915,333	161,728	600
EfficientNetB3	(128, 128, 3)	11,184,436	11,094,061	90,375	390
InceptionV3	(128, 128, 3)	22,336,805	22,298,277	38,528	316
MobileNetV2	(128, 128, 3)	2,592,325	2,555,653	36,672	159
ResNet101V2	(128, 128, 3)	43,160,581	43,058,821	101,760	382

**Table 5 diagnostics-15-02212-t005:** The two experiments’ hyperparameters.

Parameter	Value
img_size	128 × 128 × 3
Number of epochs	30
channels	3
Optimizer	Adam
Initial learning rate	0.001
Patience	10
loss	binary_crossentropy
Activation	sigmoid

**Table 6 diagnostics-15-02212-t006:** The performance of the six CNN models using the testing subset of the BFMRX dataset.

Model	Accuracy (%)	Specificity (%)	FNR (%)	NPV (%)	Precision (%)	Recall (%)	F1-Score (%)
MLFNet	99.60	99.63	0.37	99.58	99.58	99.63	99.60
DenseNet-169	95.06	95.03	4.97	95.05	95.05	95.03	48.15
EfficientNet-B3	93.68	93.98	6.02	93.94	93.94	93.98	93.68
Inception-V3	94.27	94.19	5.81	94.31	94.31	94.19	94.24
MobileNet-V2	97.04	97.04	2.96	97.02	97.02	97.04	97.03
ResNet-101	91.90	92.23	7.77	92.25	92.25	92.23	91.90

**Table 7 diagnostics-15-02212-t007:** The classwise performance of the six DL models using testing subset of the BFMRX dataset.

	Class	Accuracy (%)	Specificity (%)	FNR (%)	NPV (%)	Precision (%)	Recall (%)	F1-Score (%)
MLFNet	Fractured	99.60	99.25	0.00	100.00	99.17	100.00	99.58
	Non-fractured	99.60	100.00	0.75	99.17	100.00	99.25	99.63
	Average	99.60	99.63	0.37	99.58	99.58	99.63	99.60
DenseNet-169	Fractured	95.06	95.52	5.46	95.17	94.94	94.54	0.95
	Non-fractured	95.06	94.54	4.48	94.94	95.17	95.52	95.34
	Average	95.06	95.03	4.97	95.05	95.05	95.03	48.15
EfficientNet-B3	Fractured	93.68	88.81	0.84	99.17	88.72	99.16	93.65
	Non-fractured	93.68	99.16	11.19	88.72	99.17	88.81	93.70
	Average	93.68	93.98	6.02	93.94	93.94	93.98	93.68
Inception-V3	Fractured	94.27	95.52	7.14	93.77	94.85	92.86	93.84
	Non-fractured	94.27	92.86	4.48	94.85	93.77	95.52	94.64
	Average	94.27	94.19	5.81	94.31	94.31	94.19	94.24
MobileNet-V2	Fractured	97.04	97.01	2.94	97.38	96.65	97.06	96.86
	Non-fractured	97.04	97.06	2.99	96.65	97.38	97.01	97.20
	Average	97.04	97.04	2.96	97.02	97.02	97.04	97.03
ResNet-101	Fractured	91.90	86.57	2.10	97.89	86.62	97.90	91.91
	Non-fractured	91.90	97.90	13.43	86.62	97.89	86.57	91.88
	Average	91.90	92.23	7.77	92.25	92.25	92.23	91.90

**Table 8 diagnostics-15-02212-t008:** The performance of the six DL models by other metrics on the testing subset of the BFMRX dataset.

Model	Train Time (s)	Inference Time (s)	Memory Usage (MB)	Params Count
MLFNet	81.44337928	12.0270946	10,310.46094	35,070,000
DenseNet-169	419.1784675	34.20856285	13,284.18164	13,076,033
EfficientNet-B3	304.9635962	23.30317223	15,891.57031	11,183,408
Inception-V3	161.1117506	19.53114045	17,414.4707	22,335,777
MobileNet-V2	114.5730445	14.87542236	18,448.73633	2,591,297
ResNet-101	291.2739118	28.60760403	20,855.83203	43,191,169

**Table 9 diagnostics-15-02212-t009:** The performance of the five ensemble models over the testing subset of the BFMRX dataset.

Hybrid Model	Accuracy (%)	Specificity (%)	FNR (%)	NPV (%)	Precision (%)	Recall (%)	F1-Score (%)
MLFNet + DenseNet-169	98.81	98.76	1.24	98.87	98.87	98.76	98.81
MLFNet + EfficientNet-B3	98.02	97.99	2.01	98.04	98.04	97.99	98.02
MLFNet + Inception-V3	97.83	97.85	2.15	97.79	97.79	97.85	97.82
MLFNet + MobileNet-V2	97.04	97.08	2.92	96.99	96.99	97.08	97.03
MLFNet + ResNet-101	96.64	96.66	3.34	96.60	96.60	96.66	96.63

**Table 10 diagnostics-15-02212-t010:** The classwise performance of the six DL models using the testing subset of the BFMRX dataset.

Hybrid Model	Class	Accuracy (%)	Specificity (%)	FNR (%)	NPV (%)	Precision (%)	Recall (%)	F1-Score (%)
MLFNet + DenseNet-169	Fractured	98.81	99.63	2.10	98.16	99.57	97.90	98.73
	Non-fractured	98.81	97.90	0.37	99.57	98.16	99.63	98.89
	Average	98.81	98.76	1.24	98.87	98.87	98.76	98.81
MLFNet + EfficientNet-B3	Fractured	98.02	98.51	2.52	97.78	98.31	97.48	97.89
	Non-fractured	98.02	97.48	1.49	98.31	97.78	98.51	98.14
	Average	98.02	97.99	2.01	98.04	98.04	97.99	98.02
MLFNet + Inception-V3	Fractured	97.83	97.39	1.68	98.49	97.10	98.32	97.70
	Non-fractured	97.83	98.32	2.61	97.10	98.49	97.39	97.94
	Average	97.83	97.85	2.15	97.79	97.79	97.85	97.82
MLFNet + MobileNet-V2	Fractured	97.04	96.27	2.10	98.10	95.88	97.90	96.88
	Non-fractured	97.04	97.90	3.73	95.88	98.10	96.27	97.18
	Average	97.04	97.08	2.92	96.99	96.99	97.08	97.03
MLFNet + ResNet-101	Fractured	96.64	96.27	2.94	97.36	95.85	97.06	96.45
	Non-fractured	96.64	97.06	3.73	95.85	97.36	96.27	96.81
	Average	96.64	96.66	3.34	96.60	96.60	96.66	96.63

**Table 11 diagnostics-15-02212-t011:** The dimensions of the features extracted with the MLFNet+ CNN models using the testing set of the BFMRX dataset.

MLFNet + CNN	Layer	Output Shape
DenseNet-169	Input Layer	(1, 128, 128, 3)
	Pretrained Backbone Output	(1, 4, 4, 1664)
	GlobalAveragePooling2D	(1, 1664)
	SFNet Block 1	(1, 64, 64, 128)
	BatchNorm 1	(1, 64, 64, 128)
	SFNet Block 2	(1, 32, 32, 256)
	BatchNorm 2	(1, 32, 32, 256)
	Dropout 1	(1, 32, 32, 256)
	SFNet Block 3	(1, 16, 16, 512)
	BatchNorm 3	(1, 16, 16, 512)
	Dropout 2	(1, 16, 16, 512)
	Flatten	(1, 131,072)
	Concatenate Fusion	(1, 132,736)
	Dense 1	(1, 256)
	Dropout 3	(1, 256)
	Dense 2	(1, 128)
	Dropout 4	(1, 128)
	Output Layer	(1, 1)
EfficientNet-B3	Input Layer	(1, 128, 128, 3)
	Pretrained Backbone Output	(1, 4, 4, 1536)
	GlobalAveragePooling2D	(1, 1536)
	SFNet Block 1	(1, 64, 64, 128)
	BatchNorm 1	(1, 64, 64, 128)
	SFNet Block 2	(1, 32, 32, 256)
	BatchNorm 2	(1, 32, 32, 256)
	Dropout 1	(1, 32, 32, 256)
	SFNet Block 3	(1, 16, 16, 512)
	BatchNorm 3	(1, 16, 16, 512)
	Dropout 2	(1, 16, 16, 512)
	Flatten	(1, 131,072)
	Concatenate Fusion	(1, 132,608)
	Dense 1	(1, 256)
	Dropout 3	(1, 256)
	Dense 2	(1, 128)
	Dropout 4	(1, 128)
	Output Layer	(1, 1)
Inception-V3	Input Layer	(1, 128, 128, 3)
	Pretrained Backbone Output	(1, 2, 2, 2048)
	GlobalAveragePooling2D	(1, 2048)
	SFNet Block 1	(1, 64, 64, 128)
	BatchNorm 1	(1, 64, 64, 128)
	SFNet Block 2	(1, 32, 32, 256)
	BatchNorm 2	(1, 32, 32, 256)
	Dropout 1	(1, 32, 32, 256)
	SFNet Block 3	(1, 16, 16, 512)
	BatchNorm 3	(1, 16, 16, 512)
	Dropout 2	(1, 16, 16, 512)
	Flatten	(1, 131,072)
	Concatenate Fusion	(1, 133,120)
	Dense 1	(1, 256)
	Dropout 3	(1, 256)
	Dense 2	(1, 128)
	Dropout 4	(1, 128)
	Output Layer	(1, 1)
MobileNet-V2	Input Layer	(1, 128, 128, 3)
	Pretrained Backbone Output	(1, 4, 4, 1280)
	GlobalAveragePooling2D	(1, 1280)
	SFNet Block 1	(1, 64, 64, 128)
	BatchNorm 1	(1, 64, 64, 128)
	SFNet Block 2	(1, 32, 32, 256)
	BatchNorm 2	(1, 32, 32, 256)
	Dropout 1	(1, 32, 32, 256)
	SFNet Block 3	(1, 16, 16, 512)
	BatchNorm 3	(1, 16, 16, 512)
	Dropout 2	(1, 16, 16, 512)
	Flatten	(1, 131,072)
	Concatenate Fusion	(1, 132,352)
	Dense 1	(1, 256)
	Dropout 3	(1, 256)
	Dense 2	(1, 128)
	Dropout 4	(1, 128)
	Output Layer	(1, 1)
ResNet-101	Input Layer	(1, 128, 128, 3)
	Pretrained Backbone Output	(1, 4, 4, 2048)
	GlobalAveragePooling2D	(1, 2048)
	SFNet Block 1	(1, 64, 64, 128)
	BatchNorm 1	(1, 64, 64, 128)
	SFNet Block 2	(1, 32, 32, 256)
	BatchNorm 2	(1, 32, 32, 256)
	Dropout 1	(1, 32, 32, 256)
	SFNet Block 3	(1, 16, 16, 512)
	BatchNorm 3	(1, 16, 16, 512)
	Dropout 2	(1, 16, 16, 512)
	Flatten	(1, 131,072)
	Concatenate Fusion	(1, 133,120)
	Dense 1	(1, 256)
	Dropout 3	(1, 256)
	Dense 2	(1, 128)
	Dropout 4	(1, 128)
	Output Layer	(1, 1)

**Table 12 diagnostics-15-02212-t012:** The performance of the six CNN models as standalones on the testing set of the BF dataset.

	Class	Accuracy (%)	Specificity (%)	FNR (%)	NPV (%)	Precision (%)	Recall (%)	F1-Score (%)
MLFNet	Fractured	99.50	99.00	0.00	100.00	99.00	100.00	99.50
	Non-fractured	99.50	100.00	1.00	99.00	100.00	99.00	99.50
	Average	99.50	99.50	0.50	99.50	99.50	99.50	99.50
DenseNet-169	Fractured	99.00	98.00	0.00	100.00	98.03	100.00	99.00
	Non-fractured	99.00	100.00	2.00	98.03	100.00	98.00	98.99
	Average	99.00	99.00	1.00	99.01	99.01	99.00	99.00
EfficientNet-B3	Fractured	96.99	97.50	3.52	96.53	97.46	96.48	96.97
	Non-fractured	96.99	96.48	2.50	97.46	96.53	97.50	97.01
	Average	96.99	96.99	3.01	97.00	97.00	96.99	96.99
Inception-V3	Fractured	97.74	96.00	0.50	99.48	96.12	99.50	97.78
	Non-fractured	97.74	99.50	4.00	96.12	99.48	96.00	97.71
	Average	97.74	97.75	2.25	97.80	97.80	97.75	97.74
MobileNet-V2	Fractured	97.74	98.50	3.02	97.04	98.47	96.98	97.72
	Non-fractured	97.74	96.98	1.50	98.47	97.04	98.50	97.77
	Average	97.74	97.74	2.26	97.76	97.76	97.74	97.74
ResNet-101	Fractured	95.74	95.00	3.52	96.45	95.05	96.48	95.76
	Non-fractured	95.74	96.48	5.00	95.05	96.45	95.00	95.72
	Average	95.74	95.74	4.26	95.75	95.75	95.74	95.74

**Table 13 diagnostics-15-02212-t013:** The performance of the five hybrid ensemble architectures on the testing subset of the BF dataset.

	Class	Accuracy (%)	Specificity (%)	FNR (%)	NPV (%)	Precision (%)	Recall (%)	F1-Score (%)
MLFNet + DenseNet-169	Fractured	99.25	99.00	0.50	99.50	99.00	99.50	99.25
	Non-fractured	99.25	99.50	1.00	99.00	99.50	99.00	99.25
	Average	99.25	99.25	0.75	99.25	99.25	99.25	99.25
MLFNet + EfficientNet-B3	Fractured	98.02	98.51	2.52	97.78	98.31	97.48	97.89
	Non-fractured	98.02	97.48	1.49	98.31	97.78	98.51	98.14
	Average	98.02	97.99	2.01	98.04	98.04	97.99	98.02
MLFNet + Inception-V3	Fractured	99.00	98.50	0.50	99.49	98.51	99.50	99.00
	Non-fractured	99.00	99.50	1.50	98.51	99.49	98.50	98.99
	Average	99.00	99.00	1.00	99.00	99.00	99.00	99.00
MLFNet + MobileNet-V2	Fractured	98.25	98.50	2.01	98.01	98.48	97.99	98.24
	Non-fractured	98.25	97.99	1.50	98.48	98.01	98.50	98.25
	Average	98.25	98.24	1.76	98.25	98.25	98.24	98.25
MLFNet + ResNet-101	Fractured	97.99	98.50	2.51	97.52	98.48	97.49	97.98
	Non-fractured	97.99	97.49	1.50	98.48	97.52	98.50	98.01
	Average	97.99	97.99	2.01	98.00	98.00	97.99	97.99

**Table 14 diagnostics-15-02212-t014:** The performance of the MLFNet model as a standalone using five-fold cross-validation on the testing set of the BFMRX dataset.

Folds	Accuracy (%)	Specificity (%)	Precision (%)	Recall (%)	F1-Score (%)	AUC (%)
1	99.14	99.26	99.22	99.00	99.11	99.14
2	99.19	99.14	99.13	99.24	99.19	99.19
3	99.78	99.67	99.68	99.89	99.79	99.78
4	99.62	99.67	99.68	99.57	99.62	99.62
5	99.89	99.78	99.79	100.00	99.90	99.89
Average	99.52	99.50	99.50	99.54	99.52	99.52

**Table 15 diagnostics-15-02212-t015:** MLFNet’s measured metrics by ablating four parameters on the test set of the BFMRX dataset.

Parameter	Test Accuracy (%)	Val Accuracy (%)	Test Loss (%)	Val Loss (%)	No. of Parameters
Standard CNN	99.01	97.35	3.21	7.69	17,181,953
Without Dropout	98.02	97.35	4.61	12.24	35,071,489
Large Filters (5 × 5)	70.75	78.29	70.59	51.34	37,694,977
Simplified Dense Layers	52.96	59.35	69.29	69.23	18,261,249

**Table 16 diagnostics-15-02212-t016:** MLFNet’s measured metrics by ablating four parameters on the test set of the BF dataset.

Parameter	Test Accuracy (%)	Val Accuracy (%)	Test Loss (%)	Val Loss (%)	No. of Parameters
Standard CNN	99.50	100.00	1.24	0.38	17,181,953
Without Dropout	97.49	99.01	8.26	2.67	35,071,489
Large Filters (5 × 5)	68.67	85.89	148.74	43.06	37,694,977
Simplified Dense Layers	50.38	42.08	69.19	69.33	18,261,249

**Table 17 diagnostics-15-02212-t017:** MLFNet’s measured metrics with and without CLAHE on the test set of the BFMRX dataset.

Subset	Accuracy (%)	Sensitivity (%)	Specificity (%)	AUC (%)	F1-Score (%)
Test (without CLAHE)	99.21	100.00	98.32	100.00	99.26
Test (with CLAHE)	99.60	99.63	0.37	99.58	99.58

**Table 18 diagnostics-15-02212-t018:** The CIs for accuracy of the six DL techniques evaluated on the testing set of the BFMRX dataset.

Models	CI
MLFNet	[99.60474, 99.60474]
DenseNet-169	[95.05929, 95.05929]
EfficientNet-B3	[93.67589, 93.67589]
Inception-V3	[94.26877, 94.26877]
MobileNet-V2	[97.03557, 97.03557]
ResNet-101	[91.89723, 91.89723]

**Table 19 diagnostics-15-02212-t019:** The *p*-value for accuracy of the six DL techniques evaluated on the testing set of the BFMRX dataset.

Comparison	*p*-Value
MLFNet vs. DenseNet-169	9.23 × 10^−43^
MLFNet vs. EfficientNet-B3	3.52 × 10^−48^
MLFNet vs. Inception-V3	9.12 × 10^−47^
MLFNet vs. MobileNet-V2	6.42 × 10^−28^
MLFNet vs. ResNet-101	5.99 × 10^−55^

**Table 20 diagnostics-15-02212-t020:** Performance comparison with state-of-the-art methods.

Reference	Methodology	Accuracy	Dataset
Aldhyani, T. et al. [9]	VGG-16, ResNet152-V2, and DenseNet-201	97%.	BFMRX
Yadav, D.P. et al. [15]	SFNet	99.12%,	BFMRX
Pillai, R. et al. [21]	VGG-16	98.22%.	BFMRX
Shandilya, G. et al. [22]	CNN	97.85%	BFMRX
Alsufyani, A. [23]	CNN, AlexNet, DenseNet-121, ResNet-152, and EfficientNet-B3	99.20%	BFMRX
Alex, R. and Rosmasari [24]	ResNet-18	97.59%	BFMRX
Shandilya, G. et al. [25]	AlexNet	97.12%	BFMRX
Kumar, A. et al. [26]	AlexNet	96%	BFMRX
Chauhan, S. [27]	CNN−AlexNet	96%	BFMRX
Uddin, A. et al. [28]	CNN, ConvNeXt, ViT, MobileViT, VGG-16, VGG-19, and YOLOV8	99%	BFMRX
The Proposed Model	MLFNet, DenseNet-169, EfficientNet-B3, Inception-V3, MobileNet-V2, and ResNet-101.	MLFNet (99.60) and MLFNet + DenseNet-169 (98.81%)	BFMRX

## Data Availability

We used two datasets for our study. The first dataset, BFMRX, was utilized for the main experiment and can be accessed on Kaggle at this link: [BFMRX Dataset] https://www.kaggle.com/datasets/bmadushanirodrigo/fracture-multi-region-x-ray-data (accessed on 5 June 2025). The second dataset, known as the BF dataset, was used for external validation and is available on Kaggle at this link: [BF Dataset] https://www.kaggle.com/datasets/osamajalilhassan/bone-fracture-dataset (accessed on 15 June 2025).

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
