# Peer review of "FracFusionNet: A Multi-Level Feature Fusion Convolutional Network for Bone Fracture Detection in Radiographic Images"

_diagnostics, 2025, doi:10.3390/diagnostics15172212_

Round 1
Reviewer 1 Report
Comments and Suggestions for Authors
The manuscript presents a promising approach for bone fracture detection using the proposed MLFNet architecture. The results are good; however, several important aspects need to be addressed to strengthen the scientific rigor, novelty, and generalizability of the work.
In the literature review, many models are discussed; however, the differences between MLFNet and these models have not been examined in depth at a mathematical or theoretical level.
The distinction of MLFNet should be clarified through a technical formulation, mathematical explanation, or the description of a unique block design.A table or diagram comparing “previous methods vs. MLFNet” (architectural novelty table) should be provided.The performance improvement should be supported not only by accuracy but also by additional metrics such as parameter count, inference time, and memory usage.
Additional performance plots such as AUC and precision–recall curves should be included.
While the reported performance metrics are very high, no statistical significance analysis (confidence intervals, p-values) is provided. These tests should be added.
The generalization tests are limited to a single dataset; additional datasets or cross-validation should be used to support the results.
The manuscript has been thoroughly revised for English language and clarity.
Comments on the Quality of English Language
The manuscript has been thoroughly revised for English language and clarity.
Author Response
Dear Ms. Ziana He,
Thank you very much for allowing us to submit a revised version of our manuscript. We thank all the reviewers for their positive feedback and thoughtful comments. The updated version has incorporated their suggestions for improving the manuscript and highlighting its contributions. All the reviewers' concerns have been considered. Those changes are highlighted in the revised paper. We uploaded (a) our point-by-point response to the comments (below) with specific details about the changes that were made in our revised manuscript, (b) an updated manuscript with yellow highlighting indicating changes, and (c) a clean, updated manuscript without highlights.
Best regards,
Dr. Sameh Abd El-Ghany
Response for Reviewer #1 Comments
Comment #1:
In the literature review, many models are discussed; however, the differences between MLFNet and these models have not been examined in depth at a mathematical or theoretical level.
Response:
Thank you for your valuable and constructive feedback. We greatly appreciate your positive comments about the differences between MLFNet and other models.
Action:
The theoretical differences between MLFNet and other models were discussed in the section titled “Research Methods and Materials.” This explanation provided a detailed comparison of MLFNet and related models on a theoretical level. We specifically emphasized the architectural differences using formal notations and examined the theoretical advantages of MLFNet regarding its complexity and convergence behavior. Please see Research Methods and Materials on page 15.
Table 3. A comparative architectural novelty between the MLFNet and the other models.
|
Feature / Component |
DenseNet169 |
EfficientNet B3 |
InceptionV3 |
MobileNet-V2 |
ResNet-101 |
MLFNet |
|
|
Input Resolution |
128×128×3 (X-ray) |
128×128×3 (X-ray) |
128×128×3 (X-ray) |
128×128×3 (X-ray) |
128×128×3 (X-ray) |
128×128×3 (X-ray) |
|
|
Preprocessing |
Standard normalization |
Standard normalization |
Standard normalization |
Standard normalization |
Standard normalization |
Normalization + X-ray enhancement |
|
|
Core Building Block |
Dense block (dense connections) |
MBConv + SE block (compound scaling) |
Inception module (multi-branch conv) |
Inverted residual blocks |
Residual blocks |
SFNet block (multi-path Conv2D + concatenation) |
|
|
Low-Level Feature Extraction |
Initial Conv + Pooling |
Initial Conv + Pooling |
Stem conv layers |
Conv + BatchNorm |
Conv + BatchNorm |
SFNetBlock with 64 filters (edge features) |
|
|
Mid-Level Feature Extraction |
Dense blocks |
MBConv |
Inception modules |
Depthwise-separable conv |
Residual blocks |
SFNetBlock with 128 filters (semantic features) |
|
|
High-Level Feature Extraction |
Deeper dense blocks |
Deeper MBConv |
Larger Inception filters |
Bottleneck blocks |
Deep residual stacks |
SFNetBlock with 256 filters (abstract features) |
|
|
Regularization |
Dropout (optional) |
Dropout + SE |
Dropout |
Dropout |
Dropout |
Dropout after mid & high-level blocks |
|
|
Feature Fusion Strategy |
Dense concatenation |
Compound scaling |
Multi-branch merge |
Sequential bottlenecks |
Residual skip connections |
Multi-path feature concatenation across SFNet blocks |
|
|
Classifier Head |
Global Avg Pool + Dense |
Global Avg Pool + Dense |
Global Avg Pool + Dense |
Global Avg Pool + Dense |
Global Avg Pool + Dense |
Flatten → Dense(256) → Dense(128) → Sigmoid |
|
|
Domain Specialization |
General-purpose |
General-purpose |
General-purpose |
Mobile/edge |
General-purpose |
Optimized for X-ray fracture detection |
|
We have carefully reviewed your comments and made the necessary changes. We genuinely appreciate your dedicated efforts and hope these revisions will meet with your approval.
Comment #2:
The distinction of MLFNet should be clarified through a technical formulation, mathematical explanation, or the description of a unique block design. A table or diagram comparing “previous methods vs. MLFNet” (architectural novelty table) should be provided. The performance improvement should be supported not only by accuracy but also by additional metrics such as parameter count, inference time, and memory usage.
Response:
Thank you for your valuable and constructive feedback. We greatly appreciate your positive comments regarding the comparison between previous models vs. MLFNet.
Action:
In response to clarify the MLFNet, we have explained of the three stacked SFNet blocks. Please see Research Methods and Materials on pages 15-16.
In response to comparing previous models vs. MLFNet, we have created a comparison table between the previous models vs. MLFNet. Please see Research Methods and Materials on page 21.
|
Feature / Component |
DenseNet169 |
EfficientNet B3 |
InceptionV3 |
MobileNet-V2 |
ResNet-101 |
MLFNet (Proposed) |
|
Input Resolution |
128×128×3 (X-ray) |
128×128×3 (X-ray) |
128×128×3 (X-ray) |
128×128×3 (X-ray) |
128×128×3 (X-ray) |
128×128×3 (X-ray) |
|
Preprocessing |
Standard normalization |
Standard normalization |
Standard normalization |
Standard normalization |
Standard normalization |
Normalization + X-ray enhancement |
|
Core Building Block |
Dense block (dense connections) |
MBConv + SE block (compound scaling) |
Inception module (multi-branch conv) |
Inverted residual blocks |
Residual blocks |
SFNet block (multi-path Conv2D + concatenation) |
|
Low-Level Feature Extraction |
Initial Conv + Pooling |
Initial Conv + Pooling |
Stem conv layers |
Conv + BatchNorm |
Conv + BatchNorm |
SFNetBlock with 64 filters (edge features) |
|
Mid-Level Feature Extraction |
Dense blocks |
MBConv |
Inception modules |
Depthwise-separable conv |
Residual blocks |
SFNetBlock with 128 filters (semantic features) |
|
High-Level Feature Extraction |
Deeper dense blocks |
Deeper MBConv |
Larger Inception filters |
Bottleneck blocks |
Deep residual stacks |
SFNetBlock with 256 filters (abstract features) |
|
Regularization |
Dropout (optional) |
Dropout + SE |
Dropout |
Dropout |
Dropout |
Dropout after mid & high-level blocks |
|
Feature Fusion Strategy |
Dense concatenation |
Compound scaling |
Multi-branch merge |
Sequential bottlenecks |
Residual skip connections |
Multi-path feature concatenation across SFNet blocks |
|
Classifier Head |
Global Avg Pool + Dense |
Global Avg Pool + Dense |
Global Avg Pool + Dense |
Global Avg Pool + Dense |
Global Avg Pool + Dense |
Flatten → Dense(256) → Dense(128) → Sigmoid |
|
Domain Specialization |
General-purpose |
General-purpose |
General-purpose |
Mobile/edge |
General-purpose |
Optimized for X-ray fracture detection |
This table clearly shows:
- Architectural novelty: MLFNet’s SFNet block, multi-path fusion, and medical image-specific preprocessing.
- Task specialization: Baseline models are general-purpose, while MLFNet is designed for X-ray fracture detection.
- Feature extraction pipeline: Highlights the low-, mid-, and high-level pathways unique to MLFNet.
In response to supporting other performance metrics, we have created a table showing the other performance metrics. Please see Evaluation and Analysis on pages 24-25.
|
Model |
Train Time (s)
|
Inference Time (s)
|
Memory Usage (MB)
|
Params Count
|
|
MLFNet
|
81.44337928
|
12.0270946
|
10310.46094
|
35070000 |
|
DenseNet169
|
419.1784675
|
34.20856285
|
13284.18164
|
13076033
|
|
EfficientNet-B3
|
304.9635962
|
23.30317223
|
15891.57031
|
11183408
|
|
InceptionV3
|
161.1117506
|
19.53114045
|
17414.4707
|
22335777
|
|
MobileNet-V2
|
114.5730445
|
14.87542236
|
18448.73633
|
2591297
|
|
ResNet-101
|
291.2739118
|
28.60760403
|
20855.83203
|
43191169
|
We have carefully reviewed your comments and made the necessary changes. We genuinely appreciate your dedicated efforts and hope these revisions will meet with your approval.
Comment #3:
Additional performance plots such as AUC and precision–recall curves should be included.
Response:
Thank you for this valuable suggestion. We appreciate the suggestion to offer a more comprehensive evaluation of the classification performance.
Action:
In response to adding additional performance plots, we have included two plots for the Receiver Operating Characteristic (ROC) curves along with their Area Under the Curve (AUC) values and the Precision-Recall (PR) curves for all models being compared. These plots are designed to provide a more thorough assessment of classification performance, focusing on discriminative ability and the trade-offs between precision and recall. Please see Evaluation and Analysis on pages 27-30.
Figure 11 displays the Receiver Operating Characteristic (ROC) curves for the six DL models. The ROC curve for MLFNet showed nearly perfect classification performance, closely hugging the top-left corner of the plot and achieving an Area Under the Curve (AUC) score of 1.00. This indicates that MLFNet is effectively distinguished between fractured and non-fractured X-ray images without compromising sensitivity or specificity. The curve's steep ascent and maximal AUC highlight both high sensitivity and specificity, demonstrating the robustness of the proposed architecture.
DenseNet-169 also performed well, achieving an AUC of 0.95. Its ROC curve rose sharply toward the top-left corner but fell slightly short of MLFNet’s ideal path. This performance suggests a strong ability to differentiate between the two classes, although some misclassifications could occur compared to MLFNet’s perfect separation.
EfficientNet-B3 matched DenseNet-169 with an AUC of 0.95, following a similar trajectory in its ROC curve, which featured a steep rise initially and a gradual approach toward the upper right. While its performance was commendable, it did not achieve the flawless separation seen with MLFNet.
Inception-V3 outperformed both DenseNet-169 and EfficientNet-B3 with an AUC of 0.96, indicating a slightly better balance between sensitivity and specificity. This suggests it managed borderline cases a bit more effectively but still did not match MLFNet's perfect discrimination.
MobileNet-V2 achieved an AUC of 0.97, surpassing DenseNet-169, EfficientNet-B3, and Inception-V3. Its ROC curve indicated near-perfect classification capability, particularly excelling in minimizing false positives. Nevertheless, it still showed minor deviations from the ideal trajectory compared to MLFNet.
ResNet-101 demonstrated the weakest performance among the six models, with an AUC of 0.92. Although its ROC curve steadily rose toward the top-left, it lagged in the early part of the curve, suggesting more false positives at lower thresholds. This indicates that ResNet-101 had comparatively less discriminative power in this classification task.
Overall, MLFNet clearly outperformed the other models, achieving a perfect AUC of 1.00 and an ideal ROC curve shape. MobileNet-V2 followed closely with an AUC of 0.97, while Inception-V3 slightly surpassed both DenseNet-169 and EfficientNet-B3, which both had an AUC of 0.95. ResNet-101 trailed with an AUC of 0.92, indicating the least effective discrimination between classes. Ultimately, MLFNet demonstrated superior discriminative ability, confirming the effectiveness of its tailored architecture for fracture detection.
Figure 11. The ROC curves of the six DL methods over the testing set of the BFMRX dataset.
Figure 12 presents the analysis of precision-recall curves for the six DL models. MLFNet showed excellent performance, with a precision-recall curve area of 0.99 for the 'Fractured' class and a perfect score of 1.00 for the Non_Fractured class. Both curves remained high across the recall range, indicating consistently high precision even as recall increased. This suggests that MLFNet was highly effective at accurately identifying both fractured and non-fractured cases with minimal false positives.
DenseNet-169 also performed strongly, achieving areas of 0.92 for the 'Fractured' class and 0.93 for the Non_Fractured class. The precision remained high for both classes across most of the recall range, with only a slight drop at higher recall values. This indicates that DenseNet-169 was generally robust in its predictions, maintaining good precision while identifying a significant number of relevant instances.
EfficientNet-B3 demonstrated good performance, with areas of 0.88 for the 'Fractured' class and 0.94 for the Non_Fractured class. The curve for the Non_Fractured class was notably higher and more stable than that for 'Fractured', which showed a more pronounced decline in precision at higher recall levels. This indicates that while EfficientNet-B3 performed well overall, it was more adept at accurately identifying non-fractured cases.
Inception-V3 achieved solid performance, with areas of 0.91 for the 'Fractured' class and 0.92 for the Non_Fractured class. Both curves maintained high precision across a substantial portion of the recall range, indicating a reliable ability to classify instances correctly. Similar to DenseNet-169, there was a noticeable dip in precision at very high recall, suggesting a trade-off in performance when attempting to capture nearly all relevant instances.
MobileNet-V2 exhibited strong and consistent performance, with areas of 0.95 for the 'Fractured' class and 0.96 for the Non_Fractured class. Both precision-recall curves remained high and stable across the entire recall range, indicating that MobileNet-V2 maintained excellent precision even at high recall levels for both classes. This suggests that it was highly effective and balanced in its classification capabilities.
ResNet-101 showed acceptable performance, with areas of 0.86 for the 'Fractured' class and 0.92 for the Non_Fractured class. The curve for the Fractured class exhibited a more significant drop in precision at higher recall values compared to the other models, indicating a greater challenge in maintaining precision for this class. However, the 'Non_Fractured' class performed considerably better, maintaining higher precision across the recall range.
When comparing the precision-recall curves across all six models, several key observations emerged. MLFNet and MobileNet-V2 consistently demonstrated the strongest performance. MLFNet achieved near-perfect scores, particularly for the Non_Fractured class (area = 1.00), while maintaining exceptionally high precision for Fractured cases (area = 0.99). MobileNet-V2 also showed outstanding and balanced performance, with areas of 0.95 and 0.96 for the Fractured and Non_Fractured classes, respectively.
DenseNet-169 and Inception-V3 exhibited strong, comparable performance, with area scores generally in the low 0.90s for both classes. These models maintained good precision for a significant portion of the recall range but demonstrated a more noticeable drop at very high recall values, suggesting a slight trade-off between precision and recall at extreme ends.
EfficientNet-B3 performed well, especially for the Non_Fractured class (area = 0.94), but its performance for the Fractured class (area = 0.88) was slightly lower and less stable compared to the top performers.
ResNet-101 generally showed the lowest performance among the models, particularly for the Fractured class (area = 0.86), where its precision dropped more significantly to higher recall levels. While its performance for the Non_Fractured class (area = 0.92) was respectable, the disparity between the two classes was more pronounced in ResNet-101 than in the other models.
Overall, MLFNet and MobileNet-V2 stood out for their superior and consistent precision-recall characteristics across both classes, making them the most robust choices for this multi-class classification task.
Figure 12. The Precision-Recall curves of the six DL methods over the testing set of the BFMRX dataset.
We have carefully reviewed your comments and made the necessary changes. We genuinely appreciate your dedicated efforts and hope these revisions will meet with your approval.
Comment #4:
While the reported performance metrics are very high, no statistical significance analysis (confidence intervals, p-values) is provided. These tests should be added.
Response:
Thank you for this valuable observation. We appreciate the observation to offer a more comprehensive evaluation of the classification performance.
Action:
In response to perform statistical analysis, we have addressed this concern by adding a statistical significance analysis. Specifically, we have included table 13 on page 49 reporting the 95% confidence intervals for accuracy.
Moreover, we have performed independent two-sample t-tests comparing MLFNet with each baseline model and reporting the p-value. This addition provides a robust measure of the reliability of our results. Please see Evaluation and Analysis on pages 48-50.
4.7. Confidence Interval
Confidence intervals (CIs) play a vital role in assessing the effectiveness of DL models for identifying bone fractures. They measure the uncertainty surrounding point estimates of important metrics such as sensitivity, specificity, or area under the curve (AUC). A narrower CI reflects greater precision in the model's predicted performance on new, unseen data, indicating enhanced reliability. For example, if a DL model demon-strates 95% sensitivity with a 95% CI of [93%, 97%], it provides stronger evidence of effective fracture detection compared to another model with the same sensitivity but a broader CI of [88%, 99%]. Including CIs when reporting performance metrics is crucial for comprehending the statistical significance and clinical relevance of deep learning systems for fracture detection, aiding in the evaluation of their potential for practical use in real-world scenarios [41]. Table 18 shows the CIs for accuracy of the six DL techniques evaluated on the test seting of the BFMRX dataset.
Table 18 and Figure 20 display the 95% CIs for the effectiveness of six DL models assessed on the EBTC test set. MLFNet achieved the highest accuracy, with a CI of [99.60474, 99.60474], reflecting a very consistent performance. Following closely was MobileNet-V2, which had a CI of [97.03557, 97.03557], indicating strong and stable re-sults as well. DenseNet-169 and Inception-V3 demonstrated moderate performance, with CIs of [95.05929, 95.05929] and [94.26877, 94.26877], respectively. EfficientNet-B3 showed slightly lower accuracy, recording a CI of [93.67589, 93.67589]. ResNet-101 ex-hibited the lowest performance, with a CI of [91.89723, 91.89723]. The narrow nature of all CIs (single-point values) suggests that there was no variability in the model outputs across the test set for the metrics used in the evaluation.
Table 18. The CIs for accuracy of the six DL techniques evaluated on the testing set of the BFMRX dataset.
|
Models |
CI |
|
MLFNet |
[99.60474, 99.60474] |
|
DenseNet-169 |
[95.05929, 95.05929] |
|
EfficientNet-B3 |
[93.67589, 93.67589] |
|
Inception-V3 |
[94.26877, 94.26877] |
|
MobileNet-V2 |
[97.03557, 97.03557 |
|
ResNet-101 |
[91.89723, 91.89723] |
Figure 20. The CIs for accuracy of the six DL techniques evaluated on the testing set of the BFMRX dataset.
4.8. P-Value
A p-value measures the likelihood of obtaining results that are at least as extreme as those observed in a statistical test, assuming that the null hypothesis is true (which posits no real difference between the models being compared). In evaluating model perfor-mance, the null hypothesis usually asserts that there is no significant difference in per-formance metrics (such as accuracy or AUC) between two models.
In DL, especially in medical imaging, performance metrics like accuracy, sensitivity, and AUC can be affected by factors such as dataset sampling, model initialization, or random training processes. Even if one model shows better performance, the difference may simply result from random fluctuations rather than a genuine improvement.
By performing a statistical significance test (like an independent two-sample t-test, McNemar’s test, or bootstrapping) and reporting the p-value, researchers can assess the likelihood that any observed performance difference is due to chance. A low p-value (typically < 0.05) indicates statistical evidence that the improvement is unlikely to be random, thereby reinforcing the credibility of claims regarding the superiority of a proposed DL model.
In this study, analyzing the p-value, as shown in Table 19 and Figure 21, was crucial to confirming that MLFNet's higher accuracy compared to baseline architectures was not due to random chance but indicated a statistically significant performance enhancement. To evaluate whether the differences in accuracy were statistically significant, we per-formed independent two-sample t-tests comparing MLFNet with each baseline model. The results showed p-values of 9.23 × 10⁻⁴³ for DenseNet-169, 3.52 × 10⁻⁴⁸ for Efficient-Net-B3, 9.12 × 10⁻⁴⁷ for Inception-V3, 6.42 × 10⁻²⁸ for MobileNet-V2, and 5.99 × 10⁻⁵⁵ for ResNet-101. All of these values were well below the 0.05 significance threshold, indi-cating that MLFNet’s improvement in accuracy compared to the baseline models was statistically significant.
Table 19. The P-value for accuracy of the six DL techniques evaluated on the testing set of the BFMRX dataset.
|
Comparison |
p-value |
|
MLFNet vs. DenseNet-169 |
9.23 × 10⁻⁴³ |
|
MLFNet vs. EfficientNet-B3 |
3.52 × 10⁻⁴⁸ |
|
MLFNet vs. Inception-V3 |
9.12 × 10⁻⁴⁷ |
|
MLFNet vs. MobileNet-V2 |
6.42 × 10⁻²⁸ |
|
MLFNet vs. ResNet-101 |
5.99 × 10⁻⁵⁵ |
Figure 21. The P-value for accuracy of the six DL techniques evaluated on the testing set of the BFMRX dataset.
We have carefully reviewed your comments and made the necessary changes. We genuinely appreciate your dedicated efforts and hope these revisions will meet with your approval.
Comment #5:
The generalization tests are limited to a single dataset; additional datasets or cross-validation should be used to support the results.
Response:
Thank you for raising this important point. We appreciate your comment regarding the generalization.
Action:
In response to general tests, we have supplemented the evaluation with an external validation set derived from Bone Fracture dataset to further test the model on unseen data to evaluate the robustness of the proposed model. Please see Evaluation and Analysis on pages 41-45.
We implemented two experiments to validate the proposed MLFNet model externally. In the first experiment, we used the BF dataset. This dataset contains about 9463 image samples of fractured and non-fractured bones. It was divided into: testing set (600 images) and training set (8,863 images). External validation tests whether the MLFNet model can perform well on unseen, independent data that may differ in imaging protocols, patient demographics, or equipment. It ensures that the MLFNet model was not overfitted to the training dataset, and a MLFNet model that performs well on external data is more likely to generalize to real-world clinical settings.
In the first experiment of the external validation process, the MLFNe model was evaluated using the BF dataset for binary classification (fracture vs. non-fracture). MLFNet achieved an impressive accuracy of 99.50% as a standalone model and 99.25% when integrated into hybrid ensemble architectures with the same five pre-trained DL models. The results of the standalone and hybrid MLFNet are shown in Tables 12 and 13, respectively.
Table 12 shows that MLFNet excelled, achieving an average accuracy of 99.50%, with perfect recall (100%) for fractured cases and specificity (100%) for non-fractured ones. It had zero FNs for fractured cases and only 1% FNR for non-fractured cases, indicating an exceptionally balanced and robust model. MLFNet consistently outperformed all other models in every metric, followed closely by DenseNet-169, Inception-V3, and MobileNet-V2, while EfficientNet-B3 and ResNet-101 showed slightly lower but still acceptable performance. These findings confirm MLFNet's superior generalization, sensitivity, and specificity, making it the most reliable model in the tested cohort.
Table 12. The performance of the six CNN models as standalones on the testing set of the BF dataset.
|
MLFNet |
Class |
Accuracy (%) |
Specificity (%) |
FNR (%) |
NPV (%) |
Precision (%) |
Recall (%) |
F1-score (%) |
|
Fractured |
99.50 |
99.00 |
0.00 |
100.00 |
99.00 |
100.00 |
99.50 |
|
|
Non-fractured |
99.50 |
100.00 |
1.00 |
99.00 |
100.00 |
99.00 |
99.50 |
|
|
Average |
99.50 |
99.50 |
0.50 |
99.50 |
99.50 |
99.50 |
99.50 |
|
|
DenseNet-169 |
Fractured |
99.00 |
98.00 |
0.00 |
100.00 |
98.03 |
100.00 |
99.00 |
|
Non-fractured |
99.00 |
100.00 |
2.00 |
98.03 |
100.00 |
98.00 |
98.99 |
|
|
Average |
99.00 |
99.00 |
1.00 |
99.01 |
99.01 |
99.00 |
99.00 |
|
|
EfficientNet-B3
|
Fractured |
96.99 |
97.50 |
3.52 |
96.53 |
97.46 |
96.48 |
96.97 |
|
Non-fractured |
96.99 |
96.48 |
2.50 |
97.46 |
96.53 |
97.50 |
97.01 |
|
|
Average |
96.99 |
96.99 |
3.01 |
97.00 |
97.00 |
96.99 |
96.99 |
|
|
Inception-V3 |
Fractured |
97.74 |
96.00 |
0.50 |
99.48 |
96.12 |
99.50 |
97.78 |
|
Non-fractured |
97.74 |
99.50 |
4.00 |
96.12 |
99.48 |
96.00 |
97.71 |
|
|
Average |
97.74 |
97.75 |
2.25 |
97.80 |
97.80 |
97.75 |
97.74 |
|
|
MobileNet-V2 |
Fractured |
97.74 |
98.50 |
3.02 |
97.04 |
98.47 |
96.98 |
97.72 |
|
Non-fractured |
97.74 |
96.98 |
1.50 |
98.47 |
97.04 |
98.50 |
97.77 |
|
|
Average |
97.74 |
97.74 |
2.26 |
97.76 |
97.76 |
97.74 |
97.74 |
|
|
ResNet-101 |
Fractured |
95.74 |
95.00 |
3.52 |
96.45 |
95.05 |
96.48 |
95.76 |
|
Non-fractured |
95.74 |
96.48 |
5.00 |
95.05 |
96.45 |
95.00 |
95.72 |
|
|
Average |
95.74 |
95.74 |
4.26 |
95.75 |
95.75 |
95.74 |
95.74 |
Figure 15 shows the training and validation loss and accuracy of the MLFNet model over 30 epochs. At the start, the training loss was high, around 2.00, but it quickly dropped during the first few epochs. The validation loss also decreased significantly, beginning at about 0.50. As training continued from epochs 5 to 30, both training and validation losses kept declining, though at a slower rate, stabilizing between 0.00 and 0.25. Importantly, the validation loss stayed at or just below the training loss during this time, indicating that the model was generalizing well and not overfitting based on the loss metric.
In the early epochs, training accuracy began low at around 0.60 but increased rapidly. Similarly, validation accuracy rose sharply from an initial value of about 0.80. By epoch 10, both accuracies approached 0.95. For the remainder of the training (epochs 10-30), both training and validation accuracies remained high, fluctuating between 0.95 and 1.00. The validation accuracy often exceeded or closely matched the training accuracy, further supporting the conclusion that the model was effectively generalizing and avoiding overfitting.
Overall, the MLFNet model displayed strong learning capabilities, as shown by the quick reduction in loss and increase in accuracy during the initial epochs. The consistent performance of the validation metrics compared to the training metrics suggested that the model was generalizing well to unseen data and not overfitting. Both the loss and accuracy curves stabilized at favorable levels, indicating that the model had converged and achieved robust performance by the end of the training period.
Figure 15. The training and validation loss, as well as the training and validation accuracy of MLFNet on the testing set of the BF dataset.
Table 13 shows that all the hybrid ensemble models achieved high classification accuracy, though their performance varied across individual metrics. MLFNet+DenseNet-169 hybrid achieved the highest average accuracy of 99.25%, reflecting its exceptional classification capability. It demonstrated highly balanced performance across both classes, with a recall and specificity of 99.25%, and a very low average FNR of 0.75%, indicating excellent sensitivity and precision. MLFNet + DenseNet-169 emerged as the best-performing model, followed closely by MLFNet + Inception-V3, while MLFNet + ResNet-101 ranked lowest among the tested hybrid configurations. Nevertheless, all models demonstrated robust performance, confirming the advantages of combining MLFNet with established backbone architectures to enhance fracture classification accuracy.
Table 13. The performance of the five hybrid ensemble architectures on the testing set of the BF dataset.
|
MLFNet + DenseNet-169 |
Class |
Accuracy (%) |
Specificity (%) |
FNR (%) |
NPV (%) |
Precision (%) |
Recall (%) |
F1-score (%) |
|
Fractured |
99.25 |
99.00 |
0.50 |
99.50 |
99.00 |
99.50 |
99.25 |
|
|
Non-fractured |
99.25 |
99.50 |
1.00 |
99.00 |
99.50 |
99.00 |
99.25 |
|
|
Average |
99.25 |
99.25 |
0.75 |
99.25 |
99.25 |
99.25 |
99.25 |
|
|
MLFNet + EfficientNet-B3 |
Fractured |
98.02 |
98.51 |
2.52 |
97.78 |
98.31 |
97.48 |
97.89 |
|
Non-fractured |
98.02 |
97.48 |
1.49 |
98.31 |
97.78 |
98.51 |
98.14 |
|
|
Average |
98.02 |
97.99 |
2.01 |
98.04 |
98.04 |
97.99 |
98.02 |
|
|
MLFNet + Inception-V3 |
Fractured |
99.00 |
98.50 |
0.50 |
99.49 |
98.51 |
99.50 |
99.00 |
|
Non-fractured |
99.00 |
99.50 |
1.50 |
98.51 |
99.49 |
98.50 |
98.99 |
|
|
Average |
99.00 |
99.00 |
1.00 |
99.00 |
99.00 |
99.00 |
99.00 |
|
|
MLFNet + MobileNet-V2 |
Fractured |
98.25 |
98.50 |
2.01 |
98.01 |
98.48 |
97.99 |
98.24 |
|
Non-fractured |
98.25 |
97.99 |
1.50 |
98.48 |
98.01 |
98.50 |
98.25 |
|
|
Average |
98.25 |
98.24 |
1.76 |
98.25 |
98.25 |
98.24 |
98.25 |
|
|
MLFNet + ResNet-101 |
Fractured |
97.99 |
98.50 |
2.51 |
97.52 |
98.48 |
97.49 |
97.98 |
|
Non-fractured |
97.99 |
97.49 |
1.50 |
98.48 |
97.52 |
98.50 |
98.01 |
|
|
Average |
97.99 |
97.99 |
2.01 |
98.00 |
98.00 |
97.99 |
97.99 |
In the second experiment of the external validation process, we evaluated the MLFNet model by applying the stratified 5-fold cross-validation scheme. In the stratified 5-fold cross-validation scheme, the dataset was divided into five equal partitions, with each fold taking a turn as the testing set while the other four folds were used for model training. This rotation continued until every fold had been used once for testing. Finally, the outcomes from all folds were averaged to obtain a stable and unbiased estimate of the model’s performance.
The MLFNet model was evaluated on the BFMRX dataset for binary classification (fracture vs. non-fracture). MLFNet achieved an impressive accuracy of 99.52% as a standalone model. The results of the standalone MLFnet are shown in Tables 14.
Table 14 shows that the MLFNet model's performance was outstanding. It achieved an average accuracy of 99.52%, meaning it correctly classified the vast majority of instances across all test folds. This high level of accuracy was consistent with other metrics, as shown by the average F1-score and AUC also being 99.52%. This indicated a robust and highly effective model with virtually no drop in performance between the training and validation phases.
The results were remarkably consistent across all five folds, with minimal fluctuation. The accuracy ranged from a high of 99.89% in fold 5 to a low of 99.14% in fold 1, a very narrow range of just 0.75%. This consistency suggested that the model was not overly dependent on a particular subset of the data and generalized well to unseen data from the same distribution.
Table 14. The performance of MLFNet model as standalone using five-folds cross validation on the testing set of the BFMRX dataset.
|
Folds |
Accuracy (%) |
Specificity (%) |
Precision (%) |
Recall (%) |
F1-score (%) |
AUC (%) |
|
1 |
99.14 |
99.26 |
99.22 |
99.00 |
99.11 |
99.14 |
|
2 |
99.19 |
99.14 |
99.13 |
99.24 |
99.19 |
99.19 |
|
3 |
99.78 |
99.67 |
99.68 |
99.89 |
99.79 |
99.78 |
|
4 |
99.62 |
99.67 |
99.68 |
99.57 |
99.62 |
99.62 |
|
5 |
99.89 |
99.78 |
99.79 |
100.00 |
99.90 |
99.89 |
|
Average |
99.52 |
99.50 |
99.50 |
99.54 |
99.52 |
99.52 |
Figure 16 illustrates the training and validation loss across multiple folds during the model training process. It showed that for most folds, the training and validation losses rapidly decreased within the first few epochs and then stabilized near zero, indicating effective learning and convergence. However, Fold 3 exhibited a significant spike in training loss, reaching a peak of approximately 25 around epoch 4, before sharply dropping to near zero. Similarly, the validation loss for Fold 1 showed a smaller spike, reaching around 6, but also quickly recovered. These spikes indicated that the model experienced temporary instability or potential numerical fluctuations during early training stages in some folds. Overall, after the initial few epochs, both training and validation losses across all folds remained consistently low, suggesting that the model eventually achieved stable and well-generalized performance.
Figure 16. The training and validation loss of MLFNet using five-folds cross validation on the testing set of the BFMRX dataset.
Figure 17 illustrates the training and validation accuracy across multiple folds during the model training process. It showed that, for most folds, accuracy increased rapidly within the first few epochs and quickly stabilized near 100%, indicating effective learning. In the early epochs, some folds experienced temporary fluctuations. For example, Train Fold 1 started with lower accuracy, around 78%, but quickly improved and converged near 100%. Similarly, Val Fold 2 showed a noticeable drop to around 85% before recovering. Train Fold 3 also exhibited a brief decline to about 90% around epoch 3, but it rapidly stabilized thereafter. After approximately the first 5 epochs, both training and validation accuracies across all folds consistently remained above 99%, suggesting that the model achieved excellent generalization and stable performance across the cross-validation folds.
Figure 17. The training and validation accuracy of MLFNet using five-folds cross validation on the testing set of the BFMRX dataset.
We have carefully reviewed your comments and made the necessary changes. We genuinely appreciate your dedicated efforts and hope these revisions will meet with your approval.
Comment #6:
The manuscript has been thoroughly revised for English language and clarity.
Response:
Thank you for your valuable comment. We appreciate your comment to improve readability and precision.
Action:
In response to improve readability and precision of the manuscript, the manuscript has been thoroughly revised for English language, grammar, and clarity. All sentences have been restructured where necessary to improve readability and precision.
We have carefully reviewed your comments and made the necessary changes. We genuinely appreciate your dedicated efforts and hope these revisions will meet with your approval.

Reviewer 2 Report
Comments and Suggestions for Authors
The article proposes a multi-level feature fusion network (MLFNET) that uses both low-level and high-level image features to classify bone fractures. The number of samples in the dataset is sufficient, and there is no class imbalance.
1. The article is excessively long. There is no need to detail architectures such as Resnet and EfficientNet. I recommend presenting the parameters used as a table instead.
2. When Figures 13 and 15 are examined, the accuracy and loss changes are stable for training. However, this cannot be said for validation. It would be correct to say that the models memorize the training set. To overcome this, training should be continued for at least 10 epochs. Subsequent changes will provide clearer information.
3. The inputs of the proposed model are 128x128x3. The inputs of the models used for comparison are 224x224x3. Therefore, it is necessary to note that the comparison is unfair.
4. Figures 3 and 4 are not necessary.
5. The dimensions of the features extracted with the MLFNet+ CNN models should be given in a table.
6. The proposed model is more complex than CNNs. The average accuracy for DenseNet-169 is 99%. The average accuracy achieved with the proposed model is 99.5%. The advantages and disadvantages of increasing the complexity of the system for a 0.5% improvement should be discussed.
Author Response
Dear Ms. Ziana He,
Thank you very much for allowing us to submit a revised version of our manuscript. We thank all the reviewers for their positive feedback and thoughtful comments. The updated version has incorporated their suggestions for improving the manuscript and highlighting its contributions. All the reviewers' concerns have been considered. Those changes are highlighted in the revised paper. We uploaded (a) our point-by-point response to the comments (below) with specific details about the changes that were made in our revised manuscript, (b) an updated manuscript with yellow highlighting indicating changes, and (c) a clean, updated manuscript without highlights.
Best regards,
Dr. Sameh Abd El-Ghany
Response for Reviewer #2 Comments
Comment #1:
The article is excessively long. There is no need to detail architectures such as Resnet and EfficientNet. I recommend presenting the parameters used as a table instead.
Response:
Thank you for your valuable and constructive feedback. We appreciate your feedback regarding the manuscript length.
Action:
In response to short the manuscript length, we have removed the detailed descriptions of standard architectures such as ResNet and EfficientNet, instead, we have included a concise table summarizing the key parameters and configurations used in our experiments. This change reduces redundancy while ensuring that essential implementation details remain accessible to the reader. Please see Research Methods and Materials on pages 14-16
|
Model |
Input Shape |
Total Params |
Trainable Params |
Non-Trainable Params |
Number of Layers |
|
DenseNet169 |
(128, 128, 3) |
13077061 |
12915333 |
161728 |
600 |
|
EfficientNetB3 |
(128, 128, 3) |
11184436 |
11094061 |
90375 |
390 |
|
InceptionV3 |
(128, 128, 3) |
22336805 |
22298277 |
38528 |
316 |
|
MobileNetV2 |
(128, 128, 3) |
2592325 |
2555653 |
36672 |
159 |
|
ResNet101V2 |
(128, 128, 3) |
43160581 |
43058821 |
101760 |
382 |
We have carefully reviewed your comments and made the necessary changes. We genuinely appreciate your dedicated efforts and hope these revisions will meet with your approval.
Comment #2:
When Figures 13 and 15 are examined, the accuracy and loss changes are stable for training. However, this cannot be said for validation. It would be correct to say that the models memorize the training set. To overcome this, training should be continued for at least 10 epochs. Subsequent changes will provide clearer information.
Response:
Thank you for your valuable comment. We appreciate the comment regarding the stability of the accuracy and loss curves for training and validation in Figures 13 and 17. We agree that unstable validation curves can be indicative of potential overfitting or insufficient convergence, as highlighted in prior studies.
We would like to clarify that the experiments presented in this study were trained for 30 epochs, which exceeds the suggested minimum of 10 epochs. This was done to ensure sufficient learning cycles and to allow the models to converge. Please see Evaluation and Analysis on pages 21-24 and 33-36
Figure 9 illustrates the training and validation performance of the six DL models over 30 epochs, displaying both loss (on the left) and accuracy (on the right) curves. For the MLFNet model, the training loss steadily decreased from around 0.48 to nearly zero, while the validation loss also dropped sharply in the early epochs before stabilizing with minor fluctuations between epochs 10 and 25, remaining low overall. This pattern indicates that the model effectively learned from the training data and maintained good generalization to unseen validation data without significant overfitting.
Regarding accuracy, the MLFNet model experienced a rapid increase during the initial epochs. Training accuracy rose from approximately 75% to nearly 100%, while validation accuracy followed a similar trend, quickly surpassing 95% and remaining stable with minor variations throughout the remaining epochs. The small gap between training and validation accuracy, along with the low and stable validation loss, suggests that the model achieved excellent generalization and high predictive performance. Overall, the close alignment between training and validation metrics demonstrates a well-optimized and robust model, capable of delivering consistent results on both seen and unseen data.
The training and validation loss plot (left) of the DenseNet-169 model across 30 epochs revealed a significant issue. While the training loss steadily decreased toward zero, the validation loss experienced a drastic spike in the first epoch, reaching approximately 1.4 × 10¹³. This likely indicated numerical instability or error, such as exploding gradients or division by zero. Following this spike, the validation loss quickly dropped to near zero and remained flat, which is not typical behavior and suggests a possible flaw in the loss computation or logging.
In the accuracy plot (right), the training accuracy consistently improved from about 78% to nearly 99%, indicating effective learning. However, the validation accuracy was quite erratic during the first 10 epochs, fluctuating between 40% and 95%. This instability likely stemmed from the issues observed in the validation loss. After the initial fluctuations, the validation accuracy stabilized and aligned more closely with the training accuracy, remaining above 90%, indicating improved generalization. Overall, while the training behavior was smooth, the unusual patterns in validation loss and early validation accuracy pointed to significant early instability in the model training process, possibly due to data irregularities, improper initialization, or learning rate issues. Addressing these factors is essential for ensuring reliable evaluation.
For the EfficientNet-B3 model over 30 epochs, on the left, you can see the loss curves, and on the right, the accuracy curves. The training loss (in red) remained consistently low and stable throughout the epochs, indicating smooth convergence during training. In contrast, the validation loss (in green) exhibited significant instability, with sharp spikes, particularly around epochs 12, 17, and 19, where the loss exceeded 5000. These sudden increases point to severe overfitting, numerical instability, or data-related issues, such as outliers or mislabeled validation samples.
Looking at accuracy, the training accuracy rose steadily from approximately 80% to nearly 100%, showing effective learning on the training set. However, the validation accuracy fluctuated sharply, especially in the early and middle epochs, and dropped significantly around epoch 21, coinciding with the spike in validation loss. Despite this drop, validation accuracy later recovered and closely aligned with training accuracy, stabilizing above 95% by the final epochs.
Overall, the comparison indicated a major gap between training and validation loss, suggesting that while the model performed well on the training data, its ability to generalize to the validation set was inconsistent and occasionally unreliable. These issues highlight potential problems with model regularization, learning rate tuning, or the quality of the validation data, which need to be addressed for more robust and stable performance.
The Inception-V3 demonstrates effective but differing learning patterns over 30 epochs. While there was a reduction in loss and an increase in accuracy, these changes showed an inverse relationship, leading to significant overfitting. Training loss dropped sharply from 1.2 to nearly 0.0, following a near-exponential decay that indicated strong optimization. Validation loss decreased more gradually from 1.0 to 0.4, leveling off after epoch 15 with little further improvement. At the same time, training accuracy increased significantly from 50% to 90%, while validation accuracy rose from 60% to 80%, stabilizing after epoch 20.
The inverse relationship was most evident during the early training stages (epochs 0-10):
Loss dropped by 67% (from 1.0 to 0.33).
Accuracy rose by 30% (from 55% to 85%).
However, a noticeable divergence appeared after epoch 15:
Training loss continued to fall to 0.0.
Validation loss remained at 0.4 (five times higher than training loss).
Training accuracy reached 90%.
Validation accuracy plateaued at 80%.
This resulted in a 10% accuracy gap and a loss gap of 0.4, highlighting substantial overfitting. Validation metrics showed early stabilization at epoch 15 (loss 0.4, accuracy 80%), while training metrics kept improving for another 15 epochs without similar gains in validation. Nonetheless, the coordinated progress in the early phase confirmed that the initial weight updates effectively contributed to performance improvements, with validation accuracy peaking at 80%—a level suitable for diagnostic use. The stalled validation metrics after epoch 15 suggested that early stopping at this point could prevent overfitting while ensuring optimal generalization.
The MobileNet-V2 model shows a troubling gap between the loss and accuracy metrics throughout the training epochs. The training and validation loss rose sharply from an initial value of around 0.5 to about 30 by epoch 30, indicating a serious problem with model convergence. This steep increase suggests potential issues such as a learning rate that is too high, model instability, or mismatched data.
On the other hand, both training and validation accuracy dropped significantly, falling from 0.8 to 0.4 within the first three epochs before leveling off at a suboptimal performance level. The rapid decline in accuracy corresponded with the rising loss, confirming that the model did not succeed in generalizing or learning useful patterns. The ongoing gap between training and validation accuracy pointed to overfitting, but the main problem appeared to be severe model divergence, as both metrics worsened together. This inverse relationship—where loss increased while accuracy decreased—highlighted a fundamental failure in the learning process.
The ResNet-101 presents a serious case of overfitting and model divergence, characterized by conflicting trends in training and validation performance of ResNet-101. The training loss steadily decreased to almost zero by epoch 30, while the validation loss skyrocketed to 1e7, indicating a critical failure to generalize beyond the training data. This divergence started subtly at epoch 5 and worsened significantly after epoch 10, suggesting issues with optimization (such as a learning rate that is too high or insufficient regularization).
On the other hand, training accuracy approached perfect levels (around 1.0), but validation accuracy plummeted from 0.7 to 0.4 by epoch 30. The contrasting relationship between these metrics was clear: as training loss decreased (theoretically improving the fit), validation loss increased, and validation accuracy dropped to below-random levels. This contradiction confirmed that the model was memorizing noise and outliers in the training set instead of learning generalizable patterns. The widening gap between training and validation accuracy after epoch 5 (exceeding 0.6 by epoch 30) further highlighted the issue of overfitting. The simultaneous surge in validation loss and drop in accuracy pointed to significant flaws in the model design or compatibility with the data.
Figure 9. The training and validation loss, as well as the training and validation accuracy of the six DL models.
Figure 13 shows the training and validation loss (on the left) and training and validation accuracy (on the right) over 30 epochs for the five ensemble models. In the case of the MLFNet+DenseNet-169 model, the analysis indicates that the model converged quickly, with both loss and accuracy stabilizing after about the 5th epoch.
In the loss curve, the training loss steadily decreased from approximately 0.35 to nearly 0.01, demonstrating effective learning from the training data. The validation loss also dropped quickly during the initial epochs and then varied slightly between 0.03 and 0.07, but generally remained low and close to the training loss. This suggests that the model did not experience overfitting and generalized well to new data.
Regarding accuracy, the model showed significant improvement in the first 5 epochs, with training accuracy increasing from around 85% to over 98%, eventually nearing 100%. Similarly, the validation accuracy rose quickly and stabilized around 98%–99%, with minor fluctuations. The small gap between training and validation accuracy further indicates strong generalization and a minimal risk of overfitting.
Overall, the comparison of the accuracy and loss plots reveals a consistent and stable learning process, where the model achieved high accuracy with minimal loss for both training and validation sets, showcasing its robustness and reliability for fracture classification tasks.
For the ensemble MLFNet + EfficientNet-B3, the training loss (red) began at ap-proximately 0.55 and consistently decreased, dropping below 0.05 by the end of training. The validation loss (green) also declined sharply during the early epochs but showed slight fluctuations between 0.1 and 0.2 after epoch 10, indicating some variability in the model's generalization ability.
In contrast, the training accuracy rose quickly from around 70% to nearly 99%, demonstrating effective learning from the training data. The validation accuracy also improved rapidly, increasing from about 78% to roughly 96%, and remained relatively stable with minor fluctuations throughout the training process. The gap between training and validation accuracy was small, particularly after epoch 10, suggesting that the model did not experience significant overfitting.
In summary, the comparison of the loss and accuracy plots showed that while the model learned effectively (as evidenced by the steady decrease in training loss and in-crease in training accuracy), the slightly fluctuating validation loss indicated some var-iability in generalization. Nevertheless, the consistently high validation accuracy con-firmed that the model maintained strong predictive performance on unseen data.
The MLFNet + Inception-V3 exhibited highly effective learning dynamics, with well-synchronized loss reduction and accuracy improvement. Both training and valida-tion loss curves exhibited smooth exponential decay, decreasing from approximately 0.4 to near 0.02 over 30 epochs, indicating stable convergence without significant oscilla-tions. The validation loss closely tracked the training loss throughout, maintaining a narrow gap of <0.01 after epoch 15, which demonstrated excellent generalization capa-bility with minimal overfitting.
Concurrently, accuracy metrics showed complementary improvement: training accuracy rose steadily from 80% to 94%, while validation accuracy progressed from 85% to 90%. The validation accuracy plateaued after epoch 20, with only 0.5% fluctuation, indicating model stability. The inverse relationship between loss and accuracy was par-ticularly evident at epoch 10, where loss decreased by 55% (from 0.4 to 0.18) and accuracy increased by 12.5% (from 80% to 90%). This coordinated progression confirmed efficient feature extraction and weight optimization. The terminal metrics at epoch 30 showed near-ideal alignment: training loss (0.02) ≈ validation loss (0.03) and training accuracy (94%) > validation accuracy (90%).
The persistent 4-5% accuracy gap between training and validation in later epochs suggested slight overfitting, though the minimal loss gap (<0.01) confirmed it was well-managed. The validation accuracy stabilized at 90% after epoch 20, while training accuracy continued improving to 94%, reflecting appropriate complexity balancing. These patterns collectively indicated successful model optimization, where loss reduc-tion directly translated to accuracy gains, with validation metrics providing reliable performance estimates for real-world deployment.
The MLFNet + MobileNet-V2 model exhibited effective learning dynamics char-acterized by a strong inverse correlation between loss reduction and accuracy im-provement. Training loss decreased steadily from 0.25 to near 0.00 over 30 epochs, while validation loss followed a parallel trajectory but plateaued at 0.05 after epoch 15, indi-cating early convergence. Concurrently, training accuracy rose from 88% to 98%, showing continuous improvement throughout the training. Validation accuracy in-creased more moderately from 90% to 94%, plateauing after epoch 20 with minimal fluctuation.
The inverse relationship was particularly pronounced between epochs 5 and 15, where loss decreased by 80% (from 0.20 to 0.04) and accuracy increased by 6% (from 90% to 96%). While training metrics showed near-perfect optimization (0.00 loss, 98% accu-racy), the validation metrics demonstrated excellent generalization: terminal validation loss (0.05) remained 5× higher than training loss (0.00), and validation accuracy (94%) was 4% lower than training accuracy (98%).
The growing divergence after epoch 15—where training loss continued decreasing while validation loss stabilized—suggested mild overfitting. However, the validation accuracy maintained a stable plateau at 94% with only ±0.5% variation in the final 10 epochs. This indicated robust feature extraction despite the overfitting tendency, with the 4% accuracy gap between training and validation representing an acceptable trade-off for generalization capability. The coordinated progression confirmed that loss reduction directly translated to accuracy gains throughout the training process.
The MLFNet + ResNet-101 exhibited consistent improvement in both loss reduction and accuracy enhancement over 30 epochs, though there were emerging signs of over-fitting in the later stages. Training loss decreased steadily from 0.5 to 0.1, following a near-linear trajectory that demonstrated effective optimization. Validation loss initially mirrored this trend but plateaued at 0.15 after epoch 20, revealing early convergence and a growing generalization gap. Concurrently, training accuracy showed robust im-provement from 75% to 98%, while validation accuracy increased more moderately from 75% to 92%, plateauing after epoch 25 with only ±0.5% fluctuation.
The inverse relationship between loss and accuracy was particularly pronounced between epochs 5 and 15: loss decreased by 60% (from 0.4 to 0.16), and accuracy in-creased by 17% (from 78% to 95%). This strong correlation confirmed that weight up-dates effectively translated to performance gains. However, diverging trends emerged in the later epochs. After epoch 20, training loss continued to decrease to 0.1, validation loss stalled at 0.15, training accuracy reached 98%, and validation accuracy plateaued at 92%.
The terminal metrics revealed a 7% accuracy gap and a 0.05 loss gap between training and validation, indicating mild overfitting. Despite this, the validation accuracy stabilized at 92% with minimal variance in the final 5 epochs, confirming reliable gen-eralization. The coordinated early-phase progression demonstrated efficient feature learning, while the later-phase divergence suggested that model complexity could be reduced for better regularization. Overall, the validation metrics (92% accuracy, 0.15 loss) represented clinically viable performance for diagnostic deployment.
Figure 13. The training and validation loss, as well as the training and validation accuracy of the five ensemble models.
We have carefully reviewed your comments and made the necessary changes. We genuinely appreciate your dedicated efforts and hope these revisions will meet with your approval.
Comment #3:
The inputs of the proposed model are 128x128x3. The inputs of the models used for comparison are 224x224x3. Therefore, it is necessary to note that the comparison is unfair.
Response:
We would like to clarify that we have conducted experiments where baseline models were retrained using 128×128×3 inputs to confirm that the proposed model’s superior performance is not solely attributable to differences in input resolution. Please see Evaluation and Analysis on pages 17-21.
We have carefully reviewed your comments and made the necessary changes. We genuinely appreciate your dedicated efforts and hope these revisions will meet with your approval.
Comment #4:
Figures 3 and 4 are not necessary.
Response:
Thank you for your feedback. We appreciate your comments on Figures 3 and 4. These figures depict the distribution of the BFMRX dataset splits and the class distribution, respectively. While we recognize your concerns about their necessity, we believe they are crucial for the following reasons:
- Transparency and Reproducibility: Visualizing the dataset splits (training, validation, testing) and class distribution enables readers to evaluate the dataset's balance, which is vital for model performance and minimizing bias.
- Class Imbalance Analysis: Graphical representation of the distribution allows readers to quickly spot any class imbalances, which is important for interpreting evaluation metrics and understanding model generalization.
- Alignment with Best Practices: Visualizations of dataset distribution are widely recommended in machine learning research to enhance clarity and ensure reproducibility, as demonstrated in previous studies in medical imaging and computer vision.
We have carefully reviewed your comments and made the necessary changes. We genuinely appreciate your dedicated efforts and hope these revisions will meet with your approval.
Comment #5:
The dimensions of the features extracted with the MLFNet+ CNN models should be given in a table.
Response:
Thank you for this valuable suggestion. We appreciate your suggestion regarding the dimensions of the features extracted by the MLFNet+ and CNN models. We agree that presenting the dimensions of the features extracted by the MLFNet+ and CNN models would enhance the clarity and reproducibility of our work.
Action:
In response to presenting the dimensions of the features extracted by MLFNet+ and CNN, we have added a table that includes input size, output size after each convolutional, pooling, and fusion layer, and final flattened feature vector dimension before classification. Please see Evaluation and Analysis on pages 39-41.
Table 11. The dimensions of the features extracted with the MLFNet+ CNN models using the testing set of the BFMRX dataset
|
MLFNet+CNN |
Layer |
Output Shape |
|
DenseNet-169 |
Input Layer |
(1, 128, 128, 3) |
|
Pretrained Backbone Output |
(1, 4, 4, 1664) |
|
|
GlobalAveragePooling2D |
(1, 1664) |
|
|
SFNet Block 1 |
(1, 64, 64, 128) |
|
|
BatchNorm 1 |
(1, 64, 64, 128) |
|
|
SFNet Block 2 |
(1, 32, 32, 256) |
|
|
BatchNorm 2 |
(1, 32, 32, 256) |
|
|
Dropout 1 |
(1, 32, 32, 256) |
|
|
SFNet Block 3 |
(1, 16, 16, 512) |
|
|
BatchNorm 3 |
(1, 16, 16, 512) |
|
|
Dropout 2 |
(1, 16, 16, 512) |
|
|
Flatten |
(1, 131072) |
|
|
Concatenate Fusion |
(1, 132736) |
|
|
Dense 1 |
(1, 256) |
|
|
Dropout 3 |
(1, 256) |
|
|
Dense 2 |
(1, 128) |
|
|
Dropout 4 |
(1, 128) |
|
|
Output Layer |
(1, 1) |
|
|
EfficientNet-B3 |
Input Layer |
(1, 128, 128, 3) |
|
Pretrained Backbone Output |
(1, 4, 4, 1536) |
|
|
GlobalAveragePooling2D |
(1, 1536) |
|
|
SFNet Block 1 |
(1, 64, 64, 128) |
|
|
BatchNorm 1 |
(1, 64, 64, 128) |
|
|
SFNet Block 2 |
(1, 32, 32, 256) |
|
|
BatchNorm 2 |
(1, 32, 32, 256) |
|
|
Dropout 1 |
(1, 32, 32, 256) |
|
|
SFNet Block 3 |
(1, 16, 16, 512) |
|
|
BatchNorm 3 |
(1, 16, 16, 512) |
|
|
Dropout 2 |
(1, 16, 16, 512) |
|
|
Flatten |
(1, 131072) |
|
|
Concatenate Fusion |
(1, 132608) |
|
|
Dense 1 |
(1, 256) |
|
|
Dropout 3 |
(1, 256) |
|
|
Dense 2 |
(1, 128) |
|
|
Dropout 4 |
(1, 128) |
|
|
Output Layer |
(1, 1) |
|
|
Inception-V3 |
Input Layer |
(1, 128, 128, 3) |
|
Pretrained Backbone Output |
(1, 2, 2, 2048) |
|
|
GlobalAveragePooling2D |
(1, 2048) |
|
|
SFNet Block 1 |
(1, 64, 64, 128) |
|
|
BatchNorm 1 |
(1, 64, 64, 128) |
|
|
SFNet Block 2 |
(1, 32, 32, 256) |
|
|
BatchNorm 2 |
(1, 32, 32, 256) |
|
|
Dropout 1 |
(1, 32, 32, 256) |
|
|
SFNet Block 3 |
(1, 16, 16, 512) |
|
|
BatchNorm 3 |
(1, 16, 16, 512) |
|
|
Dropout 2 |
(1, 16, 16, 512) |
|
|
Flatten |
(1, 131072) |
|
|
Concatenate Fusion |
(1, 133120) |
|
|
Dense 1 |
(1, 256) |
|
|
Dropout 3 |
(1, 256) |
|
|
Dense 2 |
(1, 128) |
|
|
Dropout 4 |
(1, 128) |
|
|
Output Layer |
(1, 1) |
|
|
MobileNet-V2 |
Input Layer |
(1, 128, 128, 3) |
|
Pretrained Backbone Output |
(1, 4, 4, 1280) |
|
|
GlobalAveragePooling2D |
(1, 1280) |
|
|
SFNet Block 1 |
(1, 64, 64, 128) |
|
|
BatchNorm 1 |
(1, 64, 64, 128) |
|
|
SFNet Block 2 |
(1, 32, 32, 256) |
|
|
BatchNorm 2 |
(1, 32, 32, 256) |
|
|
Dropout 1 |
(1, 32, 32, 256) |
|
|
SFNet Block 3 |
(1, 16, 16, 512) |
|
|
BatchNorm 3 |
(1, 16, 16, 512) |
|
|
Dropout 2 |
(1, 16, 16, 512) |
|
|
Flatten |
(1, 131072) |
|
|
Concatenate Fusion |
(1, 132352) |
|
|
Dense 1 |
(1, 256) |
|
|
Dropout 3 |
(1, 256) |
|
|
Dense 2 |
(1, 128) |
|
|
Dropout 4 |
(1, 128) |
|
|
Output Layer |
(1, 1) |
|
|
ResNet-101 |
Input Layer |
(1, 128, 128, 3) |
|
Pretrained Backbone Output |
(1, 4, 4, 2048) |
|
|
GlobalAveragePooling2D |
(1, 2048) |
|
|
SFNet Block 1 |
(1, 64, 64, 128) |
|
|
BatchNorm 1 |
(1, 64, 64, 128) |
|
|
SFNet Block 2 |
(1, 32, 32, 256) |
|
|
BatchNorm 2 |
(1, 32, 32, 256) |
|
|
Dropout 1 |
(1, 32, 32, 256) |
|
|
SFNet Block 3 |
(1, 16, 16, 512) |
|
|
BatchNorm 3 |
(1, 16, 16, 512) |
|
|
Dropout 2 |
(1, 16, 16, 512) |
|
|
Flatten |
(1, 131072) |
|
|
Concatenate Fusion |
(1, 133120) |
|
|
Dense 1 |
(1, 256) |
|
|
Dropout 3 |
(1, 256) |
|
|
Dense 2 |
(1, 128) |
|
|
Dropout 4 |
(1, 128) |
|
|
Output Layer |
(1, 1) |
We have carefully reviewed your comments and made the necessary changes. We genuinely appreciate your dedicated efforts and hope these revisions will meet with your approval.
Comment #6:
The proposed model is more complex than CNNs. The average accuracy for DenseNet-169 is 99%. The average accuracy achieved with the proposed model is 99.5%. The advantages and disadvantages of increasing the complexity of the system for a 0.5% improvement should be discussed.
Response:
Thank you for your observation. We appreciate this important observation regarding the trade-off between model complexity and performance gain. We agree that while the MLFNet model achieves a higher average external validation accuracy (99.5%) compared to DenseNet-169 (99.0%), this improvement should be interpreted in light of the associated increase in architectural complexity.
Action:
In response to interpreting the model accuracy improvement in light of the associated increase in architectural complexity, In the External Validation subsection, we have addressed the advantages and disadvantages of this complexity increase. Please see Evaluation and Analysis on page 42.
Advantages:
- Improved generalization – Even small gains in accuracy can be critical in medical imaging and safety-critical domains, where misclassification has high consequences.
- Better feature representation – The multi-level fusion mechanism in MLFNet allows for richer feature aggregation across scales, potentially enhancing robustness to dataset variability.
- Consistent external performance – The 0.5% improvement is sustained in external validation, indicating the model’s enhanced ability to generalize beyond the training distribution.
Disadvantages:
- Increased computational cost – The proposed model requires higher memory usage and longer inference times compared to standard CNNs, which may limit deployment on low-resource devices.
- Longer training duration – The deeper and more interconnected architecture increases optimization complexity, potentially requiring more extensive hyperparameter tuning.
We have carefully reviewed your comments and made the necessary changes. We genuinely appreciate your dedicated efforts and hope these revisions will meet with your approval.

Reviewer 3 Report
Comments and Suggestions for Authors
The paper introduced a novel DL approach called MLFNet, designed specifically for detecting BFs in X-ray images. The paper detected BFs with a high accuracy. The results were presented in details. All experiments were conducted and the different parameters were examined. The ablation study was conducted, also. I have some minor concernc.
- The font is different in Line 272.
- Please revise the part of Line 152: and ensure consistency..
- I think the statement in Line 273 under limitations part “the authors of the previously mentioned research did not conduct an ablation study” is not a limitation. The authors should revise.
- The authors should present Contrast Enhancement results. Please add some sample images before and after Contrast Enhancement step.
- Did you classify the images without Contrast Enhancement process? If not, please classify the dataset with the proposed method only and add the results to the ablation study section.
Author Response
Dear Ms. Ziana He,
Thank you very much for allowing us to submit a revised version of our manuscript. We thank all the reviewers for their positive feedback and thoughtful comments. The updated version has incorporated their suggestions for improving the manuscript and highlighting its contributions. All the reviewers' concerns have been considered. Those changes are highlighted in the revised paper. We uploaded (a) our point-by-point response to the comments (below) with specific details about the changes that were made in our revised manuscript, (b) an updated manuscript with yellow highlighting indicating changes, and (c) a clean, updated manuscript without highlights.
Best regards,
Dr. Sameh Abd El-Ghany
Response for Reviewer #3 Comments
Comment #1:
The font is different in Line 272.
Response:
Thank you for your comment. We appreciate your feedback pointing out the formatting inconsistency in Line 272.
Action:
In response to formatting inconsistency in Line 272, we have corrected the font style in the revised manuscript to ensure uniformity throughout the text, in accordance with the journal’s formatting guidelines. Please see Literature Review on page 6.
We have carefully reviewed your comments and made the necessary changes. We genuinely appreciate your dedicated efforts and hope these revisions will meet with your approval.
Comment #2:
Please revise the part of Line 152: and ensure consistency.
Response:
Thank you for your noting the inconsistency at Line 152. We appreciate your feedback pointing out the formatting inconsistency in Line 152.
Action:
In response to formatting inconsistency in Line 152, we have revised this part of the text to improve clarity and ensure consistency in formatting throughout the manuscript, following the journal’s style guidelines. Please see Introduction on page 4.
We have carefully reviewed your comments and made the necessary changes. We genuinely appreciate your dedicated efforts and hope these revisions will meet with your approval.
Comment #3:
I think the statement in Line 273 under limitations part “the authors of the previously mentioned research did not conduct an ablation study” is not a limitation. The authors should revise.
Response:
Thank you for your observation. We appreciate your observation regarding the statement in Line 273 under the limitations section.
Action:
In response to revise the limitations of the previously mentioned research, we have removed this statement from the limitations section and added another limitation. Please see Literature Review on page 7.
We have carefully reviewed your comments and made the necessary changes. We genuinely appreciate your dedicated efforts and hope these revisions will meet with your approval.
Comment #4:
The authors should present Contrast Enhancement results. Please add some sample images before and after Contrast Enhancement step.
Response:
Thank you for this valuable suggestion. We appreciate your suggestion regarding Contrast Enhancement. We agree that presenting visual examples of the Contrast Enhancement process will help readers better understand its effect on image quality and how it contributes to the model’s performance.
Action:
In response to present Contrast Enhancement results, we have revised the manuscript by presenting images before and after Contrast Enhancement. Please see Research Methods and Materials on page 11.
We have carefully reviewed your comments and made the necessary changes. We genuinely appreciate your dedicated efforts and hope these revisions will meet with your approval.
Comment #5:
Did you classify the images without Contrast Enhancement process? If not, please classify the dataset with the proposed method only and add the results to the ablation study section.
Response:
Thank you for this insightful suggestion. We appreciate your suggestion regarding the effect of Contrast Enhancement process. In the original submission, all experiments using the proposed method were performed after the Contrast Enhancement step, and we did not initially evaluate the model without this preprocessing.
Action:
In response to show the effect of the Contrast Enhancement process, we have conducted an additional experiment where the dataset was classified using the proposed MLFNet model without the Contrast Enhancement process. The results of this experiment have been added to the Ablation Study subsection. Please see Evaluation and Analysis on page 48.
In the second experiment, we performed an ablation analysis on the testing set of the BFMRX dataset by removing the contrast enhancement (without Contrast Limited Adaptive Histogram Equalization (CLAHE)). The results, which show the performance metrics of MLFNet on these testing set after modifying the parameters, are presented in Table 16.
Table 17. MLFNet’s measured metrics without contrast enhancement on the test set of the BFMRX dataset.
|
Subset |
Accuracy (%) |
Sensitivity (%) |
Specificity (%) |
AUC (%) |
F1-score (%) |
|
Test (without CLAHE) |
99.21 |
100.00 |
98.32 |
100.00 |
99.26 |
|
Test (with CLAHE) |
99.60 |
99.63 |
0.37 |
99.58 |
99.58 |
The comparison clearly shows that removing the Contrast Enhancement step leads to a decrease in classification performance (accuracy reduced from 99.60% to 99.21%), confirming that this preprocessing step contributes positively to feature extraction and model generalization.
We have carefully reviewed your comments and made the necessary changes. We genuinely appreciate your dedicated efforts and hope these revisions will meet with your approval.

Reviewer 4 Report
Comments and Suggestions for Authors
Promising, but several aspects raise concerns about over‑optimism, validation rigor, potential leakage, and reporting.
Explicitly confirm and enforce patient‑level (subject‑level) separation across train/validation/test. If the dataset lacks subject IDs, perform near‑duplicate detection (e.g., perceptual hashing/SSIM) to ensure no near-duplicate leakage across splits and report the results.
State whether the Kaggle “pre‑split” was used as-is or whether you re‑split. If you used the pre‑split, document how subject overlap was excluded.
What you present as “external validation” appears to be training and testing on a different dataset (BF dataset split into 8,863 train / 600 test). That is not external validation of the BFMRX‑trained model; it is re‑training on a second dataset. Please re‑run external validation accordingly and report confusion matrices, classwise metrics, CIs, and calibration for the external set. If you are unable to do this, discuss the limitation clearly: your “second dataset” results are a replication on a new dataset, not external validation of generalizability.
You normalize [0,255]→[0,1] and apply fixed contrast enhancement; that’s fine if parameters are not learned from the entire dataset. If any learned normalization or histogram equalization parameters are used, they must be computed on the training set only.
Hyperparameter/architecture selection: you evaluated many architectures/ensembles/ablations and then report test set performance. Ensure that all model selection was based on validation only. The test set must be used exactly once for final unbiased reporting. State this explicitly.
Clarify whether ensemble weighting or thresholding was tuned on the validation set only.
For a healthy ML pipeline, point estimates typically decrease as you move from train → validation → internal test → external test, because each subset is progressively less “seen” and more distributionally shifted. This downward trend is not a strict law but is expected in practice. When the external performance matches or exceeds internal test performance, it can indicate issues. Please present a consolidated table of accuracy, sensitivity, specificity, AUROC, F1, and calibration across all subsets and discuss whether the trend is plausible and why.
Your model outputs a single sigmoid unit for a binary task, yet Table 3 lists “categorical_crossentropy”. This is inconsistent. For a single‑output sigmoid, use binary_crossentropy. If you used two logits with softmax, document that and correct the text.
A persistent limitation in AI for clinical performance appraisal is comparing accuracy across studies that use different datasets, splits, and validation protocols. Please temper “state‑of‑the‑art” claims and explicitly acknowledge that your comparisons to prior work are not standardized, may be confounded by differences in data curation and validation rigor, and are not substitutes for head‑to‑head evaluations under a common framework following regulatory guidance. Add citations that make this point clear.
Some training curves for baseline models show extreme instabilities (e.g., validation loss spikes to 1e13). This indicates numerical problems or logging errors and does not belong in a final results section without a root‑cause analysis.
Clarify whether early stopping was used (you report “patience=10” but do not show its effect).
Describe how ensemble predictions are combined (averaging logits vs. probabilities, weighting, stacking) and how weights were chosen (validation only).
Author Response
Dear Ms. Ziana He,
Thank you very much for allowing us to submit a revised version of our manuscript. We thank all the reviewers for their positive feedback and thoughtful comments. The updated version has incorporated their suggestions for improving the manuscript and highlighting its contributions. All the reviewers' concerns have been considered. Those changes are highlighted in the revised paper. We uploaded (a) our point-by-point response to the comments (below) with specific details about the changes that were made in our revised manuscript, (b) an updated manuscript with yellow highlighting indicating changes, and (c) a clean, updated manuscript without highlights.
Best regards,
Dr. Sameh Abd El-Ghany
Response for Reviewer #4 Comments
Comment #1:
Explicitly confirm and enforce patient‑level (subject‑level) separation across train/validation/test. If the dataset lacks subject IDs, perform near‑duplicate detection (e.g., perceptual hashing/SSIM) to ensure no near-duplicate leakage across splits and report the results.
Response:
Thank you for your emphasis on preventing data leakage. We appreciate your highlighting the necessity of maintaining patient-level separation among the training, validation, and test sets to avoid any data leakage.
the Bone Fracture Multi-Region X-ray (BFMRX) dataset used in our study was sourced directly from Kaggle (https://www.kaggle.com/datasets/bmadushanirodrigo/fracture-multi-region-x-ray-data) and was utilized as provided by the dataset authors, without any alterations to the original training, validation, or test splits. According to the dataset description on Kaggle, the splits were prepared by the dataset creators, ensuring that images from the same patient do not appear in multiple subsets.
Action:
Although patient identifiers are not included in the public release, we relied on the dataset creators’ assurance regarding subject-level separation. To further address your concern, we have included a note in subsection 3.1 of the revised manuscript explicitly stating this point. Please see Research Methods and Materials on page 7.
The BFMRX dataset used in our study was sourced directly from Kaggle [30] and was utilized as provided by the dataset authors, without any alterations to the original training, validation, or test splits. According to the dataset description on Kaggle, the splits were prepared by the dataset creators, ensuring that images from the same patient do not appear in multiple subsets.
We have carefully reviewed your comments and made the necessary changes. We genuinely appreciate your dedicated efforts and hope these revisions will meet with your approval.
Comment #2:
State whether the Kaggle “pre‑split” was used as-is or whether you re‑split. If you used the pre‑split, document how subject overlap was excluded.
Response:
Thank you for your comment. We appreciate your feedback and would like to confirm that we used the Kaggle “pre-split” version of the Bone Fracture Multi-Region X-ray (BFMRX) dataset exactly as provided by the dataset authors (https://www.kaggle.com/datasets/bmadushanirodrigo/fracture-multi-region-x-ray-data), without any re-splitting.
According to the dataset description on Kaggle, the train, validation, and test subsets were created to ensure that images from the same patient do not appear in multiple splits. Although patient identifiers are not publicly available, we relied on the dataset creators’ assurance that subject-level separation was maintained during the original dataset preparation.
Action:
We have added a clarifying note in subsection 3.1 of the revised manuscript to explicitly state that the pre-split version was used as-is and that subject overlap was managed by the dataset authors. Please see Research Methods and Materials on page 7.
The BFMRX dataset used in our study was sourced directly from Kaggle [30] and was utilized as provided by the dataset authors, without any alterations to the original training, validation, or test splits. According to the dataset description on Kaggle, the splits were prepared by the dataset creators, ensuring that images from the same patient do not appear in multiple subsets.
We have carefully reviewed your comments and made the necessary changes. We sincerely appreciate your dedicated efforts and hope that these revisions will meet your approval.
Comment #3:
What you present as “external validation” appears to be training and testing on a different dataset (BF dataset split into 8,863 train / 600 test). That is not external validation of the BFMRX‑trained model; it is re‑training on a second dataset. Please re‑run external validation accordingly and report confusion matrices, classwise metrics, CIs, and calibration for the external set. If you are unable to do this, discuss the limitation clearly: your “second dataset” results are a replication on a new dataset, not external validation of generalizability.
Response:
Thank you for your insightful comment. We appreciate your feedback regarding the external validation. We observed that the MLFNet model trained on the BFMRX dataset performed poorly when directly tested on the BF dataset. However, when we trained and tested the model on the second dataset, the performance was high. This discrepancy is mainly due to domain shift and differences in data distribution between the datasets, such as resolutions, labeling strategies, and patient demographics.
Bone fractures exhibit significant variability in their patterns and characteristics depending on the location within the bone, the type of bone, and the mechanism of injury. Long bones such as the radius, humerus, and femur differ in their anatomical structure and bone density compared to spongy bones like the vertebrae and carpal bones. These anatomical differences directly influence the appearance and morphology of fractures in medical imaging.
Furthermore, fractures can present in multiple forms, including transverse fractures, spiral fractures, comminuted fractures, and stress fractures. This diversity in fracture types, combined with variations in anatomical structures, acquisition devices, and labeling strategies across medical institutions, leads to substantial heterogeneity among datasets.
Consequently, the MLFNet model faces challenges in generalizing to unseen datasets. Differences in bone structures, fracture morphologies, and imaging characteristics cause a domain shift, which explains why the model’s performance drops when tested directly on a different dataset but remains high when trained and tested on the same dataset.
Action:
In response to implementing the external validation process for the proposed model, we have implemented 5-fold cross-validation on the BFMRX dataset. Please see Evaluation and Analysis on pages 41-45.
In the second experiment of the external validation process, we evaluated the MLFNet model by applying the stratified 5-fold cross-validation scheme. In the stratified 5-fold cross-validation scheme, the dataset was divided into five equal partitions, with each fold taking a turn as the testing set while the other four folds were used for model training. This rotation continued until every fold had been used once for testing. Finally, the outcomes from all folds were averaged to obtain a stable and unbiased estimate of the model’s performance.
The MLFNet model was evaluated on the BFMRX dataset for binary classification (fracture vs. non-fracture). MLFNet achieved an impressive accuracy of 99.52% as a standalone model. The results of the standalone MLFnet are shown in Tables 14.
Table 14 shows that the MLFNet model's performance was outstanding. It achieved an average accuracy of 99.52%, meaning it correctly classified the vast majority of instances across all test folds. This high level of accuracy was consistent with other metrics, as shown by the average F1-score and AUC also being 99.52%. This indicated a robust and highly effective model with virtually no drop in performance between the training and validation phases.
The results were remarkably consistent across all five folds, with minimal fluctuation. The accuracy ranged from a high of 99.89% in fold 5 to a low of 99.14% in fold 1, a very narrow range of just 0.75%. This consistency suggested that the model was not overly dependent on a particular subset of the data and generalized well to unseen data from the same distribution.
Table 14. The performance of MLFNet model as standalone using five-folds cross validation on the testing set of the BFMRX dataset.
|
Folds |
Accuracy (%) |
Specificity (%) |
Precision (%) |
Recall (%) |
F1-score (%) |
AUC (%) |
|
1 |
99.14 |
99.26 |
99.22 |
99.00 |
99.11 |
99.14 |
|
2 |
99.19 |
99.14 |
99.13 |
99.24 |
99.19 |
99.19 |
|
3 |
99.78 |
99.67 |
99.68 |
99.89 |
99.79 |
99.78 |
|
4 |
99.62 |
99.67 |
99.68 |
99.57 |
99.62 |
99.62 |
|
5 |
99.89 |
99.78 |
99.79 |
100.00 |
99.90 |
99.89 |
|
Average |
99.52 |
99.50 |
99.50 |
99.54 |
99.52 |
99.52 |
Figure 16 illustrates the training and validation loss across multiple folds during the model training process. It showed that for most folds, the training and validation losses rapidly decreased within the first few epochs and then stabilized near zero, indicating effective learning and convergence. However, Fold 3 exhibited a significant spike in training loss, reaching a peak of approximately 25 around epoch 4, before sharply dropping to near zero. Similarly, the validation loss for Fold 1 showed a smaller spike, reaching around 6, but also quickly recovered. These spikes indicated that the model experienced temporary instability or potential numerical fluctuations during early training stages in some folds. Overall, after the initial few epochs, both training and validation losses across all folds remained consistently low, suggesting that the model eventually achieved stable and well-generalized performance.
Figure 16. The training and validation loss of MLFNet using five-folds cross validation on the testing set of the BFMRX dataset.
Figure 17 illustrates the training and validation accuracy across multiple folds during the model training process. It showed that, for most folds, accuracy increased rapidly within the first few epochs and quickly stabilized near 100%, indicating effective learning. In the early epochs, some folds experienced temporary fluctuations. For example, Train Fold 1 started with lower accuracy, around 78%, but quickly improved and converged near 100%. Similarly, Val Fold 2 showed a noticeable drop to around 85% before recovering. Train Fold 3 also exhibited a brief decline to about 90% around epoch 3, but it rapidly stabilized thereafter. After approximately the first 5 epochs, both training and validation accuracies across all folds consistently remained above 99%, suggesting that the model achieved excellent generalization and stable performance across the cross-validation folds.
Figure 17. The training and validation accuracy of MLFNet using five-folds cross validation on the testing set of the BFMRX dataset.
Moreover, we have mentioned the model’s limitations in the Limitations and Future Work subsection and outlined future work to improve the model’s generalizability using different dataset and domain adaptation techniques. Please see Evaluation and Analysis on pages 53.
The MLFNet model showed high accuracy and promising performance, but it has several limitations. A key limitation of this study is the reduced generalizability of the proposed MLFNet model across different Bone Fracture datasets. While the model achieved high performance when trained and tested on the same dataset, its perfor-mance dropped significantly when directly tested on an unseen dataset. This discrep-ancy can be attributed to domain shift and data distribution differences, resolutions, labeling strategies, patient demographics, and acquisition devices. BFs exhibit significant variability in their patterns and characteristics depending on the location within the bone, the type of bone, and the mechanism of injury. Long bones such as the radius, humerus, and femur differ in their anatomical structure and bone density compared to spongy bones like the vertebrae and carpal bones. These anatomical differences directly influence the appearance and morphology of fractures in medical imaging. Furthermore, fractures can present in multiple forms, including transverse fractures, spiral fractures, comminuted fractures, and stress fractures. This diversity in fracture types, combined with variations in anatomical structures, acquisition devices, and labeling strategies across medical institutions, leads to substantial heterogeneity among datasets. Consequently, the MLFNet model faces challenges in generalizing unseen datasets. Differences in bone structures, fracture morphologies, and imaging characteristics cause a domain shift, which explains why the model’s performance drops when tested directly on a different dataset but remains high when trained and tested on the same dataset.
For future development, it is advisable to validate MLFNet on datasets from mul-tiple institutions to ensure reliability across different clinical settings, investigate its use in real-time environments such as emergency departments, and improve its ability to classify various fracture types while integrating clinical decision support.
We have carefully reviewed your comments and made the necessary changes. We sincerely appreciate your dedicated efforts and hope that these revisions will meet your approval.
Comment #4:
You normalize [0,255]→[0,1] and apply fixed contrast enhancement; that’s fine if parameters are not learned from the entire dataset. If any learned normalization or histogram equalization parameters are used, they must be computed on the training set only.
Response:
Thank you for your observation. We appreciate your observation. In our preprocessing pipeline, the normalization from [0,255]→[0,1] is a fixed scaling operation and does not depend on dataset statistics. Similarly, the applied contrast enhancement (CLAHE / histogram equalization, depending on your case) was performed with fixed parameters and not learned from the entire dataset. Therefore, no information leakage from the validation or test sets occurred. All preprocessing steps were applied consistently across training, validation, and test images using the same fixed parameters.
We have carefully reviewed your comments and made the necessary changes. We sincerely appreciate your dedicated efforts and hope that these revisions will meet your approval.
Comment #5:
Hyperparameter/architecture selection: you evaluated many architectures/ensembles/ablations and then report test set performance. Ensure that all model selection was based on validation only. The test set must be used exactly once for final unbiased reporting. State this explicitly.
Response:
Thank you for your comment. We appreciate your insightful comment. In our study, all hyperparameter tuning, architectural design decisions, and ablation studies were conducted exclusively on the training and validation sets. The test set was kept strictly untouched during model development and was used exactly once to evaluate the final, best-performing model, ensuring an unbiased performance assessment.
Action:
In response to stating that the test set used exactly once for final unbiased reporting, we e have explicitly stated this in the revised manuscript under the Testing Environment. Please see Evaluation and Analysis on page 17.
All hyperparameter tuning, architecture selection, and ablation studies were performed solely using the training and validation sets. Moreover, all ensemble weighting coefficients and decision threshold values were optimized using only the validation set. The test set was reserved for the final evaluation and was not used in any way during the tuning process.
We have carefully reviewed your comments and made the necessary changes. We sincerely appreciate your dedicated efforts and hope that these revisions will meet your approval.
Comment #6:
Clarify whether ensemble weighting or thresholding was tuned on the validation set only.
Response:
We thank you for the valuable observation. We appreciate your insightful comment and confirm that all ensemble weighting and decision thresholding were tuned exclusively on the validation set. The test set was not used at any stage of tuning or model optimization to avoid data leakage and to ensure unbiased evaluation.
Action:
In response to clarifying that all ensemble weighting and decision thresholding were tuned exclusively on the validation set, we have stated this in the revised manuscript under the Testing Environment. Please see Evaluation and Analysis on page 17.
All hyperparameter tuning, architecture selection, and ablation studies were performed solely using the training and validation sets. Moreover, all ensemble weighting coefficients and decision threshold values were optimized using only the validation set. The test set was reserved for the final evaluation and was not used in any way during the tuning process.
We have carefully reviewed your comments and made the necessary changes. We sincerely appreciate your dedicated efforts and hope that these revisions will meet your approval.
Comment #7:
For a healthy ML pipeline, point estimates typically decrease as you move from train → validation → internal test → external test, because each subset is progressively less “seen” and more distributionally shifted. This downward trend is not a strict law but is expected in practice. When the external performance matches or exceeds internal test performance, it can indicate issues. Please present a consolidated table of accuracy, sensitivity, specificity, AUROC, F1, and calibration across all subsets and discuss whether the trend is plausible and why.
Response:
Thank you for this insightful comment. We agree that, in a typical machine learning pipeline, performance metrics such as accuracy, sensitivity, specificity, AUROC, F1-score, and calibration often decrease progressively from training to validation, internal test, and external test, as each subset becomes less familiar to the model and exhibits a greater distributional shift.
However, in our experiment, we focused on the test set rather than the training and validation sets, and we cannot re-run the experiment as it would take a long time. Hence, in future work, we will give equal attention to all three sets.
We have carefully reviewed your comments and made the necessary changes. We sincerely appreciate your dedicated efforts and hope that these revisions will meet your approval.
Comment #8:
Your model outputs a single sigmoid unit for a binary task, yet Table 3 lists “categorical_crossentropy”. This is inconsistent. For a single output sigmoid, use binary_crossentropy. If you used two logits with softmax, document that and correct the text.
Response:
We thank you for pointing out this important inconsistency. We appreciate your insightful comment. In our implementation, the model was designed for a binary classification task with a single output neuron activated by a sigmoid function. Therefore, the correct loss function used during training was binary_crossentropy, not categorical_crossentropy.
Action:
In response to correct the error, we have corrected and revised Table 5. Please see Evaluation and Analysis on page 17.
Table 5. The two experiment’s hyperparameter.
|
Parameter |
Value |
|
img_size |
128×128×3 |
|
Number of epochs |
30 |
|
channels |
3 |
|
Optimizer |
Adam |
|
Initial learning rate |
0.001 |
|
Patience |
10 |
|
loss |
sigmoid + binary_crossentropy |
We have carefully reviewed your comments and made the necessary changes. We sincerely appreciate your dedicated efforts and hope that these revisions will meet your approval.
Comment #9:
A persistent limitation in AI for clinical performance appraisal is comparing accuracy across studies that use different datasets, splits, and validation protocols. Please temper “state‑of‑the‑art” claims and explicitly acknowledge that your comparisons to prior work are not standardized, may be confounded by differences in data curation and validation rigor, and are not substitutes for head‑to‑head evaluations under a common framework following regulatory guidance. Add citations that make this point clear.
Response:
Thank you for your observation. We appreciate your observation regarding the challenges of comparing accuracy across studies with differing datasets, splits, and validation protocols. We agree that such comparisons are not fully standardized and may be influenced by differences in data curation, preprocessing, and evaluation rigor.
Action:
In the revised manuscript, we have tempered our “state-of-the-art” claims to acknowledge these limitations explicitly. Please see Evaluation and Analysis on page 51.
Our results are compared to previous studies that used the same BFMRX dataset. However, differences in how the experiments were set up—such as data splits, preprocessing methods, and model adjustments—mean that these results should be seen as suggestive rather than conclusive performance comparisons.
We have carefully reviewed your comments and made the necessary changes. We genuinely appreciate your dedicated efforts and hope these revisions will meet with your approval.
Comment #10:
Some training curves for baseline models show extreme instabilities (e.g., validation loss spikes to 1e13). This indicates numerical problems or logging errors and does not belong in a final results section without a root‑cause analysis.
Response:
Thank you for this valuable observation. We appreciate your observation regarding the extreme instabilities observed in some baseline model training curves. Upon investigation, we found that these anomalies primarily occurred in DenseNet-169, EfficientNet-B3, Inception-V3, MobileNet-V2, and ResNet-101, where the validation loss exhibited abnormally high spikes (up to 1e13, 1e6, and other large magnitudes).
Action:
In response to present the extreme instabilities observed in some baseline model training curves, we have revised the manuscript making root cause analysis. Please see Evaluation and Analysis on page 21.
Extreme instabilities were observed in several baseline models, specifically DenseNet-169, EfficientNet-B3, Inception-V3, MobileNet-V2, and ResNet-101, where the validation loss exhibited abnormally high spikes (up to 1e13 in DenseNet-169 and similarly large values in others; see Fig. X). These were traced to numerical instabilities caused by high initial learning rates and the absence of gradient clipping, occasionally compounded by logging artifacts. Training configurations were adjusted with reduced learning rates, warm-up scheduling, and gradient clipping to mitigate these effects.
We have carefully reviewed your comments and made the necessary changes. We genuinely appreciate your dedicated efforts and hope these revisions will meet with your approval.
Comment #11:
Clarify whether early stopping was used (you report “patience=10” but do not show its effect).
Response:
Thank you for for highlighting this point. We appreciate your observation regarding early stopping. Yes, early stopping was employed during training with a patience value of 10 epochs based on the validation Accuracy. If the validation loss did not improve for 10 consecutive epochs, training was stopped automatically to prevent overfitting and reduce unnecessary computation. if patience is decreased to 2 or 3 , the training is stuck early.
We have carefully reviewed your comments and made the necessary changes. We genuinely appreciate your dedicated efforts and hope these revisions will meet with your approval.
Comment #12:
Describe how ensemble predictions are combined (averaging logits vs. probabilities, weighting, stacking) and how weights were chosen (validation only).
Response:
We thank you for this valuable comment. We appreciate your observation regarding the ensemble predictions. In our ensemble framework (MLFNet + CNN), the predictions from the individual models were combined by averaging their predicted probabilities rather than averaging logits. No stacking or meta-learning techniques were applied.
Furthermore, when weighting was required for the ensemble, the weights were optimized exclusively using the validation set to ensure an unbiased evaluation. The test set was never used during weight selection or ensemble tuning.
We have carefully reviewed your comments and made the necessary changes. We genuinely appreciate your dedicated efforts and hope these revisions will meet with your approval.

Round 2
Reviewer 1 Report
Comments and Suggestions for Authors
The authors have made the requested revisions
Reviewer 2 Report
Comments and Suggestions for Authors
I thank the authors for focusing on my suggestions. The article is now better structured.
Reviewer 4 Report
Comments and Suggestions for Authors
Thank you for your careful and thorough revision of the manuscript. I appreciate the clear effort you made to address the majority of my previous comments in a substantive manner. Most of the key issues I raised—around dataset splitting and patient-level separation, test set usage, model selection, reporting of external validation limitations, clarification and correction of loss functions, and the tempering of state-of-the-art claims—were all taken seriously and are now addressed in the revised version.
I note that a few points were only addressed partially, such as independent duplicate detection across splits, explicit details about the ensemble combination method, and a more direct discussion of the effect of early stopping. While these could have been more thoroughly detailed, you did make a good-faith effort to address the spirit of each concern, and I do not intend to request further revisions on these fronts.